# Towards Graph Foundation Models: Learning Generalities Across Graphs via Task-trees

## Abstract

Foundation models aim to create general, cross-task, and cross-domain machine learning models by pretraining on large-scale datasets to capture shared patterns or concepts (generalities), such as contours, colors, textures, and edges in images, or tokens, words, and sentences in text. However, discovering generalities across graphs remains challenging, which has hindered the development of graph foundation models. To tackle this challenge, in this paper, we propose a novel approach to learn generalities across graphs via task-trees. Specifically, we first define the basic learning instances in graphs as task-trees and assume that the generalities shared across graphs are, at least partially, preserved in the task-trees of the given graphs. To validate the assumption, we first perform a theoretical analysis of task-trees in terms of stability, transferability, and generalization. We find that if a graph neural network (GNN) model is pretrained on diverse task-trees through a reconstruction task, it can learn sufficient transferable knowledge for downstream tasks using an appropriate set of fine-tuning samples. To empirically validate the assumption, we further instantiate the theorems by developing a cross-task, cross-domain graph foundation model named Graph generality Identifier on task-Trees (GIT). The extensive experiments over 30 graphs from five domains demonstrate the effectiveness of GIT in fine-tuning, in-context learning, and zero-shot learning scenarios. Particularly, the general GIT model pretrained on large-scale datasets can be quickly adapted to specific domains, matching or even surpassing expert models designed for those domains.

## 1 Introduction

Foundation models have gained prominence in the era of artificial general intelligence as a general-purpose, cross-task, and cross-domain machine learning approach. These models are exemplified by large language models (LLMs) for text (Achiam et al., 2023; Touvron et al., 2023) and large vision models (LVMs) for images (He et al., 2022; Yuan et al., 2021). Pretrained on large-scale datasets, they capture generalizable and transferable knowledge, including contours, colors, textures, and edges in images, as well as tokens, words, and sentences in text. These patterns and concepts represent modality-specific generalities, allowing the models to adapt rapidly to new downstream tasks via in-context learning (Xie et al., 2022) or zero-shot learning (Wei et al., 2021).

Despite the success of foundation models across various modalities, their development of graph-structured data remains in its infancy (Liu et al., 2024). This is primarily due to the high degree of variability among graph-structured datasets (Mao et al., 2024), making it challenging to identify shared generalities across graphs. In particular, graphs from different domains typically represent distinct phenomena; for example, social networks capture relationships between people (Freeman, 2004), while molecular networks depict the structures of molecules (Zeng et al., 2022). These distinctions are evident not only in the differences between feature and label spaces (Huang et al., 2023; Liu et al., 2024) but also in structural heterogeneity (Qiu et al., 2020; Wang et al., 2024b). Additionally, different graph-related tasks often do not share basic learning instances—such as nodes in node-level tasks and entire graphs in graph-level tasks—making it difficult to use a single model for diverse tasks (Mao et al., 2024). The above challenges make it extremely difficult to develop a graph foundation model (GFM) that can identify generalities applicable across graphs.

*Is it possible to identify generalities across graphs?* Despite the challenges, some researchers have attempted to uncover shared generalities across graphs, which can be mainly categorized into two groups. (1) One line of works draws on graph theory. In particular, they use the concept of graphon (Ruiz et al., 2020) to describe transferable patterns across graphs. If two graphs are generated from the same graphon, they are expected to share key topological properties, which can result in high transferability between them (Ruiz et al., 2020; Cao et al., 2023). However, the strong assumptions underlying graphon theory are often unrealistic for real-world graphs (Levie et al., 2021), and even when the assumptions hold, finding the graphon shared from a large set of graphs presents another significant challenge. (2) Another approach involves leveraging substructures that are present across graphs, such as triangles, stars, and $k$-cliques (Zhao et al., 2023; Mao et al., 2024). For instance, triangle structures frequently appear in citation networks, social networks, and molecular networks, albeit with varying semantics. Based on this observation, some works (Sun et al., 2023; Liu et al., 2024) sample subgraphs consisting of these substructures (as shown in Figure 1, Left) and then use a graph neural network (GNN) to encode the subgraph-level embeddings for prediction. However, identifying which substructures are shared across distinct graphs remains difficult. Even if useful substructures can be identified, message-passing GNNs are proved to struggle with capturing such basic substructures in learning (sub)graph embeddings (Garg et al., 2020; Esser et al., 2021; Zhang et al., 2024a), limiting the practicality of these methods.

Given the limitations of existing approaches, we propose a novel perspective to answer the question by focusing on the learning process of GNNs. In message-passing GNNs (Kipf & Welling, 2017; Hamilton et al., 2017), predictions are made based on the so-called ***task-relevant nodes*** in the graph. For node-level tasks, the task-relevant node is the node itself; for edge-level tasks, it includes the start and end nodes of the edge; and for graph-level tasks, all nodes in the graph are task-

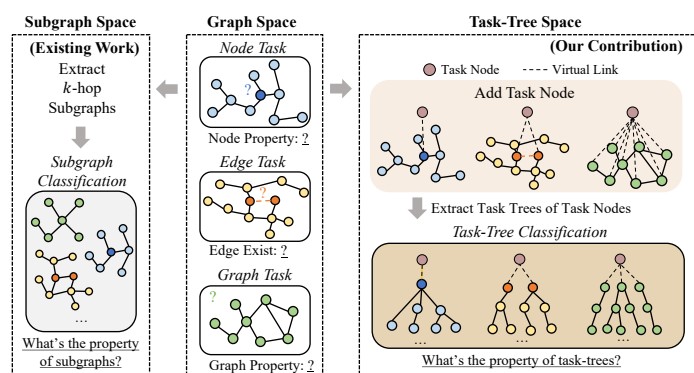

Figure 1: The formulation of task-trees.

relevant. Regardless of the task, basic GNN models only aggregate the embeddings of task-relevant nodes for predictions (Srinivasan & Ribeiro, 2020). This aggregation process is analogous to introducing a virtual "task node" that connects all task-relevant nodes, and learn the embedding of the task nodes for prediction. We refer the computation tree surrounding the "task node" as ***task-tree***, as illustrated in Figure 1 (Right). The task-tree formulation offers three distinct advantages. (1) Learnability: the information within task-trees can be fully captured by message-passing GNNs (Gupta et al., 2024), allowing the model to encode diverse structural information across domains. (2) Uniformity: task-trees are applicable to node-, edge-, and graph-level tasks, providing a unified task alignment paradigm across graphs (Sun et al., 2023). (3) Efficiency: encoding task-trees involves learning the embeddings of virtual nodes appended to the original graph, without extra time-consuming operations. Building on these benefits, it is feasible for task-trees to serve as the basic learning instances in graphs, similar to images in vision and sentences in language. Therefore, we propose that task-trees may maintain transferable patterns or concepts shared across graphs from different domains and tasks, leading to the following assumption:

> ***Task-Tree Generality Assumption: the generalities shared across graphs are (at least partially) preserved within the task-trees of the involved graphs.***

To validate this assumption, we conduct a theoretical analysis to examine the properties of task-trees in terms of stability, transferability, and generalization (Verma & Zhang, 2019; Garg et al., 2020), showing the feasibility of building GFMs based on task-trees. Our key finding is that if a GNN model is pretrained on diverse task-trees using a reconstruction task, it will acquire sufficient transferable knowledge to downstream tasks with a reasonable number of fine-tuning samples. Additionally, the analysis highlights the potential for specializing the pretrained model into specific domains through post-training (Wei et al., 2021) using domain-specific instances.

To validate our theoretical findings, we introduce a cross-task, cross-domain graph foundation model called **G**raph generality **I**dentifier on task-**T**rees (GIT). The core idea is to pretrain GIT on task-trees extracted from a wide range of graphs spanning diverse domains and tasks. We evaluate GIT across 32 graphs from 5 distinct domains, using settings such as basic fine-tuning, in-context learning (few-shot learning without fine-tuning), and zero-shot learning. The experimental results demonstrate that pretraining the model on a small set of graphs significantly benefits a wide range of graph tasks, supporting the idea that task-trees indeed capture generalities shared across graphs. We also propose an instruction tuning method to adapt the pretrained general model to specific downstream domains, which enhances performance across all datasets within the domain, matching or even surpassing domain expert models.

## 2 PRELIMINARY

We begin with a brief introduction to message-passing GNNs and some related concepts, where a comprehensive discussion on related works is presented in Appendix A. Let $\mathcal{G} = (\mathcal{V}, \mathcal{E})$ represent a graph with node set $\mathcal{V}$ and edge set $\mathcal{E}$, where each node $v \in \mathcal{V}$ is associated with a feature vector $\mathbf{x} \in \mathbb{R}^d$. A GNN encoder $\phi$ takes the graph as input and performs message passing to learn node embeddings $\boldsymbol{Z} = \phi(\mathcal{V}, \mathcal{E})$. Specifically, a GNN encoder can be defined as:

$$\boldsymbol{z}_i^{(l)} = \sigma\Big(\boldsymbol{W}_1 \boldsymbol{z}_i^{(l-1)} + \boldsymbol{W}_2 \rho\Big(\sum_{j \in \mathcal{N}(i)} g(\boldsymbol{z}_j^{(l-1)})\Big)\Big), \tag{1}$$

where $\mathcal{N}_i$ denotes the 1-hop neighbors of node $i$, $\boldsymbol{z}^{(l)}$ represents the node embedding at the $l$-th GNN layer with $\boldsymbol{z}^{(0)} = \boldsymbol{x}$, and $\boldsymbol{W}_1, \boldsymbol{W}_2$ are learnable matrices. The functions $\sigma$, $\rho$, and $g$ are the activation function, aggregation function and update function, respectively. To simplify the analysis, we assume $\rho$ is an averaging operation and $g$ is the identity function. Without loss of generality (WLOG), these functions can be replaced with any permutation-invariant and Lipschitz-continuous functions, respectively, without affecting the subsequent analysis. We now present some preliminary definitions before outlining the theoretical results.

**Definition 2.1** (Task-Relevant Nodes). Graph tasks can be roughly categorized into node-level, edge-level, and graph-level tasks, where the basic learning instances are nodes, edges, and entire graphs, respectively. For node classification, the task-relevant node $v_i^t$ is the target node $v_i$ that needs to be classified. In edge classification, the task-relevant nodes are the start and end nodes $\{v_i^t, v_j^t\}$ of the target edge $e_{ij}$. For graph classification, the task-relevant nodes $\{v_i^t\}_{i=1}^{|\mathcal{V}|}$ include all nodes in the target graph $\mathcal{G}$.

*Remark* 2.2. For any graph task instance, the prediction relies solely on the embeddings of the corresponding task-relevant nodes. These node embeddings capture the surrounding subtree structures, also known as computation trees.

**Definition 2.3** (Computation Trees (Chuang & Jegelka, 2022)). Given a node $v$ in graph $\mathcal{G}$, the $L$-layer computation tree $T_v^L$ is constructed by recursively expanding the subtrees of its neighboring nodes, starting with $T_v^1 = v$.

The learning process of GNNs involves recursively integrating information from the computation trees, progressing from the bottom to the top. Therefore, for a given node $v$, the node embedding produced by an $L$-layer GNN corresponds to the embedding of its computation tree $T_v^L$. Since the prediction for any graph task depends solely on the embeddings of task-relevant nodes, and these embeddings are determined by their respective computation trees, we can construct a task-tree for each instance—whether it be a node, edge, or graph—as illustrated in Figure 1 (Right).

**Definition 2.4** (Task-Trees). For any graph-related instance—whether a node, edge, or graph—we have a set of task-relevant nodes $\{v_1^t, ..., v_n^t\}$ and their corresponding $L$-layer computation trees $\{T_1, ..., T_n\}$. These computation trees can be reformulated into a larger task-tree $T^t$ by introducing an additional task node that connects all task-relevant nodes.

**Task-Tree Encoding.** We use a straightforward method to encode task-trees. Given a task-tree $T^t$, consisting of a virtual task node $v^t$ and a set of task-relevant nodes associated with computation trees $T_1, ..., T_n$, we first apply a GNN $\phi$ to encode each computation tree, obtaining node embeddings $\boldsymbol{z}_1, ..., \boldsymbol{z}_n$. We then aggregate these node embeddings into the task node, with the task node embedding serving as the embedding of the task-tree. Specifically, we use basic averaging for aggregation: $\boldsymbol{z}^t = \phi(T^t) = \frac{1}{n}\sum_{i=1}^n \phi(T_i)$.

## 3 THEORETICAL ANALYSIS

In this section, we theoretically analyze the properties of task-trees to demonstrate their stability, transferability, and generalization in acting as the basic learning instances, thereby validating the *Task-Tree Generality Assumption* from a theoretical perspective. It is important to note that the theoretical analysis is not to prove the superiority of task-trees over other methods, such as graphon or subgraph. Rather, we aim to show the feasibility of building GFMs based on task-trees.

We begin by examining the stability of GNNs in learning task-tree representations, showing that task-trees with similar subtree structures produce analogous embeddings. To facilitate this analysis, we first define the notation for describing subtree information:

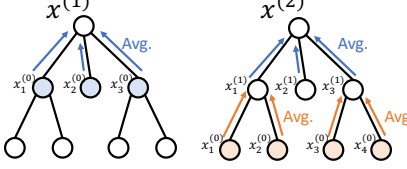

Figure 2: subtree information examples.

$$\boldsymbol{x}_i^{(l)} = \frac{1}{|\mathcal{N}_i|} \sum_{j \in \mathcal{N}_i} \boldsymbol{x}_j^{(l-1)}, \qquad (2)$$

where $\boldsymbol{x}_i^{(0)} = \boldsymbol{x}_i$ denotes the original node feature, and $x^{(l)}$ denotes the subtree information of nodes in $l$-th layer, as illustrated in Figure 2. In this figure, for $l = 1$, only the nodes in the first layer of the tree are considered, and for $l = 2$, only the nodes in the second layer are considered.

**Theorem 3.1** (Stability on Task-Trees). *Given two $L$-layer task-trees $T_t^1$ and $T_t^2$, with task-relevant nodes $\{v_1, ..., v_n\}$ and $\{v_1, ..., v_m\}$, respectively. The distance between task-trees is defined as $\Delta := \|\phi(T_1^t) - \phi(T_2^t)\|$ with the following bound:*

$$\Delta = \|\phi(T_1^t) - \phi(T_2^t)\| = \|\frac{1}{n}\sum_{i=1}^n \phi(T_i) - \frac{1}{m}\sum_{j=1}^m \phi(T_j)\| \leq \frac{1}{nm}\sum_{i=1}^n \sum_{j=1}^m \Big( \mathcal{C}_1 \|\boldsymbol{x}_i^{(0)} - \boldsymbol{x}_j^{(0)}\|$$

$$+ ... + \mathcal{C}_1 \mathcal{C}_2^{L-1} \|\boldsymbol{x}_i^{(L-1)} - \boldsymbol{x}_j^{(L-1)}\| \Big) \leq 2\mathcal{B}_{\boldsymbol{x}} \cdot \mathcal{C}_1 \frac{\mathcal{C}_2^L - 1}{\mathcal{C}_2 - 1}, \qquad (3)$$

*where $\phi$ is the GNN encoder, $T_i$ is the computation tree corresponding to node $i$, and $\mathcal{C}_1, \mathcal{C}_2$ are constants related to the encoder, and $\mathcal{B}_{\boldsymbol{x}}$ represents the bounded norm of $\boldsymbol{x}$.*

The proof can be found in Appendix D.1. Theorem 3.1 suggests that two task-trees are likely to have similar representations if their subtrees are similar. This theorem highlights the significance of similarity between pairs of subtrees, while downplaying the impact of the number of subtrees (i.e., the width of the task-trees), despite having more subtrees could potentially increase diversity and thus magnify discrepancy. The theorem also implies that increasing the number of GNN layers may lead to a loose bound, which aligns with previous analyses (Garg et al., 2020; Ju et al., 2023).

**Illustration 3.2.** *This theorem provides theoretical support for using task-trees as basic learning instances in graph tasks. Consider two task-trees: one representing a node (with a single subtree) and the other representing a graph (with multiple subtrees). While the widths of these task-trees differ significantly, if their subtrees share some degree of similarity, they can produce similar representations. Thus, this theorem ensures that task-trees of nodes, edges, or graphs can potentially be similar, making it possible to use a GNN encoder to capture the shared patterns among them.*

Following the stability analysis, we now examine the transferability of task-trees in pretraining and fine-tuning scenario. Specifically, assuming a model is pretrained on a task-tree reconstruction task, we aim to quantify how the knowledge acquired during pretraining can be transferred to downstream tasks. The pretraining objective is defined as $\mathcal{L}_{\mathcal{P}}(g \circ \phi) := \mathbb{E}_{(\hat{T}, T) \sim \mathcal{P}} \|g(\phi(\hat{T})) - \phi(T)\|^2$, where $\mathcal{P}$ represents the task-tree distribution used for pretraining, $\phi \in \Phi$ and $g \in G$ are the GNN encoder and the reconstruction head, respectively. $T$ denotes the task-tree and $\hat{T}$ is the corrupted version of $T$, generated using arbitrary augmentations. Note that the reconstruction head $g$ is used only during pretraining and is discarded during fine-tuning. Then, we define the risk on downstream task as $\mathcal{R}_{\mathcal{T}}(f \circ \phi) := \mathbb{E}_{(T, y) \sim \mathcal{T}} \kappa(f(\phi(T)), y)$, where $f \in \mathcal{F}$ is a linear head for predictions, $\mathcal{T}$ represents the downstream task distribution with task-tree $T$ and label $y$, and $\kappa$ denotes the loss function.

**Theorem 3.3** (Transferability on Task-Trees). *Given two task-tree encoders $\phi, \phi' \in \Phi$, we have*

$$\min_{f \in \mathcal{F}} \mathcal{R}_{\mathcal{T}}(f \circ \phi) - \min_{f' \in \mathcal{F}} \mathcal{R}_{\mathcal{T}}(f' \circ \phi') \leq \mathcal{C}_{\delta} \Big( \min_{g \in G} \mathcal{L}_{\mathcal{P}}(g \circ \phi) - \min_{g' \in G} \mathcal{L}_{\mathcal{P}}(g' \circ \phi') \Big)^{\delta}, \qquad (4)$$

*where $\mathcal{C}_{\delta} \approx O(1)$ and $\delta = \frac{1}{2}$.*

The proof is provided in Appendix D.2. In summary, Theorem 3.3 demonstrates that knowledge gained through pretraining on task-tree reconstruction tasks is transferable to downstream tasks, and it quantifies the extent of this transfer. The left-hand side (LHS) of the theorem shows how different representations impact performance on downstream tasks, while the right-hand side (RHS) reflects the difference in pretraining losses between two encoders. Therefore, if two encoders exhibit similar losses during pretraining, their transferability to a new task should be comparable.

**Illustration 3.4.** *To give a better understanding on why Theorem 3.3 imply the model pretrained on task-trees can bring transferable information to downstream tasks, we present an example. Let's consider the case where $\phi$ is the pretrained encoder and $\phi'$ is a randomly initialized encoder. The LHS term $\min_{f \in \mathcal{F}} \mathcal{R}_{\mathcal{T}}(f \circ \phi) - \min_{f' \in \mathcal{F}} \mathcal{R}_{\mathcal{T}}(f' \circ \phi')$ measures the amount of knowledge that is acquired during pretraining and is capable to be transferred to downstream tasks, and the RHS term $\min_{g \in G} \mathcal{L}_{\mathcal{P}}(g \circ \phi) - \min_{g' \in G} \mathcal{L}_{\mathcal{P}}(g' \circ \phi')$ measures the total knowledge acquired during pretraining. Thus, the constants $\mathcal{C}_{\delta}$ and $\delta$ quantify how much of this knowledge is transferable to downstream tasks. Since both $\mathcal{C}_{\delta}$ and $\delta$ are reasonably small, we conclude that pretraining on task-trees provides sufficient knowledge to benefit downstream tasks.*

To further explain why the task-tree-based pretraining and fine-tuning framework is effective for downstream tasks, we derive the following generalization bound.

**Theorem 3.5** (Generalization on Task-Trees). *Given two task-tree distributions, $\mathcal{P}$ for pretraining and $\mathcal{T}$ for fine-tuning, suppose the encoder $\phi$ is pretrained on a set of task-trees $\{T_i\}_{i=1}^{m}$ sampled from $\mathcal{P}$ and finetuned on task-trees $\{T_i\}_{i=1}^{n}$ sampled from $\mathcal{T}$, the generalization bound of the fine-tuned model, with probability at least $1 - v$, is*

$$
\mathcal{R}_{\mathcal{T}}(f \circ \phi) \leq \min_{f' \in \mathcal{F}} \mathcal{R}_{\mathcal{T}}(f' \circ \phi^*) + \mathcal{C}_{\delta}\left(\mathcal{E}_{\mathcal{P}}(g, \phi)\right)^{\delta} + \frac{4\mathcal{C}_1}{n}\sqrt{\sum_{i=1}^{n}\left\|\phi(T_i)\right\|^2}
$$

$$
+ 2\mathcal{C}_2\Big(\sum_{x \in \mathcal{X}_{\phi}}\left\|\mathcal{T}_{\phi}(x) - \mathcal{P}_{\phi}(x)\right\| + 2\sqrt{\frac{\log(1/v)}{n}}\Big), \qquad (5)
$$

*where $\phi^* = \arg\min_{\phi \in \Phi} \min_{g \in G} \mathcal{L}_{\mathcal{P}}(g \circ \phi)$ is the optimal task-tree encoder obtained on $\mathcal{P}$, $\mathcal{E}_{\mathcal{P}}(g, \phi) = \mathcal{L}_{\mathcal{P}}(g \circ h) - \min_{g' \in G, \phi' \in \Phi} \mathcal{L}_{\mathcal{P}}(g' \circ \phi')$ defines the excess risk during pretraining. Constants $\mathcal{C}_1$ and $\mathcal{C}_2$ are related to downstream tasks, while $\mathcal{C}_{\delta} \approx O(1)$ and $\delta = \frac{1}{2}$ are the same as Theorem 3.3. $\mathcal{X}_{\phi}$ denotes the distribution of task-tree embeddings encoded via $\phi$, and $\|\mathcal{T}_{\phi}(x) - \mathcal{P}_{\phi}(x)\|$ measures the distance between task-tree distributions of pretraining and fine-tuning data.*

The proof can be found in Appendix D.3. This theorem outlines key factors affecting model generalization on downstream tasks, such as the transferability of task-trees ($\mathcal{C}_{\delta}(\mathcal{E}_{\mathcal{P}}(g, \phi))^{\delta}$) and the quality of the pretrained encoder ($\mathcal{E}_{\mathcal{P}}(g, \phi)$). With regard to the number of task-trees, we find that while increasing the number of fine-tuning samples contributes to more stable optimization ($\frac{4\mathcal{C}_1}{n}\sqrt{\sum_{i=1}^{n}\|\phi(T_i)\|^2}$), it does not significantly reduce the generalization bound ($2\sqrt{\frac{\log(1/v)}{n}}$). This provides theoretical evidence that a reasonable number of fine-tuning samples can be sufficient for training a model with strong generalization capabilities. Moreover, the discrepancy between the pretraining and fine-tuning distributions ($\sum_{x \in \mathcal{X}_{\phi}} \|\mathcal{T}_{\phi}(x) - \mathcal{P}_{\phi}(x)\|$) is crucial—smaller distribution gaps lead to better generalization. This highlights the importance of increasing the diversity of pretraining data, which provides a boarder pretraining distribution $\mathcal{P}$. It also supports the potential of developing specialized models for specific domains based on a pretrained general model, which is discussed in Section 4.2.

# 4 GIT: GRAPH GENERALITY IDENTIFIER ON TASK-TREES

The theoretical analysis demonstrates the feasibility of building graph foundation models based on task-trees. In this section, we apply these theorems to develop a cross-task, cross-domain GFM called **GIT**, with the aim of empirically validating the *Task-Tree Generality Assumption*.

## 4.1 General Model: Pretraining to Acquire General Knowledge

We propose a task-tree reconstruction task as a pretext for pretraining. The key idea is to use two corrupted task-trees to reconstruct each other, thereby capturing corruption-invariant semantics of the task-trees. Given a set of task-trees $\{T_1^t, ..., T_n^t\}$ sampled from a graph database[1], we apply corruption techniques to generate two views of each task-tree, denoted as $\{\hat{T}_1^t, ..., \hat{T}_n^t\}$ and $\{\tilde{T}_1^t, ..., \tilde{T}_n^t\}$. For corruption, we use random edge masking and random attribute masking, as proposed by Zhu et al. (2020), due to its computational efficiency. We then use an encoder $\phi$ to obtain embeddings for the corrupted task-trees, resulting in $\{\hat{z}_1, ..., \hat{z}_n\}$ and $\{\tilde{z}_1, ..., \tilde{z}_n\}$. Note that the task-tree embedding is defined as the average of the embeddings of task-relevant nodes, avoiding the introduction of inductive biases that could lead the model to learn incorrect patterns. The loss function is

$$\mathcal{L} = \underbrace{\frac{1}{2n}\sum_{i=1}^{n}\Big[\big\|\rho(g(\hat{z}_i)) - \text{sg}[\rho(\tilde{z}_i)]\big\|^2 + \big\|\rho(g(\tilde{z}_i)) - \text{sg}[\rho(\hat{z}_i)]\big\|^2\Big]}_{\text{Reconstruction}} + \underbrace{\sum_{i=1}^{n} D_{\text{KL}}(\boldsymbol{h}\|\boldsymbol{z}_i)}_{\text{Reg.}}, \quad (6)$$

where $g$ is a non-linear MLP projector, $\rho(\boldsymbol{z}) = (\boldsymbol{z}/\|\boldsymbol{z}\|)$ serves for normalization, sg is the stop-gradient operation, and $\boldsymbol{h}$ is the average of all instances $\boldsymbol{z}$. The reconstruction loss captures the semantics of the task-trees in a predictive manner, while the regularizer ensures the embeddings are projected into a shared space by minimizing the KL divergence between individual instances and their center. Additional analysis is provided in Appendix C.

## 4.2 Specialized Model: Specification via Instruction Tuning

Theorem 3.5 highlights the relationship between model generalization and the distribution gap between pretraining data $\mathcal{P}$ and fine-tuning data $\mathcal{T}$, showing that a smaller gap leads to better generalization. Based on this finding, it is feasible to develop a specialized model for specific domains from a pretrained general model. This is based on the mild assumption that *graphs from the same domain have similar task-tree distributions* $\{\mathcal{T}_1, .., \mathcal{T}_n\}$. If the pretrained model is post-trained on a task-tree distribution $\mathcal{P}_{post}$ sampled from $\{\mathcal{T}_1, .., \mathcal{T}_n\}$, the pretraining data distribution $\mathcal{P}$ can be adjusted towards these task-tree distributions. This reduces the discrepancy $\sum_{x \in \mathcal{X}_\phi} \|\mathcal{T}_\phi(x) - \mathcal{P}_\phi(x)\|$ in Theorem 3.5, thereby improving model generalization on the target domain. To achieve this, we propose an instruction-tuning method for post-training the pretrained model.

Instruction tuning is a supervised fine-tuning (SFT) technique designed to enhance the capabilities of a pretrained model by post-training it on a small dataset. Our goal is to fine-tune the model using instructions to specialize it for a particular domain of interest. Given a pretrained model $\phi^*$ and a set of task-trees $\{T_1, ..., T_n\}$ from the target domain, we post-train the model using the SFT loss:

$$\mathcal{L}_{SFT} = \frac{1}{n}\sum_{i=1}^{n} \kappa(\phi^*(T_i), \psi(T_i)), \quad (7)$$

where $\psi$ is the instruction generation function for each task-tree, and $\kappa$ is the corresponding loss function. In this paper, as we use text-attributed graphs in our experiments, we define instructions as the embeddings of label descriptions encoded by a LLM, which is similar to Liu et al. (2024), and we use mean squared error as the loss function $\kappa$.

## 5 Experiment

### 5.1 Experimental Setup

The detailed experimental settings are provided in Appendix E, and the comprehensive experimental results and analysis are presented in Appendix F. In the following, we briefly introduce the datasets, baselines, and basic evaluation protocols.

---

[1]We assume the node attributes of all graphs have been aligned using methods such as LLMs (Liu et al., 2024) or SVD (Zhao et al., 2024a). While proper node attribute alignment is important for GFMs, it is beyond the scope of this paper. We use text-attributed graphs in our experiments due to the data availability (Chen et al., 2024b; Zhang et al., 2024b; Feng et al., 2024), and use Sentence Bert (Reimers & Gurevych, 2019) to convert node features of all graphs into 768-dim vectors, following Liu et al. (2024).

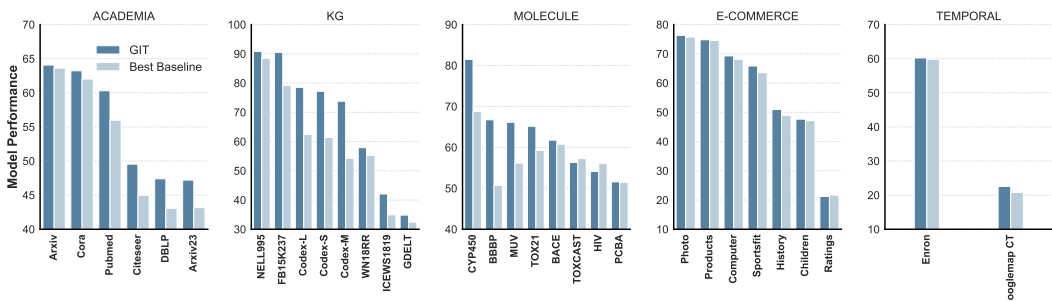

Figure 3: The model performance on all datasets in the in-context setting.

Table 1: We report the model performance across five graph domains: academia, e-commerce, knowledge base, molecular, and temporal graphs, with results averaged over all graphs within each domain. Note that -G and -S represent the general and specialized versions of GIT, respectively. The comprehensive results with additional baselines and ablation studies can be found in Appendix F.

|  | Domain | Academic | E-commerce | KG | Molecule | Temporal | Held-out Avg. | Avg. |
|---|---|---|---|---|---|---|---|---|
| **0-shot** | Sup. GNN | - | - | - | - | - | - | - |
|  | GraphMAE | 15.42 | 8.19 | - | 47.19 | - | 26.67 | 25.11 |
|  | OFA | 13.98 | 8.73 | - | 50.49 | - | 27.20 | 26.14 |
|  | GIT - G | 14.88 | 8.79 | - | 53.34 | - | 28.56 | 27.50 |
|  | GIT - S | **23.45** | **17.06** | - | **62.83** | - | **35.19** | **36.32** |
| **3-shot** | Sup. GNN | - | - | - | - | - | - | - |
|  | GraphMAE | 49.25 | 48.20 | 56.56 | 56.01 | 40.31 | 50.15 | 52.07 |
|  | OFA | 45.93 | 57.06 | 56.97 | 57.03 | 38.92 | 51.84 | 53.70 |
|  | GIT - G | 54.00 | 57.22 | 67.55 | 55.96 | 39.95 | 56.09 | 57.82 |
|  | GIT - S | **55.18** | **58.01** | **67.80** | **62.82** | **41.38** | **58.69** | **60.15** |
| **Finetune** | Sup. GNN | 73.57 | 78.21 | 66.86 | 73.65 | 62.61 | 71.14 | 72.25 |
|  | GraphMAE | 73.81 | 76.57 | 72.61 | 71.41 | 62.75 | 71.37 | 72.79 |
|  | OFA | 72.18 | 76.64 | 72.38 | 74.03 | 62.31 | 71.48 | 73.08 |
|  | GIT - G | 75.82 | 78.55 | 75.73 | 74.57 | 64.59 | 73.84 | 75.37 |
|  | GIT - S | **75.88** | **78.83** | **76.15** | **75.20** | **64.68** | **74.19** | **75.72** |

**Datasets.** Our experiments are based on text-attributed graphs due to data availability. Specifically, we include over 30 graphs spanning five domains: academic networks, e-commerce networks, knowledge graphs, molecular graphs, and temporal graphs. Detailed information is provided in Appendix E.1. For model pretraining, we use the citation network `Arxiv`, the e-commerce network `Products`, knowledge graphs `WN18RR` and `FB15K237`, and molecular graphs `Chemblpre` and `PCBA`. For supervised fine-tuning (SFT) during specialization, we use `Arxiv` for academic networks, `Products` for e-commerce networks, `FB15K237` for knowledge graphs, and `PCBA` for molecular networks. For temporal graphs, which are e-commerce temporal graphs, we also use `Products` for SFT to assess robustness under temporal distribution shifts.

**Baselines.** We employ a wide range of baselines, including supervised GNNs (Kipf & Welling, 2017; Veličković et al., 2018; Xu et al., 2019), self-supervised GNNs such as BGRL (Thakoor et al., 2022) and GraphMAE (Hou et al., 2022), graph foundation models like OFA (Liu et al., 2024), and domain-specific expert models. These include ULTRA (Galkin et al., 2024) for knowledge graphs and KVPLM (Zeng et al., 2022), MoMu (Su et al., 2022), Galactica (Taylor et al., 2022), and GIMLET (Zhao et al., 2023) for molecular graphs. Further details are provided in Appendix E.2.

**Experimental Protocols.** We use GraphSAGE (Hamilton et al., 2017) as the encoder, and each experiment is repeated at least five times with different seeds to reduce the impact of randomness. We evaluate three settings: fine-tuning, in-context learning, and zero-shot learning. Fine-tuning is the standard approach where the model is fine-tuned on downstream tasks. In-context learning is a few-shot learning method without fine-tuning, where we randomly select $k$ samples from a given class, average these samples to form prototypes, and use the prototypes for classification. Following Liu et al. (2024); He & Hooi (2024), we randomly sample 500 5-way 3-shot tasks. If the number of classes is fewer than 5, the number of ways is set to the number of classes. Zero-shot learning

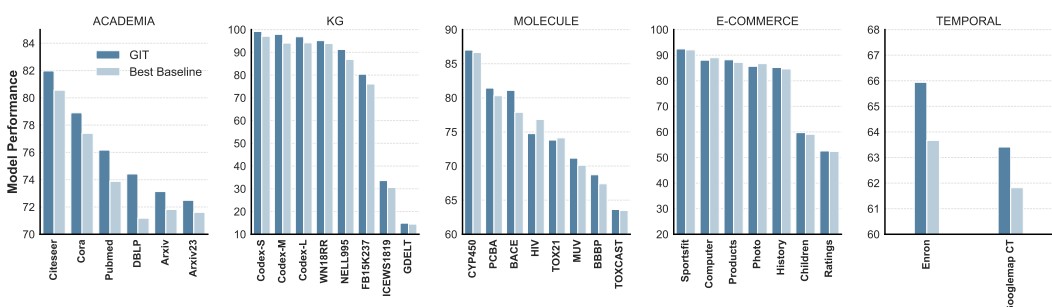

Figure 4: The model performance on all datasets in the fine-tuning setting.

Table 2: In-context learning results on knowledge graphs, where ULTRA is the domain expert. We **bold** the best results and underline the second-best. The results with *std.* are in Table 20 and 21.

|  |  | WN18RR | Codex-S | Codex-M | Codex-L | NELL995 | GDELT | ICEWS1819 | FB15K237 | **Avg.** |
|---|---|---|---|---|---|---|---|---|---|---|
| Domain Expert | ULTRA | 60.69 | 82.45 | 74.35 | 75.98 | 90.22 | 33.89 | 41.37 | 89.29 | 68.53 |
| General Model | GraphMAE | 55.20 | 61.41 | 54.30 | 61.01 | 86.42 | 32.43 | 31.58 | 70.15 | 56.56 |
|  | OFA | 55.27 | 55.14 | 50.20 | 62.40 | 88.41 | 30.23 | 34.94 | 79.15 | 56.97 |
|  | GIT - S | 57.90 | 77.19 | 72.14 | **76.99** | **90.80** | **34.85** | **42.02** | **90.49** | 67.80 |

Table 3: Zero-shot results on molecule graphs with domain experts KVPLM (Zeng et al., 2022), MoMu (Su et al., 2022), Galactica (Taylor et al., 2022), and GIMLET (Zhao et al., 2023).

|  |  | HIV | BBBP | BACE | TOXCAST | CYP450 | TOX21 | MUV | PCBA | **Avg.** |
|---|---|---|---|---|---|---|---|---|---|---|
| Domain Expert | KVPLM | 61.20 | 60.20 | 51.26 | 50.96 | 59.22 | 49.17 | 61.72 | 48.11 | 55.23 |
|  | MoMu | 50.26 | 49.81 | 66.56 | 52.38 | 57.98 | 57.57 | 60.51 | 51.50 | 55.82 |
|  | Galactica-1.3B | 33.85 | 53.94 | 56.48 | 51.23 | 46.86 | 49.46 | 57.15 | 52.02 | 50.12 |
|  | GIMLET | **66.24** | 59.39 | **69.57** | **59.04** | **71.25** | 61.19 | 64.39 | 62.11 | **64.15** |
| General Model | BGRL | 55.27 | 53.72 | 33.74 | 49.00 | 60.99 | 46.40 | 39.90 | 42.39 | 47.68 |
|  | GraphMAE | 46.48 | 49.08 | 30.76 | 48.22 | 60.55 | 49.17 | 48.17 | 45.10 | 47.19 |
|  | OFA | 47.96 | 50.61 | 34.35 | 49.70 | 61.96 | 52.73 | 52.48 | 54.14 | 50.49 |
|  | GIT - S | 66.14 | **62.16** | 52.27 | 58.30 | 69.75 | **63.45** | **65.32** | **65.26** | 62.83 |

operates similarly, but instead of using sample averages, it employs class description embeddings encoded by an LLM as the prototypes for prediction. Considering the metrics, we use accuracy in node classification and edge classification, and AUC in graph classification and link prediction.

## 5.2 MAIN RESULTS

**Results on all graphs.** We present the model performance across all graphs for both the basic fine-tuning and in-context learning settings in Figure 4 and Figure 3, respectively. Results for zero-shot learning are provided in Appendix F.1. In these figures, we compare the performance of GIT[2] and the best baselines. Note that we do not include domain expert models here since they are only applicable to specific domains. We observe that GIT consistently outperforms the best baselines across most datasets, highlighting the advantages of treating task-trees as the fundamental learning instance and providing empirical validation of the *Task-Tree Generality Assumption*.

**Results by domains.** For a clearer understanding, we present the experimental results averaged across each domain in Table 1, with additional results and extra baselines provided in Appendix F, categorized by domain. In this table, we compare model performance across three settings: zero-shot learning, 3-shot in-context learning, and basic fine-tuning, alongside Supervised GNN (Sup. GNN), self-supervised GraphMAE (Hou et al., 2022), and the graph foundation model OFA (Liu et al., 2024). The "held-out avg." indicates the average performance on all graphs, excluding those used in pretraining and specialization, to eliminate their influence on downstream tasks. We observe that even the general GIT significantly outperforms the baselines in some settings. After specialization, the specialized GIT further boosts performance, particularly in zero-shot and in-context learning settings, which supports our theoretical findings that post-training a pretrained model on domain-specific task-trees can make the model more suitable for specific domains.

---

[2]We report the best performance across general GIT and specialized GIT.

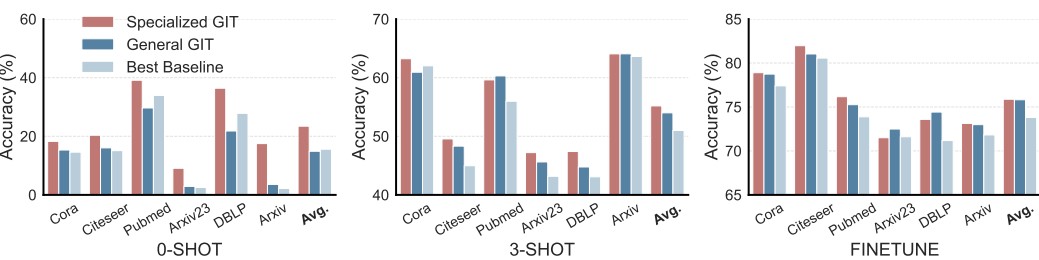

Figure 5: Comparison between GIT and best baselines on academia networks.

Table 4: The link prediction results on academia networks.

|  | Cora | Citeseer | Pubmed | Arxiv23 | DBLP | Arxiv | **Avg.** |
|---|---|---|---|---|---|---|---|
| Best Baseline | 87.34 ± 0.88 | 87.52 ± 0.98 | **89.83 ± 0.35** | 91.45 ± 0.44 | **98.29 ± 0.07** | **97.50 ± 0.08** | 90.96 |
| GIT - G | 87.79 ± 2.07 | 87.59 ± 0.96 | 84.35 ± 0.26 | 91.47 ± 0.46 | 98.25 ± 0.09 | 97.14 ± 0.06 | 91.10 |
| GIT - S | **88.58 ± 1.88** | **88.50 ± 1.15** | 87.78 ± 0.13 | **91.86 ± 0.38** | 98.27 ± 0.05 | 97.30 ± 0.05 | **92.05** |

**The analysis of specialization.** More encouragingly, applying the proposed specialization method to other graph learning models also improves their performance in specific domains (Table 17). This highlights the potential of post-training to enhance the capabilities of GNNs. To further investigate the specialization process, we conducted experiments to assess the impact of SFT data (Figure 12 and Table 24). The results show that SFT data can improve the performance of the general model across most settings. However, different SFT datasets have varying effects on different downstream tasks, underscoring the need for better SFT data selection methods. Additionally, we observe that the specialized model may suffer from reduced general inference capability, similar to the *specialization tax* seen in LLMs (Zhang et al., 2023). It is crucial to develop techniques that balance GIT's specialized and general capabilities.

## 5.3 CASE STUDIES BY DOMAINS

**Molecule Graphs.** We present the zero-shot graph classification results in Table 3. Following Zhao et al. (2023), we evaluate zero-shot performance on the original testing set. In addition to comparing with general algorithms applicable to all graphs, like GraphMAE, and OFA, we include domain-expert models designed specifically for molecules, such as KVPLM (Zeng et al., 2022), MoMu (Su et al., 2022), Galactica (Taylor et al., 2022), and the recent SOTA GIMLET (Zhao et al., 2023). Notably, our specialized GIT model achieves the second-best average performance following GIMLET, and even surpasses GIMLET on 4 out of 8 datasets, while significantly outperforming other general models and domain experts. This observation supports our theoretical analysis, demonstrating that post-training a pretrained model with domain-specific instances can effectively build a specialized model for that domain.

**Knowledge Bases.** We present the in-context edge classification results on knowledge graphs in Table 2, comparing our GIT model with general models and the domain expert ULTRA (Galkin et al., 2024). The specialization is done using FB15K237. We observe that our specialized GIT significantly outperforms the baselines, even approaching the performance of ULTRA, a foundation model specifically designed for knowledge graphs. Even more impressively, GIT surpasses ULTRA on 5 out of 8 datasets, highlighting the potential of using task-trees as basic learning instances and aligning with our theoretical analysis.

**Academic Networks.** We present the node classification and link prediction results on academic networks in Figure 5 and Table 4, respectively. We compare the performance of GIT specialized via Arxiv, general GIT, and the best baseline. Both versions of GIT consistently outperform the best baselines across datasets and settings, with the specialized version generally performing better than the general one. For link prediction, we randomly split the train/val/test sets in a 70%/15%/15% ratio. GIT also achieves the best performance in this setting. Moreover, specialization consistently improves model performance, even though the process is optimized at the node level rather than the edge level, demonstrating the potential for cross-task transferability through specialization.

Figure 6: Training efficiency between task-tree and subgraph versions of GIT.

Table 5: Performance comparison between methods with different learning instances.

| Instances | Subgraph | | Task-tree |
|---|---|---|---|
| Domains | OFA | GIT - SubG | GIT - Tree |
| Academia | 72.18 | 73.48 | **75.82** |
| KG | 72.38 | 73.59 | **75.73** |
| Molecule | 74.03 | 72.67 | **75.73** |
| Held-out Avg. | 71.31 | 70.88 | **73.81** |
| Avg. | 72.93 | 73.01 | **75.33** |

### 5.4 COMPARISON BETWEEN TASK-TREES AND SUBGRAPHS

Unlike our proposed task-trees, some approaches use $k$-hop subgraphs extracted from graphs as the basic learning units (Huang et al., 2023; Liu et al., 2024). In node classification, for example, ego-graphs are extracted around each node, and the label of the central node is assigned to the induced subgraph, transforming node classification into subgraph classification. A similar process can be applied to convert edge-level and graph-level tasks into subgraph-level tasks. This approach involves (1) extracting ego-graphs around task-relevant nodes and (2) applying GNNs to learn graph-level embeddings for classification. However, this extraction process introduces additional computational overhead, increasing both time consumption and memory usage due to the need to store and process the induced subgraphs. Furthermore, the information within these subgraphs is not always easily learned by basic GNNs, as they may struggle to capture essential substructures needed for learning graph-level embeddings (Garg et al., 2020; Chen et al., 2020; Zhang et al., 2024a), thereby limiting the effectiveness of using subgraphs as basic learning instances.

To illustrate this, we compare the efficiency and performance of subgraphs and task-trees in the basic fine-tuning setting, as shown in Figure 6 and Table 5, respectively. We implement a subgraph version of GIT by replacing task-trees with subgraphs (GIT-Subgraph). The results show that task-trees offer both better efficiency and improved model performance. Although Table 5 presents results from only three domains, the evaluation covers a range of tasks, including node classification (academic networks), edge classification (knowledge graphs), and graph classification (molecule networks).

## 6 RELATED WORK

Foundation models, like LLMs in NLP, excel at diverse tasks such as summarization, translation, and question-answering. However, building foundation models for graphs is challenging due to the inherent heterogeneity in features, structures, and tasks. Existing works, like Qiu et al. (2020), Sun et al. (2023), Huang et al. (2023), and Liu et al. (2024), tackle these issues by leveraging subgraphs to align task spaces and LLMs to align feature spaces. More recent approaches, such as He & Hooi (2024) and Li et al. (2024), jointly train GNNs and LLMs to enhance embeddings for few-shot and zero-shot tasks. However, reliance on subgraphs often leads to inefficiencies. Other methods like SVD (Zhao et al., 2024a; Yu et al., 2024) or non-parametric encoders (Zhao et al., 2024b) focus on resolving feature heterogeneity. In contrast, models designed for specific domains, such as Galkin et al. (2024) for knowledge graphs or Zhao et al. (2023) for molecular graphs, avoid these complexities but lack generalizability to other domains. Unlike these, our GIT model pretrains on diverse task trees, enabling it to generalize across domains and quickly specialize via instruction tuning. More related works are presented in the Appendix A.

## 7 CONCLUSION

We introduce the concept of task-trees as basic learning instances for graphs and provide both theoretical and empirical validation of the task-tree generality assumption: the generalities shared across graphs are (at least partially) preserved in the task-trees of the involved graphs. Building on task-trees, we develop GIT, a cross-domain and cross-task GFM that can be quickly specialized for specific domains. By pretraining on a small set of graphs, GIT improves performance across more than 30 graphs spanning five domains. After specialization with a small set of domain-specific task-trees, GIT can match or even surpass expert models designed for specific domains.

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

APPENDIX: TABLE OF CONTENT

# A    RELATED WORK

**Graph Neural Networks.** GNNs are a class of learning models specifically designed to operate on graph-structured data and have demonstrated substantial success across a variety of domains. Their strength lies in their ability to perform relational learning, where information from neighboring nodes is aggregated and used to enhance node representations. For instance, GCN (Kipf & Welling, 2017) utilizes message-passing to aggregate information from neighboring nodes to central nodes. Building on this, models such as GraphSAGE (Hamilton et al., 2017) and GAT (Veličković et al., 2018) introduce innovative techniques like neighborhood sampling and attention mechanisms, respectively, further advancing performance on graph learning tasks. However, these methods are limited to solving a single task by training from the scratch.

**Transferability of GNNs.** Existing works that analyze the shared concepts (generalities) across different graphs primarily follow two approaches. The first is graphon theory, which provides bounds on the distance between graphs generated from the same graphon. This method has been used to study transferability in pretraining and fine-tuning settings (Cao et al., 2023), to develop more expressive fine-tuning techniques (Sun et al., 2024), and to design new model architectures (Ruiz et al., 2020). However, despite its theoretical advantages, graphon-based approaches face practical challenges, particularly the strong assumptions required and the difficulty of identifying graphons in large-scale graphs, which limits their applicability in building graph foundation models. The second approach involves leveraging substructures within graphs to identify transferable patterns (Mao et al., 2024). This method focuses on extracting subgraphs composed of meaningful substructures for prediction tasks. While this approach offers theoretical insights into stability (Levie et al., 2019; Zhu et al., 2021), it struggles to fully capture substructures that are beneficial for downstream tasks (Zhang et al., 2024a).

**Graph Foundation Models.** Foundation models are designed as general-purpose solvers capable of handling various tasks across different domains. For instance, LLMs, the foundation models in natural language processing, are capable of performing tasks such as summarization, translation, and entity recognition, as well as question-answering. However, building such versatile foundation models for graphs presents unique challenges due to the inherent feature, structural, and task heterogeneity across different graph domains and tasks. To address these challenges, Qiu et al. (2020) pretrained GNNs using subgraphs as basic units, mitigating structural heterogeneity. Building on this, Sun et al. (2023) reformulated node-, edge-, and graph-level tasks into subgraph-level tasks, tackling task heterogeneity. Additionally, Huang et al. (2023) and Liu et al. (2024) applied LLMs to unify the feature spaces of cross-domain graphs, addressing feature heterogeneity. These approaches enable models to operate on cross-domain and cross-task graphs. Further advancements, such as He & Hooi (2024) and Li et al. (2024), improve node embeddings by jointly optimizing GNN and LLM encoders, facilitating various downstream tasks like few-shot learning and zero-shot learning. However, most of these approaches rely on subgraphs as the primary learning instances, which can result in inefficient training and reduced expressiveness, as discussed in the main paper. Other efforts to resolve feature heterogeneity include methods like singular vector decomposition (SVD) (Zhao et al., 2024a; Yu et al., 2024) and non-parametric encoders (Zhao et al., 2024b).

Another line of research focuses on designing GFMs for single tasks or domains, thereby avoiding the complexities of feature, structural, or task heterogeneity. For example, Galkin et al. (2024) propose a foundation model for reasoning tasks on knowledge graphs, using triplets as basic transferable patterns. Zhao et al. (2023) introduce a foundation model for molecular graphs, employing LLMs to align semantics between datasets and encode key motifs. In node classification, Li et al. (2024) propose a zero-shot learning foundation model, while Zhao et al. (2024a) present a feature alignment method based on SVD for node-level graph foundation models. Recently, Zhao et al. (2024b) designed a foundation model for node classification using a non-parametric classifier. Meanwhile, Chen et al. (2024a), Tang et al. (2024), Guo et al. (2023), and Wang et al. (2024a) have explored using LLMs as graph reasoners to solve graph tasks, similar to their role in vision language models. While these methods excel at specific tasks or domains, they are not suitable as general graph solvers across diverse tasks. In contrast to these approaches, our proposed GIT model is pretrained on diverse task trees to acquire general reasoning capabilities, allowing it to quickly specialize in specific domains through instruction tuning.

# B POTENTIAL MODEL EXTENSIONS

## B.1 PRETRAINING

**How to Design Reconstruction Tasks?** Theorem 3.5 suggests that a well-designed encoder, capable of effectively handling reconstruction tasks during pretraining, can improve the model's generalization ability. One approach is to use more powerful encoders to enhance reconstruction performance. Another approach is to introduce additional reconstruction losses to further refine the encoder. For example, methods such as those proposed by Qiu et al. (2020) and Hou et al. (2022), or designing more comprehensive reconstruction objectives could be explored.

**How to Improve Transferability?** The pretraining task, i.e., task-tree reconstruction, differs from the downstream task of task-tree classification, as the task heterogeneity may hinder model transferability (Hu et al., 2020). To mitigate this, one could develop more effective adaptation methods, such as graph prompt learning (Sun et al., 2022), to reduce task heterogeneity.

## B.2 SPECIALIZATION VIA INSTRUCTION TUNING

**How to Define Instructions?** In this paper, as we focus on experiments with text-attributed graphs, we define instructions as label descriptions encoded by LLMs. However, this approach is not applicable to non-textual graphs. Other methods could be explored to define instructions, such as using proxy models (Hu et al., 2019) or graph heuristics (Jin et al., 2020) to generate instructions.

**How to Choose SFT Data?** We manually select graphs as supervised fine-tuning datasets for each domain, though the selected graphs may not be fully representative. Unlike textual data, evaluating the quality of graph datasets poses a challenge. Improved dataset selection methods could enhance the SFT process by identifying more representative or diverse data from graph databases. Additionally, while we perform instruction tuning over entire graphs, it is possible that only specific subgraphs are beneficial (Hashemi et al., 2024). Developing data selection methods that focus on high-quality subgraphs within a single SFT dataset could improve task-tree selection. Another worthy research line is to select SFT data that aligns with user preferences (Song et al., 2024).

**How to Leverage SFT Data?** In scenarios with limited instructions, standard supervised fine-tuning may struggle to capture sufficient knowledge of the target domain. To address this, methods could be proposed to better utilize the unlabeled instances in the SFT dataset, thus enhancing model adaptation (Sohn et al., 2020).

**How to Maintain General Inference Capability?** While instruction tuning specializes the model for a specific domain, it may compromise the model's general inference capabilities across other domains. This could hinder the model's performance when it needs to function both as a domain expert and a general reasoner. To mitigate this, regularization techniques could be designed to preserve the general knowledge encoded in the model during the instruction tuning process.

**Why SFT Works on Graphs?** Instruction tuning is a common post-training process in modern large language models (e.g., LLAMA, GPT) that significantly improves instruction-following capabilities. The success of this method in LLMs may stem from the fact that natural language serves as an interface between humans and models (Wei et al., 2021). However, the reason instruction tuning works for graphs remains an open question and presents a potential direction for future research.

## B.3 MORE SCENARIOS.

The paper leverages text-attributed graphs to align node features. However, the pre-processing of TAGs can be time-consuming, raising the challenge of how to effectively apply the model to graphs without aligned node features. Furthermore, while we primarily focus on homogeneous graphs in this work, most real-world applications involve heterogeneous graphs. Addressing the question of how to design a single model capable of handling various types of graphs remains an open challenge. Finally, applying the model to specific applications Zhang et al. (2024c), which may exhibit unique characteristics, is another important consideration for future research.

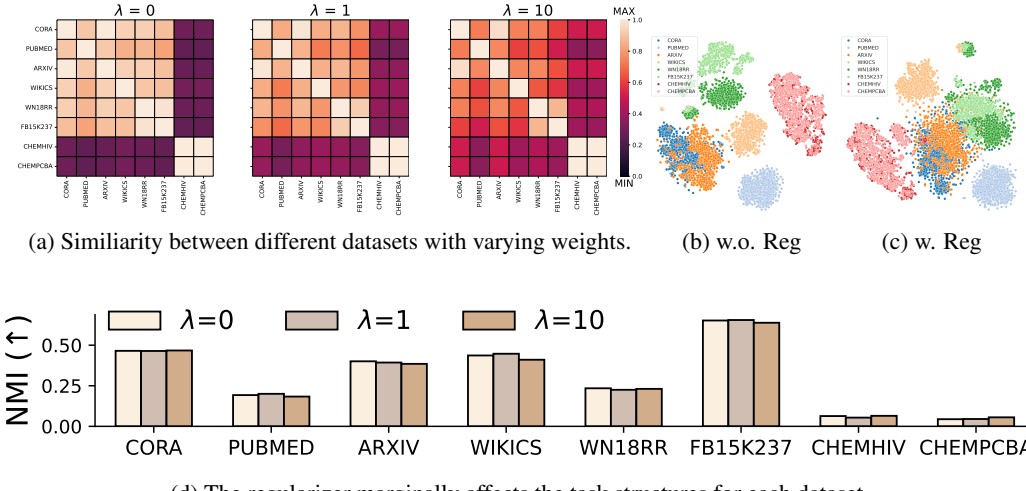

(a) Similiarity between different datasets with varying weights.  (b) w.o. Reg  (c) w. Reg

(d) The regularizer marginally affects the task structures for each dataset.

Figure 7: The domain regularizer controls the distance between datasets while preserving the structure within each of them.

Table 6: Model performance across settings with different scaling weights of domain regularizer.

|          | $\lambda = 0$ | $\lambda = 1$ | $\lambda = 10$ |
|----------|---------------|---------------|----------------|
| 0-shot   | 20.39         | 27.55         | **29.60**      |
| 3-shot   | 53.10         | 57.53         | **60.21**      |
| Finetune | 74.78         | **75.41**     | 75.37          |

## C  ADDITIONAL DISCUSSION

### C.1  WHY DOES THE GENERAL MODEL NEED SPECIALIZATION?

It is challenging for a single graph model to handle tasks across various domains due to pattern conflicts, where the same structural pattern can have different meanings in different domains. To illustrate this issue, we provide an intuitive example[3]. Consider a pretraining process involving datasets from multiple domains, such as social networks, molecular networks, academic networks, and knowledge graphs. Suppose the model learns triangle structures during pretraining. In social networks, the semantic meaning of these triangles is *stable*, following the principle of "the friend of my friend is my friend". However, in molecular graphs, the meaning of triangle patterns may be *unstable* due to chemical properties. This pattern conflict can significantly degrade the performance of graph models (Cao et al., 2023; Mao et al., 2024). Specialization helps resolve this issue by aligning the meanings of certain structural patterns with the semantics specific to the target domain.

### C.2  MORE ANALYSIS ON DOMAIN REGULARIZER

**The Necessity of Domain Alignment.** Datasets from multiple domains are often projected into different subspaces, potentially due to misalignment of node attributes (Chen et al., 2024b) and the frequent patterns across domains. As a result, the model may "memorize" information specific to each domain rather than learning transferable patterns. This can lead to misunderstandings when the same pattern appeared across different graphs is projected into different subspaces. Consequently, the model struggles to acquire transferable knowledge that would benefit unseen tasks and specialized domains. Properly aligning the embedding spaces of different domains is crucial for obtaining transferable knowledge and improving performance on unseen graphs and specialized domains.

**How to Regulate Domain Distances?** We propose a domain regularizer to control domain distances by projecting cross-domain graphs with different characteristics into a shared embedding space.

---

[3]This example was first illustrated in Cao et al. (2023)

Specifically, we define a shared subspace across domains and pull the subspaces of other domains into alignment with this defined space. The shared subspace should be positioned at the center of the cross-domain datasets to minimize the effort required to adjust the subspaces of all domains. In particular, the basis vector of the shared subspace is defined as the average of all instances:

$$\boldsymbol{h}_{Basis} = \mathbb{E}_{D \sim P(\mathcal{D})} \mathbb{E}_{\mathcal{T}_i \sim P(\mathcal{T}_D)} \phi(T_i), \tag{8}$$

where $P(\mathcal{D})$ represents the domain distribution, $\mathcal{T}_D$ is a distribution of task-trees within domain $D$, and $\phi(T_i)$ is the embedding of the task-tree $T_i$. Given the shared subspace basis, we optimize the KL divergence between each instance and the basis. However, obtaining the global basis vector $\boldsymbol{h}_{Basis}$ directly is impractical due to dataset size, so we approximate it by averaging the embeddings of all instances within a batch to compute the local basis $\hat{\boldsymbol{h}}_{Basis}$. We then optimize the KL divergence for all instances in the batch. To mitigate randomness, we empirically use a relatively large batch size (4,096). Formally, the domain regularizer is defined as

$$\mathcal{L}_{align} = \lambda \cdot \frac{1}{|\mathcal{B}|} \sum_{i \in \mathcal{B}} \mathrm{KL}(H \| Z_i) = -\lambda \cdot \frac{1}{|\mathcal{B}|} \sum_{i \in \mathcal{B}} \sum_j H(j) \log\Big(\frac{H(j)}{Z_i(j)}\Big), \tag{9}$$

where $\mathcal{B}$ denotes the batch, and $H$ and $Z_i$ represent the distributions of the local basis vector $\hat{\boldsymbol{h}}_{Basis}$ and instance embedding $\boldsymbol{z}_i$, respectively.

**How the Domain Regularizer Works?** To better understand how the domain regularizer functions, we conduct an analysis to demonstrate its benefits in regulating domain distances while preserving task structures for each dataset. We use eight datasets provided by Liu et al. (2024) for pretraining: `Cora`, `Pubmed`, `Arxiv`, `WikiCS`, `WN18RR`, `FB15K237`, `CHEMHIV`, and `CHEMPCBA`. The analysis results are presented in Figure 7.

In the figure, we display (a) a heatmap of similarity between different datasets with varying weights, and visualizations of the embedding space before (b) and after (c) applying the domain regularizer. The results show that the domain regularizer effectively adjusts the distances between datasets by pushing apart overly similar graphs and bringing closer those that are too distinct. Furthermore, we show that the regularizer does not significantly alter the task structures of each dataset, as illustrated in subfigure (d). In this subfigure, we apply $k$-means algorithm on each dataset, setting $k$ to the number of classes, and compare to the ground-truth by using NMI as the metric. The assumption is that if two sets of vectors yield similar clustering results, the classification outcomes of the same classifier will be similar, indicating that the task structure across the two sets is consistent. The results demonstrate that changing the regularizer weight does not significantly affect task structures. This may be because the regularizer acts by translating vectors toward a central point without altering the relationships between individual pairs of vectors. To further evaluate the impact of domain regularizer for the downstream tasks, we present the model performance average over the used eight datasets across all settings in Table 6. We observe the use of domain regularizer can boost the model performance, especially in in-context and zero-shot settings. In addition, we empirically find that $\lambda = 10$ can lead to better performance. Thus, we set $\lambda = 10$ as the default weight in this paper.

## C.3 DISCUSSION ON HOMOPHILY AND HETEROPHILY

In node-level tasks, it is important to consider both graph homophily and heterophily (Ma et al., 2022). Homophily describes the close relationships between connected entities, while heterophily refers to distant relationships between connected entities. Empirically, basic message-passing GNNs tend to perform well on homophily graphs but struggle with heterophily graphs (Luan et al., 2022). Despite using GraphSAGE as the backbone in our GIT, it still performs well on heterophily graphs, such as `Children` and `Ratings`[4]. The experimental results for node classification and link prediction on these two graphs are presented in Table 18 and Table 19, where GIT generally achieves the best performance. We hypothesize that the proposed task-tree structure captures not only homophily relationships but also heterophily relationships. A potential question is whether our message-passing GNN can effectively capture these heterophily relationships, despite Ma et al. (2022) suggesting that basic GNNs may handle heterophily graphs by memorizing the patterns between the target node and

---

[4]Our collected graphs include two heterophily graphs, `Children` and `Ratings`, with homophily ratios of 0.42 and 0.38, respectively, whereas other graphs generally have a homophily ratio greater than 0.60 (Table 12, (Chen et al., 2024b)).

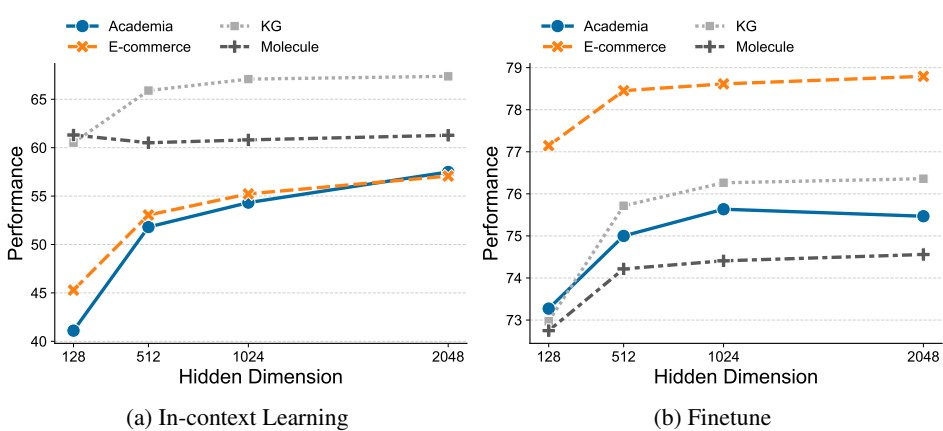

(a) In-context Learning  (b) Finetune

Figure 9: The impact of model sizes on performance.

its neighbors. We plan to use more advanced GNNs capable of encoding heterophily structures to further validate our hypothesis.

### C.4   DISCUSSION ON MODEL EXPRESSIVENESS

**Node-level Task.** The task-tree structure is an approximation of the original graph structure, but converting graphs into tree structures inevitably results in some loss of information. To better preserve the structural details of the original graph, one could use more expressive or advanced GNNs, thereby expanding the potential tree vocabulary (Mao et al., 2024).

**Edge-level Task.** Existing message-passing GNNs struggle with the edge isomorphism problem (Srinivasan & Ribeiro, 2020). For instance, in Figure 8, the links $(v_1, v_2)$ and $(v_3, v_4)$ are isomorphic, while $(v_1, v_2)$ and $(v_1, v_3)$ are not. However, when using a mean aggregator to learn edge embeddings, the embeddings of $(v_1, v_2)$ and $(v_1, v_3)$ become indistinguishable. We consider that GIT may still encounter this issue, as the task-tree encoding currently averages the embeddings of task-relevant nodes. Addressing this challenge could involve techniques like Zhang et al. (2021), which ensure that isomorphic edges have distinct embeddings without impairing the model's basic inductive learning capabilities.

Figure 8: Edge Isomorphic (Figure 1, (Zhang et al., 2021)).

**Graph-level Task.** Message-passing GNNs are limited by the 1-WL test (Xu et al., 2019), which can restrict their performance on graph-level tasks. As we apply GraphSAGE as the backbone, our GIT also encounters this limitation. Zhang et al. (2024a) analyze the ability of different GNNs to detect graph substructures and conclude that more expressive GNNs, beyond the 1-WL test, can learn graph embeddings with richer information. Therefore, to improve model expressiveness, one can employ more expressive GNNs in our GIT. Additionally, techniques like Zhang et al. (2021) can be used to further enhance the model's discriminative capabilities.

### C.5   SCALING LAW

**Model Size.** We evaluated the model's performance with different hidden dimensions, with results by domain presented in Figure 9. The results cover both basic fine-tuning and in-context learning, and comprehensive details are provided in Appendix F.8. We observe that increasing the number of hidden dimensions from 128 to 2,048 significantly improves model performance across all domains. We hypothesize that this improvement is due to the additional parameters, which enhance the model's ability to memorize shared patterns across graphs. The observation indicates the potential existence of scaling laws when using task-trees as the basic learning instances.

**Data Size.** We attempted to evaluate the scaling law by increasing the pretraining data, but unfortunately, we did not observe a clear trend where more data led to better performance. We consider there

are three potential reasons. (1) From a model perspective, we use a GraphSAGE encoder with limited layers and parameters, which may not fully capture the knowledge contained in the pretraining data. Additionally, we apply basic mean pooling to derive task-tree embeddings from task-relevant node embeddings, which may prevent the model from identifying the relative importance of task-relevant nodes, thereby limiting its expressiveness. (2) From a training paradigm perspective, we employ a negative-free contrastive learning framework similar to Thakoor et al. (2022), but this basic approach may not be expressive enough to extract meaningful knowledge from the pretraining graphs. (3) From a data perspective, despite using over 30 graphs in this study, the number of instances is still significantly lower than that of textual or visual instances extracted from the Internet. Furthermore, the pretraining datasets may not be well-aligned. Although we used a textual encoder to align node features, we cannot guarantee that the encoded node features are in the same embedding space (Chen et al., 2024b).

### C.6 LIMITATIONS AND FUTURE WORK.

We treat task-trees as basic learning instances in graphs, similar to how images are used in vision and text in language. Despite we have validated that cross-graph generalities are, at least partially, preserved in task-trees, identifying the exact transferable patterns shared across graphs remains a challenge. Further exploration of this issue from both empirical and theoretical perspectives is worthwhile. Additionally, for message-passing GNNs, converting graphs into tree structures inevitably leads to some loss of information, limiting the model expressiveness. Advanced GNNs (Morris et al., 2019; 2023) or techniques (Zhang et al., 2021) could be used to enhance expressiveness. Moreover, as GIT is a relatively simple model, investigating potential extensions is also a promising direction, as discussed in Appendix B.

### C.7 BROADER IMPACT.

From an industry perspective, we offer GIT as a foundational tool for graph-structured data. Additionally, since GIT can be quickly adapted to specific domains, we hope it will support applications where label acquisition is difficult and model training is time-consuming.

## D PROOF

### D.1 PROOF OF THEOREM 3.1

*Proof.* We begin by introducing the basic GNN architecture used in the proof. Given a GNN encoder $\phi(\cdot)$ with parameters $\boldsymbol{W} = (\boldsymbol{W}_1, \boldsymbol{W}_2)$, we use a GraphSAGE-like architecture, defined as follows (with some notation abuse):

$$\boldsymbol{z}_i = \phi(T_i^L) = \sigma\Big(\boldsymbol{W}_1 \boldsymbol{x}_i + \boldsymbol{W}_2 \frac{1}{|\mathcal{N}_i|} \sum_{k \in \mathcal{N}_i} \phi(T_k^{L-1})\Big),$$

where $\sigma$ is the non-linear activation function, $\boldsymbol{x}_i$ is the node feature of node $i$, and $\mathcal{N}_i$ represents the neighbors of node $i$, corresponding to its children in the computation tree. $T_i^L$ denotes the computation tree of node $i$ with $L$ layers. Neighborhood information is incorporated by averaging the embeddings of neighboring nodes. WLOG, the averaging operation can be replaced with any permutation-invariant set operation without affecting the analysis in this paper. For simplicity, we assume all GNN layers share the same parameters; this assumption does not affect the validity of our proofs. Since these functions and neural networks exhibit Lipschitz continuity, we denote the Lipschitz constant of $\sigma(\cdot)$ as $\mathcal{C}_\sigma$. Additionally, we assume the norm of node features is bounded by $\|\boldsymbol{x}\| \le \mathcal{B}_{\boldsymbol{x}}$, and the model weights by $\|\boldsymbol{W}_1\| \le \mathcal{B}_{\boldsymbol{W}_1}$ and $\|\boldsymbol{W}_2\| \le \mathcal{B}_{\boldsymbol{W}_2}$. While real-world graphs typically exhibit varied node features, standard techniques (as employed in this paper) like normalization can ensure that $\mathcal{B}_{\boldsymbol{x}}$ remains a small value. We define the distance between task-trees $T_1^t$ with $n$ task-relevant nodes $\{v_1, ..., v_n\}$ and $T_2^t$ with $m$ task-relevant nodes $\{v_1, ..., v_m\}$ as:

$$\Delta := \left\|\phi(T_1^t) - \phi(T_2^t)\right\| = \left\|\frac{1}{n} \sum_{i=1}^{n} \phi(T_i) - \frac{1}{m} \sum_{j=1}^{m} \phi(T_j)\right\|,$$

where $\|\cdot\|$ is the L2 distance. Following, we expand the stability term $\Delta$ as:

$$\Delta = \left\| \frac{1}{n}\sum_{i=1}^{n}\phi(T_i) - \frac{1}{m}\sum_{j=1}^{m}\phi(T_j) \right\|$$

$$= \left\| \frac{1}{n}\sum_{i=1}^{n}\sigma\Big(\boldsymbol{W}_1\boldsymbol{x}_i + \boldsymbol{W}_2\frac{1}{|\mathcal{N}_i|}\sum_{k\in\mathcal{N}_i}\phi(T_k^{L-1})\Big) - \frac{1}{m}\sum_{j=1}^{m}\sigma\Big(\boldsymbol{W}_1\boldsymbol{x}_j + \boldsymbol{W}_2\frac{1}{|\mathcal{N}_j|}\sum_{k\in\mathcal{N}_j}\phi(T_k^{L-1})\Big) \right\|$$

$$\leq \mathcal{C}_\sigma\left\| \frac{1}{n}\sum_{i=1}^{n}\Big(\boldsymbol{W}_1\boldsymbol{x}_i + \boldsymbol{W}_2\frac{1}{|\mathcal{N}_i|}\sum_{k\in\mathcal{N}_i}\phi(T_k^{L-1})\Big) - \frac{1}{m}\sum_{j=1}^{m}\Big(\boldsymbol{W}_1\boldsymbol{x}_j + \boldsymbol{W}_2\frac{1}{|\mathcal{N}_j|}\sum_{k\in\mathcal{N}_j}\phi(T_k^{L-1})\Big) \right\|$$

$$\leq \mathcal{C}_\sigma\underbrace{\left\| \frac{1}{n}\sum_{i=1}^{n}\boldsymbol{W}_1\boldsymbol{x}_i - \frac{1}{m}\sum_{j=1}^{m}\boldsymbol{W}_1\boldsymbol{x}_j \right\|}_{(a)}$$

$$+ \mathcal{C}_\sigma\underbrace{\left\| \frac{1}{n}\sum_{i=1}^{n}\boldsymbol{W}_2\frac{1}{|\mathcal{N}_i|}\sum_{k\in\mathcal{N}_i}\phi(T_k^{L-1}) - \frac{1}{m}\sum_{j=1}^{m}\boldsymbol{W}_2\frac{1}{|\mathcal{N}_j|}\sum_{k\in\mathcal{N}_j}\phi(T_k^{L-1}) \right\|}_{(b)}.$$

Then, we separately analyze the term (a) and term (b). The term (a) can be bounded as follows:

$$\text{Term (a)} = \mathcal{C}_\sigma\left\| \frac{1}{n}\sum_{i=1}^{n}\boldsymbol{W}_1\boldsymbol{x}_i - \frac{1}{m}\sum_{j=1}^{m}\boldsymbol{W}_1\boldsymbol{x}_j \right\| \leq \mathcal{C}_\sigma\mathcal{B}_{\boldsymbol{W}_1}\left\| \frac{1}{n}\sum_{i=1}^{n}\boldsymbol{x}_i - \frac{1}{m}\sum_{j=1}^{m}\boldsymbol{x}_j \right\|.$$

That is, term (a) is bounded by the distance between the average features of nodes in the first layer of the task-trees (i.e., the nodes directly connected to the root). Next, we bound term (b):

$$\text{Term (b)} = \mathcal{C}_\sigma\left\| \frac{1}{n}\sum_{i=1}^{n}\boldsymbol{W}_2\frac{1}{|\mathcal{N}_i|}\sum_{k1\in\mathcal{N}_i}\phi(T_{k1}^{L-1}) - \frac{1}{m}\sum_{j=1}^{m}\boldsymbol{W}_2\frac{1}{|\mathcal{N}_j|}\sum_{k2\in\mathcal{N}_j}\phi(T_{k2}^{L-1}) \right\|$$

$$\leq \mathcal{C}_\sigma\mathcal{B}_{\boldsymbol{W}_2}\left\| \frac{1}{n}\sum_{i=1}^{n}\frac{1}{|\mathcal{N}_i|}\sum_{k1\in\mathcal{N}_i}\phi(T_{k1}^{L-1}) - \frac{1}{m}\sum_{j=1}^{m}\frac{1}{|\mathcal{N}_j|}\sum_{k2\in\mathcal{N}_j}\phi(T_{k2}^{L-1}) \right\|$$

$$= \mathcal{C}_\sigma\mathcal{B}_{\boldsymbol{W}_2}\left\| \frac{1}{n}\sum_{i=1}^{n}\frac{1}{|\mathcal{N}_i|}\sum_{k1\in\mathcal{N}_i}\sigma\Big(\boldsymbol{W}_1\boldsymbol{x}_{k1} + \boldsymbol{W}_2\frac{1}{|\mathcal{N}_{k1}|}\sum_{s1\in\mathcal{N}_{k1}}\phi(T_{s1}^{L-2})\Big) \right.$$

$$\left. - \frac{1}{m}\sum_{j=1}^{m}\frac{1}{|\mathcal{N}_j|}\sum_{k2\in\mathcal{N}_j}\sigma\Big(\boldsymbol{W}_1\boldsymbol{x}_{k2} + \boldsymbol{W}_2\frac{1}{|\mathcal{N}_{k2}|}\sum_{s2\in\mathcal{N}_{k2}}\phi(T_{s2}^{L-2})\Big) \right\|$$

$$\leq \mathcal{C}_\sigma\mathcal{B}_{\boldsymbol{W}_2}\mathcal{C}_\sigma\mathcal{B}_{\boldsymbol{W}_1}\underbrace{\left\| \frac{1}{n}\sum_{i=1}^{n}\frac{1}{|\mathcal{N}_i|}\sum_{k1\in\mathcal{N}_i}\boldsymbol{x}_{k1} - \frac{1}{m}\sum_{j=1}^{m}\frac{1}{|\mathcal{N}_j|}\sum_{k2\in\mathcal{N}_j}\boldsymbol{x}_{k2} \right\|}_{(c)}$$

$$+ \mathcal{C}_\sigma\mathcal{B}_{\boldsymbol{W}_2}\mathcal{C}_\sigma\mathcal{B}_{\boldsymbol{W}_2}\underbrace{\left\| \frac{1}{n}\sum_{i=1}^{n}\frac{1}{|\mathcal{N}_i|}\sum_{k1\in\mathcal{N}_i}\frac{1}{|\mathcal{N}_{k1}|}\sum_{s1\in\mathcal{N}_{k1}}\phi(T_{s1}^{L-2}) - \frac{1}{m}\sum_{j=1}^{m}\frac{1}{|\mathcal{N}_j|}\sum_{k2\in\mathcal{N}_j}\frac{1}{|\mathcal{N}_{k2}|}\sum_{s2\in\mathcal{N}_{k2}}\phi(T_{s2}^{L-2}) \right\|}_{(d)}.$$

Term (c) describes the distance between the average features of nodes in the second layer of the task-trees, while term (d) follows a recursive formula, similar to term (b). By combining terms (a)

and (b), we have:

$$\Delta \leq \mathcal{C}_\sigma \mathcal{B}_{\boldsymbol{W}_1} \left\| \frac{1}{n} \sum_{i=1}^{n} \boldsymbol{x}_i - \frac{1}{m} \sum_{j=1}^{m} \boldsymbol{x}_j \right\|$$

$$+ \mathcal{C}_\sigma \mathcal{B}_{\boldsymbol{W}_2} \mathcal{C}_\sigma \mathcal{B}_{\boldsymbol{W}_1} \left\| \frac{1}{n} \sum_{i=1}^{n} \frac{1}{|\mathcal{N}_i|} \sum_{k1 \in \mathcal{N}_i} \boldsymbol{x}_{k1} - \frac{1}{m} \sum_{j=1}^{m} \frac{1}{|\mathcal{N}_j|} \sum_{k2 \in \mathcal{N}_j} \boldsymbol{x}_{k2} \right\|$$

$$+ \mathcal{C}_\sigma \mathcal{B}_{\boldsymbol{W}_2} \mathcal{C}_\sigma \mathcal{B}_{\boldsymbol{W}_2} \left\| \frac{1}{n} \sum_{i=1}^{n} \frac{1}{|\mathcal{N}_i|} \sum_{k1 \in \mathcal{N}_i} \frac{1}{|\mathcal{N}_{k1}|} \sum_{s1 \in \mathcal{N}_{k1}} \phi(T_{s1}^{L-2}) - \frac{1}{m} \sum_{j=1}^{m} \frac{1}{|\mathcal{N}_j|} \sum_{k2 \in \mathcal{N}_j} \frac{1}{|\mathcal{N}_{k2}|} \sum_{s2 \in \mathcal{N}_{k2}} \phi(T_{s2}^{L-2}) \right\|.$$

We can extend the formula recursively through all layers until the final layer. The recursive nature of the formula allows us to more easily reformulate the bound:

$$\Delta \leq \mathcal{C}_1 \Delta_1 + \mathcal{C}_2 \Delta_2 + ... + \mathcal{C}_{L-1} \Delta_{L-1}, \tag{10}$$

where $\Delta_l$ denotes the distance between task-trees at the $l$-th layer. For clarity, we can interpret $\mathcal{C}_1 \Delta_1$ as corresponding to Term (a) and $\mathcal{C}_2 \Delta_2$ as corresponding to Term (c). Next, we explain how to determine $\mathcal{C}_l$ and $\Delta_l$ for each layer.

By analyzing the recursive formula, we determine $\mathcal{C}_l$ as follows:

$$\begin{cases} \mathcal{C}_1 & = \mathcal{C}_\sigma \mathcal{B}_{\boldsymbol{W}_1}, \\ \mathcal{C}_2 & = \mathcal{C}_\sigma \mathcal{B}_{\boldsymbol{W}_2} \times \mathcal{C}_\sigma \mathcal{B}_{\boldsymbol{W}_1}, \\ ... \\ \mathcal{C}_l & = (\mathcal{C}_\sigma \mathcal{B}_{\boldsymbol{W}_2})^l \times \mathcal{C}_\sigma \mathcal{B}_{\boldsymbol{W}_1}. \end{cases}$$

We then define the $\Delta_l$. For a concise definition, we introduce an additional notation for describing the subtree information:

$$\begin{cases} \boldsymbol{x}_i^{(0)} & = \boldsymbol{x}_i, \\ \boldsymbol{x}_i^{(1)} & = \frac{1}{|\mathcal{N}_i|} \sum_{j \in \mathcal{N}_i} \boldsymbol{x}_j^{(0)}, \\ \boldsymbol{x}_i^{(2)} & = \frac{1}{|\mathcal{N}_i|} \sum_{j \in \mathcal{N}_i} \boldsymbol{x}_j^{(1)}, \\ ... \\ \boldsymbol{x}_i^{(l)} & = \frac{1}{|\mathcal{N}_i|} \sum_{j \in \mathcal{N}_i} \boldsymbol{x}_j^{(l-1)}. \end{cases}$$

By using the term, we can define the $\Delta_l$ as:

$$\begin{cases} \Delta_1 & = \left\| \frac{1}{n} \sum_{i=1}^{n} \boldsymbol{x}_i^{(0)} - \frac{1}{m} \sum_{j=1}^{m} \boldsymbol{x}_j^{(0)} \right\|, \\ \Delta_2 & = \left\| \frac{1}{n} \sum_{i=1}^{n} \boldsymbol{x}_i^{(1)} - \frac{1}{m} \sum_{j=1}^{m} \boldsymbol{x}_j^{(1)} \right\|, \\ ... \\ \Delta_l & = \left\| \frac{1}{n} \sum_{i=1}^{n} \boldsymbol{x}_i^{(l-1)} - \frac{1}{m} \sum_{j=1}^{m} \boldsymbol{x}_j^{(l-1)} \right\|. \end{cases}$$

By using a formulation like expression 10, we can decompose the impact of different layers, facilitating further analysis of the upper bound on the distance between two task-trees. Next, we will analyze the upper bound of each term. To begin, we first introduce a lemma.

**Lemma D.1.** *Given two sets of random vectors $\mathcal{S}_1 = \{\boldsymbol{v}_1, ..., \boldsymbol{v}_n\}$ and $\mathcal{S}_2 = \{\boldsymbol{v}_1, ..., \boldsymbol{v}_m\}$, the following holds:*

$$\left\| \frac{1}{n} \sum_{i=1}^{n} \boldsymbol{v}_i - \frac{1}{m} \sum_{j=1}^{m} \boldsymbol{v}_j \right\| \leq \frac{1}{nm} \sum_{i=1}^{n} \sum_{j=1}^{m} \left\| \boldsymbol{v}_i - \boldsymbol{v}_j \right\|.$$

*Proof.* Let's consider two sets $A = \{\boldsymbol{a}_1, \boldsymbol{a}_2\}$ and $B = \{\boldsymbol{b}_1, \boldsymbol{b}_2\}$, and $\overline{\boldsymbol{a}} = (\boldsymbol{a}_1 + \boldsymbol{a}_2)/2$, $\overline{\boldsymbol{b}} = (\boldsymbol{b}_1 + \boldsymbol{b}_2)/2$. We have:

$$\begin{aligned} \|\overline{\boldsymbol{a}} - \overline{\boldsymbol{b}}\| &= \|(\boldsymbol{a}_1 + \boldsymbol{a}_2)/2 - (\boldsymbol{b}_1 + \boldsymbol{b}_2)/2\| \\ &= \|2\boldsymbol{a}_1 + 2\boldsymbol{a}_2 - 2\boldsymbol{b}_1 + 2\boldsymbol{b}_2\|/4 \\ &= \|(\boldsymbol{a}_1 - \boldsymbol{b}_1) + (\boldsymbol{a}_1 - \boldsymbol{b}_2) + (\boldsymbol{a}_2 - \boldsymbol{b}_1) + (\boldsymbol{a}_2 - \boldsymbol{b}_2)\|/4 \\ &\leq (\|\boldsymbol{a}_1 - \boldsymbol{b}_1\| + \|\boldsymbol{a}_1 - \boldsymbol{b}_2\| + \|\boldsymbol{a}_2 - \boldsymbol{b}_1\| + \|\boldsymbol{a}_2 - \boldsymbol{b}_2\|)/4 \end{aligned}$$

WLOG, this analysis can be extended to cases where the size of $A$ is $n$ and the size of $B$ is $m$. $\quad\square$

Based on the Lemma, we have

$$\Delta \leq \mathcal{C}_1 \|\frac{1}{n}\sum_{i=1}^{n}\boldsymbol{x}_i^{(0)} - \frac{1}{m}\sum_{j=1}^{m}\boldsymbol{x}_j^{(0)}\| + ... + \mathcal{C}_1\mathcal{C}_2^{L-1}\|\frac{1}{n}\sum_{i=1}^{n}\boldsymbol{x}_i^{(L-1)} - \frac{1}{m}\sum_{j=1}^{m}\boldsymbol{x}_j^{(L-1)}\|$$

$$\leq \frac{1}{nm}\sum_{i=1}^{n}\sum_{j=1}^{m}\Big(\mathcal{C}_1\|\boldsymbol{x}_i^{(0)} - \boldsymbol{x}_j^{(0)}\| + ... + \mathcal{C}_1\mathcal{C}_2^{L-1}\|\boldsymbol{x}_i^{(L-1)} - \boldsymbol{x}_j^{(L-1)}\|\Big),$$

which is displayed in Theorem 3.1.

We then use the Lemma to bound the $\Delta_l$. For example, the upper bound of $\Delta_1$ is:

$$\Delta_1 = \Big\|\frac{1}{n}\sum_{i=1}^{n}\boldsymbol{x}_i^{(0)} - \frac{1}{m}\sum_{j=1}^{m}\boldsymbol{x}_j^{(0)}\Big\|$$

$$\leq \frac{1}{nm}\sum_{i=1}^{n}\sum_{j=1}^{m}\|\boldsymbol{x}_i^{(0)} - \boldsymbol{x}_j^{(0)}\|$$

$$\leq \frac{1}{nm}\sum_{i=1}^{n}\sum_{j=1}^{m}(\|\boldsymbol{x}_i^{(0)}\| + \|\boldsymbol{x}_j^{(0)}\|) \leq 2\mathcal{B}_{\boldsymbol{x}}.$$

Similarly, the upper bound of $\Delta_2$ is:

$$\Delta_2 = \Big\|\frac{1}{n}\sum_{i=1}^{n}\boldsymbol{x}_i^{(1)} - \frac{1}{m}\sum_{j=1}^{m}\boldsymbol{x}_j^{(1)}\Big\|$$

$$\leq \frac{1}{nm}\sum_{i=1}^{n}\sum_{j=1}^{m}\Big\|\boldsymbol{x}_i^{(1)} - \boldsymbol{x}_j^{(1)}\Big\|$$

$$= \frac{1}{nm}\sum_{i=1}^{n}\sum_{j=1}^{m}\Big\|\frac{1}{d_i}\sum_{k_i^{(1)}\in\mathcal{N}_i}\boldsymbol{x}_{k_i^{(1)}}^{(0)} - \frac{1}{d_j}\sum_{k_j^{(1)}\in\mathcal{N}_j}\boldsymbol{x}_{k_j^{(1)}}^{(0)}\Big\|$$

$$\leq \frac{1}{nm}\sum_{i=1}^{n}\sum_{j=1}^{m}\frac{1}{d_id_j}\sum_{k_i^{(1)}\in\mathcal{N}_i}\sum_{k_j^{(1)}\in\mathcal{N}_j}\|\boldsymbol{x}_{k_i^{(1)}}^{(0)} - \boldsymbol{x}_{k_j^{(1)}}^{(0)}\| \leq 2\mathcal{B}_{\boldsymbol{x}},$$

where $d_i = |\mathcal{N}_i|$ and $d_j = |\mathcal{N}_j|$ represent the number of children (i.e., the degree) of nodes $i$ and $j$, respectively. Thus, the upper bound of $\Delta_l$ is:

$$\Delta_l \leq \frac{1}{nm}\sum_{i=1}^{n}\sum_{j=1}^{m}\frac{1}{d_id_j}\underbrace{\sum_{k_i^{(1)}\in\mathcal{N}_i}\sum_{k_j^{(1)}\in\mathcal{N}_j}\frac{1}{d_{k_i^{(1)}}d_{k_j^{(1)}}}\sum_{k_i^{(2)}\in\mathcal{N}_{k_i^{(1)}}}\sum_{k_j^{(2)}\in\mathcal{N}_{k_j^{(1)}}}...}_{(l-1)\times}\|\boldsymbol{x}_{k_i^{(l-1)}}^{(0)} - \boldsymbol{x}_{k_j^{(l-1)}}^{(0)}\| \leq 2\mathcal{B}_{\boldsymbol{x}}.$$

Now we can have the highest upper bound of $\Delta$ as:

$$\Delta \leq (\mathcal{C}_1 + \mathcal{C}_2 + ... + \mathcal{C}_L)2\mathcal{B}_{\boldsymbol{x}}$$

$$= (\mathcal{C}_\sigma\mathcal{B}_{\boldsymbol{W}_1} + ... + (\mathcal{C}_\sigma\mathcal{B}_{\boldsymbol{W}_2})^l \times \mathcal{C}_\sigma\mathcal{B}_{\boldsymbol{W}_1})2\mathcal{B}_{\boldsymbol{x}}$$

$$= 2\mathcal{B}_{\boldsymbol{x}} \cdot \mathcal{C}_\sigma\mathcal{B}_{\boldsymbol{W}_1}\frac{(\mathcal{C}_\sigma\mathcal{B}_{\boldsymbol{W}_2})^L - 1}{\mathcal{C}_\sigma\mathcal{B}_{\boldsymbol{W}_2} - 1}.$$

Note that this upper bound is an extreme case; the bound can be tightened by adding supplementary information or making additional assumptions. Additionally, the distance between task-trees can be reduced by applying techniques like normalization, which lowers the values of the constants.

$\square$

### D.2    PROOF OF THEOREM 3.3

*Proof.* We begin by restating the notations used in the theorem. Let $\mathcal{P}$ represent the task-tree distribution, $\mathcal{T}$ the downstream task distribution, $\phi \in \Phi$ the GNN encoder, $g \in G$ the predictor head used during pretraining, and $f \in \mathcal{F}$ the predictor head for the downstream task. The pretraining objective is defined as $\mathcal{L}_{\mathcal{P}}(g \circ \phi) := \mathbb{E}_{(\hat{T},T)\sim\mathcal{P}}\|g(\phi(\hat{T})) - \phi(T)\|^2$, where $T$ is the task-tree and $\hat{T}$ is the corresponding corrupted task-tree obtained via arbitrary corruption functions. The risk on the downstream task is defined as $\mathcal{R}_{\mathcal{T}}(f \circ \phi) := \mathbb{E}_{(T,y)\sim\mathcal{T}}\kappa(f(\phi(T)), y)$, where $T$ is the task-tree, $y$ is the associated label, and $\kappa$ denotes the loss function. Before we begin the proof, we present an additional helper proposition.

**Proposition D.2** (Ruhe's Trace Inequality (Ruhe, 1970))**.** *If $\boldsymbol{X}$ and $\boldsymbol{Y}$ are positive semi-definite Hermitian matrices with eigenvalues, $x_1 \geq x_2 \geq ... \geq x_n \geq 0$ and $y_1 \geq y_2 \geq ... \geq y_n \geq 0$, then*

$$\sum_{i=1}^{n} x_i y_{n-i+1} \leq \operatorname{tr}(\boldsymbol{XY}) \leq \sum_{i=1}^{n} x_i y_i.$$

Note that we are going to prove

$$\min_{f\in\mathcal{F}} \mathcal{R}_{\mathcal{T}}(f \circ \phi) - \min_{f'\in\mathcal{F}} \mathcal{R}_{\mathcal{T}}(f' \circ \phi') \leq \mathcal{C}_{\delta}\Big(\min_{g\in G} \mathcal{L}_{\mathcal{P}}(g \circ \phi) - \min_{g'\in G} \mathcal{L}_{\mathcal{P}}(g' \circ \phi')\Big)^{\delta},$$

where $\mathcal{C}_{\delta} \approx O(1)$ and $\delta = \frac{1}{2}$. The proof involves deriving the upper bound for the term $\min_{f\in\mathcal{F}} \mathcal{R}_{\mathcal{T}}(f \circ \phi) - \min_{f'\in\mathcal{F}} \mathcal{R}_{\mathcal{T}}(f' \circ \phi')$, followed by the lower bound for $\min_{g\in G} \mathcal{L}_{\mathcal{P}}(g \circ \phi) - \min_{g'\in G} \mathcal{L}_{\mathcal{P}}(g' \circ \phi')$. To simplify the proof, we assume the downstream task is a binary classification task, though this approach can be extended to multi-classification scenarios.

We analyze the upper bound of $\min_{f\in\mathcal{F}} \mathcal{R}_{\mathcal{T}}(f \circ \phi) - \min_{f'\in\mathcal{F}} \mathcal{R}_{\mathcal{T}}(f' \circ \phi')$ as follows:

$$\min_{f\in\mathcal{F}} \mathcal{R}_{\mathcal{T}}(f \circ \phi) - \min_{f'\in\mathcal{F}} \mathcal{R}_{\mathcal{T}}(f' \circ \phi') = \min_{f\in\mathcal{F}} \mathbb{E}_{\mathcal{T}}\kappa(f(\phi(T))) - \min_{f'\in\mathcal{F}} \mathbb{E}_{\mathcal{T}}\kappa(f'(\phi'(T))),$$

where $(T, y) \sim \mathcal{T}$ and $\kappa(f(\phi(T)))$ is shorthand for $\kappa(f(\phi(T)), y)$ for notational convenience. Since we define $f \in \mathcal{F}$ as a linear predictor for binary classification, we can rewrite this equation in the following form:

$$\min_{\|\boldsymbol{\theta}\|\leq\mathcal{B}_{\boldsymbol{\theta}}} \mathbb{E}_{\mathcal{T}}\kappa(\boldsymbol{\theta}^{\top}\phi(T)) - \min_{\|\boldsymbol{\theta}\|\leq\mathcal{B}_{\boldsymbol{\theta}}} \mathbb{E}_{\mathcal{T}}\kappa(\boldsymbol{\theta'}^{\top}\phi'(T))$$

$$\leq \min_{\|\boldsymbol{\theta}\|\leq\mathcal{B}_{\boldsymbol{\theta}}} \mathbb{E}_{\mathcal{T}}\Big|\boldsymbol{\theta}^{\top}\phi(T) - \boldsymbol{\theta'}^{\top}\phi'(T)\Big|$$

$$\leq \min_{\|\boldsymbol{\theta}\|\leq\mathcal{B}_{\boldsymbol{\theta}}} \sqrt{\mathbb{E}_{\mathcal{T}}\Big(\boldsymbol{\theta}^{\top}\phi(T) - \boldsymbol{\theta'}^{\top}\phi'(T)\Big)^2}$$

$$= \min_{\|\boldsymbol{\theta}\|\leq\mathcal{B}_{\boldsymbol{\theta}}} \sqrt{\mathbb{E}_{\mathcal{T}}\Big(\boldsymbol{\theta}^{\top}\phi(T)\phi(T)^{\top}\boldsymbol{\theta} + \boldsymbol{\theta'}^{\top}\phi'(T)\phi'(T)^{\top}\boldsymbol{\theta'} - 2\boldsymbol{\theta}^{\top}\phi(T)\phi'(T)^{\top}\boldsymbol{\theta'}\Big)}$$

$$= \min_{\|\boldsymbol{\theta}\|\leq\mathcal{B}_{\boldsymbol{\theta}}} \sqrt{\boldsymbol{\theta}^{\top}\mathbb{E}[\phi(T)\phi(T)^{\top}]\boldsymbol{\theta} + \boldsymbol{\theta'}^{\top}\mathbb{E}[\phi'(T)\phi'(T)^{\top}]\boldsymbol{\theta'} - 2\boldsymbol{\theta}^{\top}\mathbb{E}[\phi(T)\phi'(T)^{\top}]\boldsymbol{\theta'}}$$

$$\leq \sqrt{\boldsymbol{\theta'}^{\top}\Lambda\boldsymbol{\theta'}}, \Lambda = \mathbb{E}[\phi(T)\phi(T)^{\top}] - \mathbb{E}[\phi(T)\phi'(T)^{\top}]\Big(\mathbb{E}[\phi'(T)\phi'(T)^{\top}]\Big)^{\dagger}\mathbb{E}[\phi(T)\phi'(T)^{\top}]$$

Note that in the previous formula, we define $\boldsymbol{\theta} = (\mathbb{E}[\phi(T)\phi(T)^{\top}])^{\dagger}(\mathbb{E}[\phi(T)\phi'(T)^{\top}])\boldsymbol{\theta'}$. Under the unconstrained setting, the minimum of $\boldsymbol{\theta}^{\top}\mathbb{E}[\phi(T)\phi(T)^{\top}]\boldsymbol{\theta} + \boldsymbol{\theta'}^{\top}\mathbb{E}[\phi'(T)\phi'(T)^{\top}]\boldsymbol{\theta'} - 2\boldsymbol{\theta}^{\top}\mathbb{E}[\phi(T)\phi'(T)^{\top}]\boldsymbol{\theta'}$ reduces to $\boldsymbol{\theta'}^{\top}\Lambda\boldsymbol{\theta'}$ (Deng et al., 2024). We can select a sufficiently large $\mathcal{B}_{\boldsymbol{\theta}}$ to ensure an adequately large function space. Additionally, we define $\boldsymbol{\theta'}$ as the optimal head for the encoder $\phi'$. The expression $\sqrt{\boldsymbol{\theta'}^{\top}\Lambda\boldsymbol{\theta'}}$ is equivalent to $\sqrt{\operatorname{tr}(\Lambda\boldsymbol{\theta'}^{\top}\boldsymbol{\theta'})}$, which can be further simplified using Proposition D.2.

$$\sqrt{\operatorname{tr}(\Lambda\boldsymbol{\theta'}^{\top}\boldsymbol{\theta'})} \leq \sqrt{\sum_{i=1}^{d} \sigma_i(\Lambda)\sigma_i(\boldsymbol{\theta'}\boldsymbol{\theta'}^{\top})} \leq \sqrt{d\sigma_{\max}(\Lambda)\sigma_{\max}(\boldsymbol{\theta'}\boldsymbol{\theta'}^{\top})},$$

where $\sigma_i$ is the $i$-th eigenvalue for the matrix and $\sigma_{\max}$ denotes the maximum eigenvalue.

Then, we are going to demonstrate the lower bound of $\min_{g \in G} \mathcal{L}_{\mathcal{P}}(g \circ \phi) - \min_{g' \in G} \mathcal{L}_{\mathcal{P}}(g' \circ \phi')$.

$$\min_{g \in G} \mathcal{L}_{\mathcal{P}}(g \circ \phi) - \min_{g' \in G} \mathcal{L}_{\mathcal{P}}(g' \circ \phi') = \min_{g \in G} \mathbb{E}_{\mathcal{P}} \|g(\phi(\hat{T})) - \phi(T)\|^2 - \min_{g' \in G} \mathbb{E}_{\mathcal{P}} \|g'(\phi'(\hat{T})) - \phi'(T)\|^2,$$

where $(\hat{T}, T) \sim \mathcal{P}$. Here we also consider the predictor $g$ as a linear function, so that we have the following form:

$$\min_{\|\boldsymbol{W}\| \in \mathcal{B}_{\boldsymbol{W}}} \mathbb{E}_{\mathcal{P}} \|\boldsymbol{W} \phi(\hat{T}) - \phi(T)\|^2 - \min_{\|\boldsymbol{W}'\| \in \mathcal{B}_{\boldsymbol{W}}} \mathbb{E}_{\mathcal{P}} \|\boldsymbol{W}' \phi'(\hat{T}) - \phi'(T)\|^2$$

$$= \min_{\|\boldsymbol{W}\| \in \mathcal{B}_{\boldsymbol{W}}} \mathbb{E}_{\mathcal{P}} \|\boldsymbol{W} \phi(\hat{T}) - \boldsymbol{W}' \phi'(\hat{T})\|^2 + \mathcal{C}_{\mathcal{P}}$$

$$\geq \sum_{r=1}^{d} \min_{\boldsymbol{w}_r} \mathbb{E}_{\mathcal{P}} \|\boldsymbol{w}_r^\top \phi(\hat{T}) - \boldsymbol{w}_r'^\top \phi'(\hat{T})\|^2$$

$$\geq \sum_{r=1}^{d} \min_{\boldsymbol{w}_r} \mathbb{E}_{\mathcal{P}}(\boldsymbol{w}_r^\top \phi(\hat{T}) \phi(\hat{T})^\top \boldsymbol{w}_r + \boldsymbol{w}_r'^\top \phi'(\hat{T}) \phi'(\hat{T})^\top \boldsymbol{w}_r' - 2 \boldsymbol{w}_r^\top \phi(\hat{T}) \phi'(\hat{T})^\top \boldsymbol{w}_r')$$

$$\geq \sum_{r=1}^{d} \boldsymbol{w}_r'^\top \Lambda \boldsymbol{w}_r', \Lambda = \mathbb{E}[\phi(\hat{T})\phi(\hat{T})^\top] - \mathbb{E}[\phi(\hat{T})\phi'(\hat{T})^\top] \left( \mathbb{E}[\phi'(\hat{T})\phi'(\hat{T})^\top] \right)^\dagger \mathbb{E}[\phi(\hat{T})\phi'(\hat{T})^\top].$$

where $\mathcal{C}_{\mathcal{P}} = \mathbb{E}_{\mathcal{P}} \|\phi'(T) - \phi(T)\|^2$ is a constant, and $\boldsymbol{W}'$ is defined as the optimal transformation matrix for the encoder $\phi'$. Based on this formula, we can further simplify the bound on

$$\sum_{r=1}^{d} \boldsymbol{w}_r'^\top \Lambda \boldsymbol{w}_r' = \mathrm{tr}(\Lambda \sum_{r=1}^{d} \boldsymbol{w}_r' \boldsymbol{w}_r'^\top) \geq \sigma_{\max}(\Lambda) \sigma_{\min}(\sum_{r=1}^{d} \boldsymbol{w}_r' \boldsymbol{w}_r'^\top).$$

Now that we have the upper bound for $\min_{f \in \mathcal{F}} \mathcal{R}_{\mathcal{T}}(f \circ \phi) - \min_{f' \in \mathcal{F}} \mathcal{R}_{\mathcal{T}}(f' \circ \phi')$ and the lower bound for $\min_{g \in G} \mathcal{L}_{\mathcal{P}}(g \circ \phi) - \min_{g' \in G} \mathcal{L}_{\mathcal{P}}(g' \circ \phi')$, we can establish the relationship between them as follows:

$$\min_{f \in \mathcal{F}} \mathcal{R}_{\mathcal{T}}(f \circ \phi) - \min_{f' \in \mathcal{F}} \mathcal{R}_{\mathcal{T}}(f' \circ \phi')$$

$$\leq O \left( \frac{\sqrt{d \sigma_{\max}(\boldsymbol{\theta}' \boldsymbol{\theta}'^\top)}}{\sqrt{\sigma_{\min}(\sum_{r=1}^{d} \boldsymbol{w}_r' \boldsymbol{w}_r'^\top)}} \right) (\min_{g \in G} \mathcal{L}_{\mathcal{P}}(g \circ \phi) - \min_{g' \in G} \mathcal{L}_{\mathcal{P}}(g' \circ \phi')).$$

Based on our definition, $\boldsymbol{\theta}'$ and $\boldsymbol{W}'$ are optimal heads for the encoder $\phi'$, the complexity term $O \left( \frac{\sqrt{d \sigma_{\max}(\boldsymbol{\theta}' \boldsymbol{\theta}'^\top)}}{\sqrt{\sigma_{\min}(\sum_{r=1}^{d} \boldsymbol{w}_r' \boldsymbol{w}_r'^\top)}} \right)$ would be a constant which in the order of $O(1)$.

$\square$

### D.3 PROOF OF THEOREM 3.5

*Proof.* We begin by introducing some essential notations. Let $\mathcal{P}$ represent the pretraining task-tree distribution and $\mathcal{T}$ the downstream task-tree distribution. The pair $(T, y) \sim \mathcal{T}$ denotes a task-tree and its corresponding label, where we define the labeling function as $\psi$, meaning $y = \psi(T)$. The GNN encoder is $\phi \in \Phi$, the pretraining predictor is $g \in G$, and the downstream predictor head is $f \in \mathcal{F}$. As in the previous proof, we consider a binary classification task for simplicity, though this can be extended to multi-class settings. The downstream risk is given by $\mathcal{R}_{\mathcal{T}}(f \circ \phi) := \mathbb{E}_{(T, \psi(T)) \sim \mathcal{T}} \kappa(f(\phi(T)), \psi(T))$, where $\kappa$ is a loss function.

Then, we define the excess risk on the downstream distribution $\mathcal{T}$ as

$$
\begin{aligned}
\mathcal{E}(f, \phi) =& \mathcal{R}_\mathcal{T}(f \circ \phi) - \min_{f' \in \mathcal{F}, \phi' \in \Phi} \mathcal{R}_\mathcal{T}(f' \circ \phi') \\
=& \Big( \mathcal{R}_\mathcal{T}(f \circ \phi) - \min_{f' \in \mathcal{F}, \phi' \in \Phi} \mathcal{R}_\mathcal{T}(f' \circ \phi') \Big) + \Big( \min_{f' \in \mathcal{F}} \mathcal{R}_\mathcal{T}(f' \circ \phi) - \min_{f' \in \mathcal{F}} \mathcal{R}_\mathcal{T}(f' \circ \phi) \Big) \\
&+ \Big( \min_{f' \in \mathcal{F}} \mathcal{R}_\mathcal{P}(f' \circ \phi) - \min_{f' \in \mathcal{F}} \mathcal{R}_\mathcal{P}(f' \circ \phi) \Big) + \Big( \min_{f' \in \mathcal{F}} \mathcal{R}_\mathcal{P}(f' \circ \phi^*) - \min_{f' \in \mathcal{F}} \mathcal{R}_\mathcal{P}(f' \circ \phi^*) \Big) \\
=& \underbrace{\mathcal{R}_\mathcal{T}(f \circ \phi) - \min_{f' \in \mathcal{F}} \mathcal{R}_\mathcal{T}(f' \circ \phi)}_{(a)} + \underbrace{\min_{f' \in \mathcal{F}} \mathcal{R}_\mathcal{T}(f' \circ \phi) - \min_{f' \in \mathcal{F}} \mathcal{R}_\mathcal{P}(f' \circ \phi)}_{(b)} \\
&+ \underbrace{\min_{f' \in \mathcal{F}} \mathcal{R}_\mathcal{P}(f' \circ \phi) - \min_{f' \in \mathcal{F}} \mathcal{R}_\mathcal{P}(f' \circ \phi^*)}_{(c)} + \underbrace{\min_{f' \in \mathcal{F}} \mathcal{R}_\mathcal{P}(f' \circ \phi^*) - \min_{f' \in \mathcal{F}, \phi' \in \Phi} \mathcal{R}_\mathcal{T}(f' \circ \phi')}_{(d)},
\end{aligned}
$$

where $\phi$ and $f$ represent encoder obtained during pretraining and the prediction head learned in downstream task, respectively, while $\phi'$ and $f'$ are the optimal encoder and predictor head on the downstream distribution. $\phi^*$ is the optimal encoder obtained during pretraining, defined as $\phi^* = \arg\min_{\phi \in \Phi} \min_{g \in G} \mathcal{L}_\mathcal{P}(g \circ \phi)$. We will analyze these four terms separately.

To bound the term (a), we need to introduce the empirical Rademacher complexity (Definition 1, (Deng et al., 2024)) as

$$
\hat{\mathfrak{R}}_\mathcal{T} := \mathbb{E}_{\varepsilon \in \{\pm 1\}^n} \Big[ \sup_{f \in \mathcal{F}} \frac{1}{n} \sum_{i=1}^n \varepsilon_i \kappa(f \circ \phi(T), \psi(T)) \Big],
$$

where $\varepsilon_i$ is i.i.d., and $\mathbb{P}(\varepsilon = 1) = \mathbb{P}(\varepsilon = -1) = \frac{1}{2}$.

Using this definition, we can bound the term (a):

$$
\begin{aligned}
\text{Term (a)} =& \mathcal{R}_\mathcal{T}(f \circ \phi) - \min_{f' \in \mathcal{F}} \mathcal{R}_\mathcal{T}(f' \circ \phi) \\
=& \underbrace{\mathcal{R}_\mathcal{T}(f \circ \phi) - \hat{\mathcal{R}}_\mathcal{T}(f \circ \phi)}_{(a.1)} + \underbrace{\hat{\mathcal{R}}_\mathcal{T}(f^* \circ \phi) - \min_{f' \in \mathcal{F}} \mathcal{R}_\mathcal{T}(f' \circ \phi)}_{(a.2)} + \underbrace{\hat{\mathcal{R}}_\mathcal{T}(f \circ \phi) - \hat{\mathcal{R}}_\mathcal{T}(f^* \circ \phi)}_{(a.3)},
\end{aligned}
$$

where $f^*$ is the optimal predictor head over the distribution $\mathcal{T}$, defined as $f^* = \arg\min_{f \in \mathcal{F}} \mathcal{R}_\mathcal{T}(f \circ \phi)$. The term (a.3) represents the empirical risk gap between the learned head $f$ and the best head $f^*$, which implies that the term is a constant greater than or equal to 0. Term (a.1) and (a.2) describe the gap between the risk and the empirical risk. According to uniform convergence, these two terms can be expressed in terms of empirical Rademacher complexity. Thus, term (a) can be bounded as:

$$
\text{Term (a)} \leq 4\hat{\mathfrak{R}}_\mathcal{T} + 4\mathcal{B}_\kappa \sqrt{\frac{\log(1/v)}{n}},
$$

where $\mathcal{B}_\kappa$ is the bound of the Lipschitz of loss function $\kappa$. We then further simplify the empirical Rademacher complexity for a more reasonable expression.

$$
\begin{aligned}
\text{Term (a)} \leq& 4\mathbb{E}_{\varepsilon \in \{\pm 1\}^n} \Big[ \sup_{f \in \mathcal{F}} \frac{1}{n} \sum_{i=1}^n \varepsilon_i \kappa(f \circ \phi(T_i), \psi(T_i)) \Big] + 4\mathcal{B}_\kappa \sqrt{\frac{\log(1/v)}{n}} \\
\leq& 4\mathcal{C}_\kappa \mathbb{E}_{\varepsilon \in \{\pm 1\}^n} \Big[ \sup_{f \in \mathcal{F}} \frac{1}{n} \sum_{i=1}^n \varepsilon_i f \circ \phi(T_i) \Big] + 4\mathcal{B}_\kappa \sqrt{\frac{\log(1/v)}{n}} \\
\leq& 4\mathcal{C}_\kappa \mathcal{C}_f \mathbb{E}_{\varepsilon \in \{\pm 1\}^n} \Big\| \frac{1}{n} \sum_{i=1}^n \varepsilon_i \phi(T_i) \Big\| + 4\mathcal{B}_\kappa \sqrt{\frac{\log(1/v)}{n}} \\
\leq& \frac{4}{n} \mathcal{C}_\kappa \mathcal{C}_f \mathbb{E}_{\varepsilon \in \{\pm 1\}^n} \sqrt{\Big\| \sum_{i=1}^n \varepsilon_i \phi(T_i) \Big\|^2} + 4\mathcal{B}_\kappa \sqrt{\frac{\log(1/v)}{n}}.
\end{aligned}
$$

As the $\varepsilon_i$ are i.i.d. with zero mean as our definition, we cancel the term, thus

$$\text{Term (a)} \leq \frac{4}{n} \mathcal{C}_\kappa \mathcal{C}_f \sqrt{\sum_{i=1}^{n} \left\| \phi(T_i) \right\|^2} + 4\mathcal{B}_\kappa \sqrt{\frac{\log(1/v)}{n}}.$$

Then, we bound the term (b). To do this, we introduce a notation $f^*_\mathcal{P} = \arg\min_{f' \in \mathcal{F}} \mathcal{R}_\mathcal{P}(f' \circ h)$.

$$\begin{aligned}
\text{Term (b)} &= \min_{f' \in \mathcal{F}} \mathcal{R}_\mathcal{T}(f' \circ \phi) - \min_{f' \in \mathcal{F}} \mathcal{R}_\mathcal{P}(f' \circ \phi) \\
&\leq \mathcal{R}_\mathcal{T}(f^*_\mathcal{P} \circ \phi) - \mathcal{R}_\mathcal{P}(f^*_\mathcal{P} \circ \phi) \\
&= \mathbb{E}_{T \sim \mathcal{T}} \Big[ \kappa(f^*_\mathcal{P} \circ \phi(T), \psi(T)) \Big] - \mathbb{E}_{T \sim \mathcal{P}} \Big[ \kappa(f^*_\mathcal{P} \circ \phi(T), \psi(T)) \Big] \\
&= \mathbb{E}_{x \sim \mathcal{T}_\phi} \Big[ \kappa(f^*_\mathcal{P}(x), \psi(T)) \Big] - \mathbb{E}_{x \sim \mathcal{P}_\phi} \Big[ \kappa(f^*_\mathcal{P}(x), \psi(T)) \Big] \\
&\leq \mathcal{B}_\kappa \sum_{x \in \mathcal{X}_\phi} \left\| \mathcal{T}_\phi(x) - \mathcal{P}_\phi(x) \right\|,
\end{aligned}$$

where $\mathcal{B}_\kappa$ represents the upper bound of the Lipschitz constant of $\kappa$, and $\mathcal{X}_\phi$ denotes the distribution of task-tree embeddings produced by the encoder $\phi$. This term measures the distributional distance of task-trees between the pretraining and downstream distributions.

Following, we bound the term (c), as

$$\begin{aligned}
\text{Term (c)} &= \min_{f' \in \mathcal{F}} \mathcal{R}_\mathcal{P}(f' \circ \phi) - \min_{f' \in \mathcal{F}} \mathcal{R}_\mathcal{P}(f' \circ \phi^*) \\
&\leq \mathcal{C}_\delta \Big( \min_{g' \in G} \mathcal{L}_\mathcal{P}(g' \circ h) - \min_{g' \in G} \mathcal{L}_\mathcal{P}(g' \circ \phi^*) \Big)^\delta \\
&\leq \mathcal{C}_\delta \Big( \mathcal{L}_\mathcal{P}(g \circ h) - \min_{g' \in G, \phi' \in \Phi} \mathcal{L}_\mathcal{P}(g' \circ \phi') \Big)^\delta.
\end{aligned}$$

The term $\mathcal{L}_\mathcal{P}(g \circ h) - \min_{g' \in G, \phi' \in \Phi} \mathcal{L}_\mathcal{P}(g' \circ \phi')$ describes the excess risk on pretraining task, which can be replaced by a notation $\mathcal{E}_\mathcal{P}(g, \phi)$.

Lastly, we bound the term (d),

$$\begin{aligned}
\text{Term (d)} &= \min_{f' \in \mathcal{F}} \mathcal{R}_\mathcal{P}(f' \circ \phi^*) - \min_{f' \in \mathcal{F}, \phi' \in \Phi} \mathcal{R}_\mathcal{T}(f' \circ \phi') \\
&= \min_{f' \in \mathcal{F}} \mathcal{R}_\mathcal{P}(f' \circ \phi^*) - \min_{f' \in \mathcal{F}} \mathcal{R}_\mathcal{T}(f \circ \phi^*) + \min_{f' \in \mathcal{F}} \mathcal{R}_\mathcal{T}(f \circ \phi^*) - \min_{f' \in \mathcal{F}, \phi' \in \Phi} \mathcal{R}_\mathcal{T}(f' \circ \phi') \\
&\leq \mathcal{B}_\kappa \sum_{x \in \mathcal{X}_\phi} \left\| \mathcal{T}_\phi(x) - \mathcal{P}_\phi(x) \right\| + \min_{f' \in \mathcal{F}} \mathcal{R}_\mathcal{T}(f' \circ \phi^*) - \min_{f' \in \mathcal{F}, \phi' \in \Phi} \mathcal{R}_\mathcal{T}(f' \circ \phi').
\end{aligned}$$

By combining the four terms, we obtain the generalization bound for a model pretrained on task-tree distribution $\mathcal{P}$ and fine-tuned on task-tree distribution $\mathcal{T}$:

$$\begin{aligned}
\mathcal{R}_\mathcal{T}(f \circ \phi) \leq{}& \mathcal{C}_\delta \Big( \mathcal{E}_\mathcal{P}(g, \phi) \Big)^\delta + \frac{4\mathcal{C}_\kappa \mathcal{C}_f}{n} \sqrt{\sum_{i=1}^{n} \left\| \phi(T_i) \right\|^2} + \min_{f' \in \mathcal{F}} \mathcal{R}_\mathcal{T}(f' \circ \phi^*) \\
&+ 2\mathcal{B}_\kappa \Big( \sum_{x \in \mathcal{X}_\phi} \left\| \mathcal{T}_\phi(x) - \mathcal{P}_\phi(x) \right\| + 2\sqrt{\frac{\log(1/v)}{n}} \Big).
\end{aligned}$$

We can set $\mathcal{C}_1 = \mathcal{C}_\kappa \mathcal{C}_f$ and $\mathcal{C}_2 = \mathcal{B}_\kappa$ as two downstream task-related constants. $\qquad\square$

## E EXPERIMENTAL SETTINGS

### E.1 DATASETS

**Dataset Statistics.** We utilize 32 datasets spanning five domains in this paper. Since these datasets are text-attributed graphs, we use Sentence-BERT (Reimers & Gurevych, 2019) to align the node

Table 7: Statistics of 32 graphs used in the paper.

| Dataset | Domain | Task | # Nodes | # Edges | # Classes | # Task-Trees | Source |
|---|---|---|---|---|---|---|---|
| Products | E-commerce | Node, Link | 316,513 | 19,337,745 | 39 | 316,513 | (Chen et al., 2024b) |
| History | E-commerce | Node, Link | 41,551 | 503,180 | 12 | 41,551 | (Chen et al., 2024b) |
| Children | E-commerce | Node, Link | 76,875 | 2,325,044 | 24 | 76,875 | (Chen et al., 2024b) |
| Computer | E-commerce | Node, Link | 87,229 | 1,256,548 | 10 | 87,229 | (Chen et al., 2024b) |
| Photo | E-commerce | Node, Link | 48,362 | 873,793 | 12 | 48,362 | (Chen et al., 2024b) |
| Sportsfit | E-commerce | Node, Link | 173,055 | 3,020,134 | 13 | 173,055 | (Chen et al., 2024b) |
| Ratings | E-commerce | Node, Link | 24,492 | 186,100 | 5 | 24,492 | (Chen et al., 2024b) |
| Arxiv | Academia | Node, Link | 169,343 | 2,315,598 | 40 | 169,343 | (Chen et al., 2024b) |
| Cora | Academia | Node, Link | 2,708 | 10,556 | 7 | 2,708 | (Chen et al., 2024b) |
| Citeseer | Academia | Node, Link | 3,186 | 8,450 | 6 | 3,186 | (Chen et al., 2024b) |
| Pubmed | Academia | Node, Link | 19,717 | 88,648 | 3 | 19,717 | (Chen et al., 2024b) |
| Arxiv 23 | Academia | Node, Link | 46,198 | 77,726 | 40 | 46,198 | (Chen et al., 2024b) |
| DBLP | Academia | Node, Link | 14,376 | 431,326 | 4 | 14,376 | (Chen et al., 2024b) |
| WN18RR | knowledge Base | Link | 40,943 | 93,003 | 11 | 93,003 | (Galkin et al., 2024) |
| FB15K237 | knowledge Base | Link | 14,541 | 310,116 | 237 | 310,116 | (Galkin et al., 2024) |
| Codex Small | knowledge Base | Link | 2,034 | 36,543 | 42 | 36,543 | (Galkin et al., 2024) |
| Codex Median | knowledge Base | Link | 17,050 | 206,205 | 51 | 206,205 | (Galkin et al., 2024) |
| Codex Large | knowledge Base | Link | 77,951 | 612,437 | 69 | 612,437 | (Galkin et al., 2024) |
| NELL995 | knowledge Base | Link | 74,536 | 153,039 | 200 | 153,039 | (Galkin et al., 2024) |
| GDELT | knowledge Base | Link | 5,849 | 943,956 | 237 | 943,956 | (Zhang et al., 2024b) |
| ICEWS1819 | knowledge Base | Link | 31,796 | 1,100,071 | 266 | 1,100,071 | (Zhang et al., 2024b) |
| Chemblpre | Molecule | Graph | 8,845,648 | 19,123,034 | 1,295 | 341,952 | (Feng et al., 2024) |
| PCBA | Molecule | Graph | 11,349,235 | 24,566,048 | 128 | 437,092 | (Feng et al., 2024) |
| HIV | Molecule | Graph | 1,049,163 | 2,259,376 | 1 | 41,127 | (Feng et al., 2024) |
| BBBP | Molecule | Graph | 49,068 | 105,842 | 1 | 2,039 | (Feng et al., 2024) |
| BACE | Molecule | Graph | 51,577 | 111,536 | 1 | 1,513 | (Feng et al., 2024) |
| TOXCAST | Molecule | Graph | 161,002 | 330,180 | 588 | 8,575 | (Feng et al., 2024) |
| CYP450 | Molecule | Graph | 414,367 | 895,886 | 5 | 16,896 | (Feng et al., 2024) |
| TOX21 | Molecule | Graph | 145,459 | 302,190 | 12 | 7,831 | (Feng et al., 2024) |
| MUV | Molecule | Graph | 2,255,846 | 4,892,252 | 17 | 93,087 | (Feng et al., 2024) |
| Enron | Temporal | Link | 42,712 | 797,907 | 10 | 797,907 | (Zhang et al., 2024b) |
| Googlemap CT | Temporal | Link | 111,169 | 1,380,623 | 5 | 1,380,623 | (Zhang et al., 2024b) |

textual features into 768-dimensional vectors. The dataset statistics are presented in Table 7. For the temporal graphs, we split each graph into 10 snapshots, with the statistics shown in Figure 10. We classify Children and Ratings as heterophily graphs due to their relatively low homophily ratios (Chen et al., 2024b).

**Splitter.** For each dataset, we use the same splitting strategy as provided in the original paper (Chen et al., 2024b; Galkin et al., 2024; Feng et al., 2024; Zhang et al., 2024b). If multiple splits are provided, we evaluate model performance on each split using different random seeds. For datasets with a single split, we repeat the experiments five times with different random seeds. For GDELT and ICEWS1819, which are originally temporal knowledge graphs, we apply an 80%/10%/10% split based on timestamps for train/validation/test settings. For the temporal graphs Enron and Googlemap CT used for edge classification, we split each snapshot by timestamps, using the first 70% for training, the next 15% for validation, and the remaining 15% for testing.

### E.2 BASELINES

#### BASELINES APPLICABLE FOR ALL GRAPHS

**GCN (Kipf & Welling, 2017).** A supervised message-passing GNN trained from scratch for each task. As a result, it cannot be applied to in-context learning or zero-shot learning.

**GAT (Veličković et al., 2018).** A supervised GNN that uses an attention mechanism to learn the importance of received messages.

**GIN (Xu et al., 2019).** A supervised GNN specifically designed for graph-level tasks.

**BGRL (Thakoor et al., 2022).** A popular self-supervised learning framework for graphs that employs a contrastive learning loss without negative samples.

**GraphMAE (Hou et al., 2022).** A graph learning framework pretrained in a masked auto-encoder fashion.

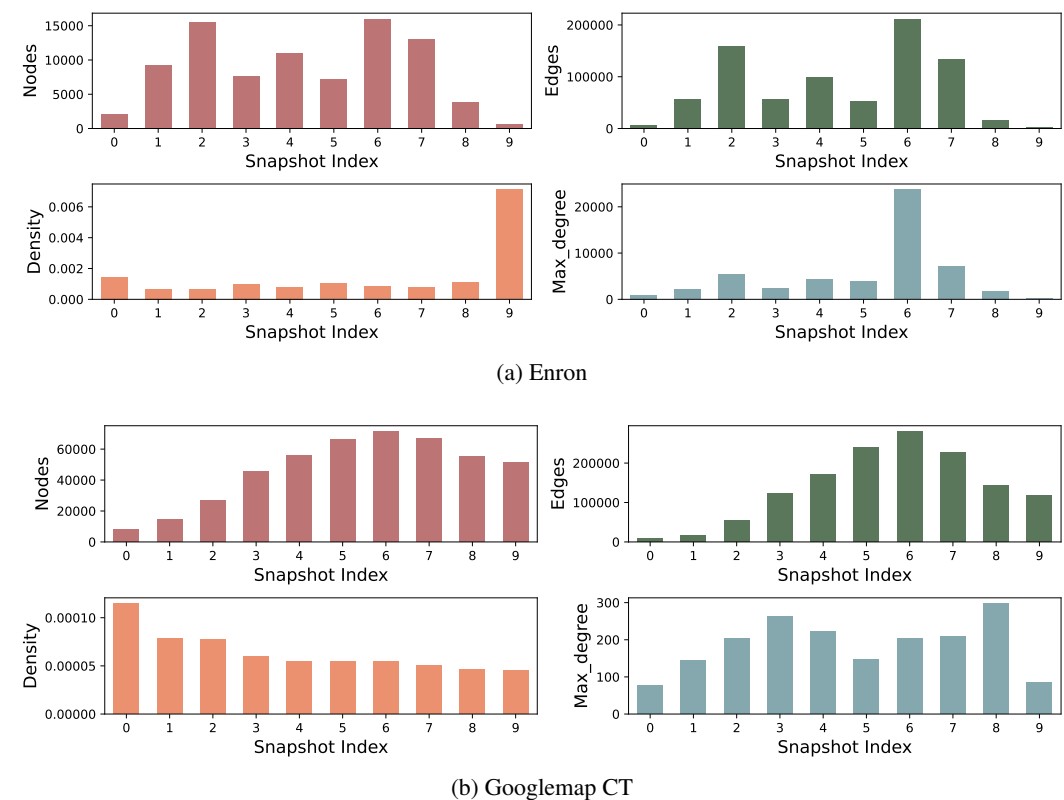

(a) Enron

(b) Googlemap CT

Figure 10: The statistics of temporal graphs.

**OFA (Liu et al., 2024).** A cross-task and cross-domain graph foundation model that treats subgraphs as the basic learning instances. It introduces a graph prompt learning framework to enable in-context and zero-shot learning.

EXPERT MODELS DESIGNED FOR SPECIFIC DOMAINS

**ULTRA (Galkin et al., 2024).** A foundation model designed specifically for knowledge graphs, which we treat as the domain expert for KGs.

**KVPLM (Zeng et al., 2022).** A language model based on SMILES representations of molecules, serving as an expert model for molecular graphs.

**MoMu (Su et al., 2022).** Another expert model for molecules that leverages GNNs to improve molecular representations.

**Galactica (Taylor et al., 2022).** A foundation model for molecular graphs that utilizes multi-task learning with instructions.

**GIMLET (Zhao et al., 2023).** A foundation model for molecules that incorporates advanced models and instruction-based learning.

FEW-SHOT LEARNING METHODS

**GPN (Ding et al., 2020).** GPN is a framework that leverages GNN and meta-learning to address few-shot node classification by learning a transferable metric space.

**TENT (Wang et al., 2022b).** TENT introduces three levels of adaptation—node-level, class-level, and task-level—to mitigate task variance and improve the model's generalization performance across different meta-tasks.

Table 8: The hyper-parameters used in the pretraining.

| Hidden Dim | Layers | Dropout | Activation | Epochs | LR |
|:---:|:---:|:---:|:---:|:---:|:---:|
| 768 | 2 | 0.15 | ReLU | 10 | 1e-7 |
| Feature Drop | Edge Drop | $\lambda$ | Decay | BS | Fanout |
| 0.2 | 0.2 | 10 | 1e-8 | 4,096 | 10 |

**GLITTER (Wang et al., 2022a).** Enhance few-shot node classification by learning task-specific graph structures for each meta-task using node influence and mutual information.

**TLP (Tan et al., 2022).** transfer pretrained node embeddings fine-tunes a simple linear classifier on novel classes.

**Prodigy (Huang et al., 2023).** Enable in-context learning over graphs by designing graph prompt learning template.

### E.3 Evaluation Protocol

**Pretraining Datasets.** We select six datasets for pretraining, including `Arxiv`, `Products`, `WN18RR`, `FB15K237`, `Chemblpre`, and `PCBA`, due to their diversity in domains and tasks. For self-supervised learning methods, these six datasets are used for pretraining unless otherwise specified.

**SFT Datasets.** For specialization via SFT in each domain, we use `Arxiv` for academic networks, `Products` for e-commerce networks, `FB15K237` for knowledge graphs, and `PCBA` for molecular networks. For temporal graphs, which are e-commerce-based, we also use `Products` for SFT to evaluate robustness under temporal distribution shifts.

**Backbone.** We use a GraphSAGE-like encoder (Hamilton et al., 2017). Following the encoding of task-trees, we add an additional linear transformation as the projector. Note that we does not leverage edge features to make the task harder except for `Enron` and `Googlemap CT` where node features are IDs and edge contains messages. As the edge information may significantly benefit some tasks like knowledge graph completion and molecule property prediction.

### E.4 Evaluation Settings

**Finetune.** This is the basic setting that directly finetune the full parameters of the pretrained model by appending a linear classifier on the top of the model encoder.

**In-context Learning.** This is a kind of few-shot learning without fine-tuning the model parameters. We randomly select $k$ samples from a certain class, and average the selected samples to form prototypes, which is used for classification. We follow existing GFM works (Liu et al., 2024; He & Hooi, 2024) to conduct 500 randomly sampled 5-way 3-shot learning tasks. If the number of classes is less than 5, the number of ways is set to the number of classes.

**Zero-shot Learning.** The zero-shot learning is similar to in-context learning, yet we use the LLM-encoded class description embeddings as the prototypes for prediction. Similar to in-context learning, we also randomly sample 500 tasks for evaluation. Another zero-shot setting involves using an additional LLM for zero-shot inference (Chen et al., 2024a). We leave this in our future work.

### E.5 Hyper-Parameters

**Baselines.** For the baseline methods, we follow the hyperparameters reported in (Liu et al., 2024; Chen et al., 2024b). If the hyperparameters are not provided, we set the number of epochs to 1,000, the batch size to 4,096, early stopping at 200, and the hidden dimension to 768, using a 2-layer GraphSAGE as the backbone with batch normalization and ReLU activation. For optimization, we use AdamW with a weight decay of 1e-6 and tune the learning rate from 1e-3, 1e-4, 1e-5, reporting the best performance. For methods with attention mechanisms, we set 4 attention heads.

Table 9: The hyper-parameters used in fine-tuning on academic networks.

| Academia | Cora | Citeseer | Pubmed | Arxiv23 | DBLP | Arxiv |
|---|---|---|---|---|---|---|
| Normalize | None | BN | None | None | None | BN |
| Learning Rate | 1e-4 | 1e-4 | 1e-5 | 1e-4 | 1e-4 | 1e-3 |
| Weight Decay | 0 | 0 | 1e-6 | 0 | 1e-6 | 1e-6 |
| Epochs | 1000 | 1000 | 1000 | 1000 | 1000 | 1000 |
| Early Stop | 200 | 200 | 200 | 200 | 200 | 200 |
| SFT Learning Rate | 1e-7 | 1e-4 | 1e-6 | 1e-5 | 1e-7 | 1e-6 |
| SFT Epochs | 100 | 100 | 100 | 100 | 100 | 100 |

Table 10: The hyper-parameters used in fine-tuning on e-commerce networks.

| E-commerce | History | Children | Computer | Photo | Sportsfit | Ratings | Products |
|---|---|---|---|---|---|---|---|
| Normalize | None | None | None | None | BN | BN | BN |
| Learning Rate | 1e-3 | 1e-4 | 1e-3 | 1e-4 | 1e-4 | 1e-4 | 1e-4 |
| Weight Decay | 0 | 1e-6 | 0 | 1e-6 | 1e-6 | 1e-6 | 1e-6 |
| Epochs | 1000 | 1000 | 1000 | 1000 | 1000 | 1000 | 1000 |
| Early Stop | 200 | 200 | 200 | 200 | 200 | 200 | 200 |
| SFT Learning Rate | 1e-5 | 1e-7 | 1e-7 | 1e-7 | 1e-8 | 1e-7 | 1e-7 |
| SFT Epochs | 100 | 100 | 100 | 100 | 100 | 100 | 100 |

Table 11: The hyper-parameters used in fine-tuning on knowledge graphs.

| KG | WN18RR | Codex-S | Codex-M | Codex-L | NELL995 | GDELT | ICEWS1819 | FB15K237 |
|---|---|---|---|---|---|---|---|---|
| Normalize | BN | BN | BN | BN | BN | BN | BN | BN |
| Learning Rate | 1e-4 | 1e-4 | 1e-3 | 1e-3 | 1e-3 | 1e-3 | 1e-3 | 1e-3 |
| Weight Decay | 1e-6 | 1e-6 | 1e-6 | 1e-6 | 0 | 0 | 1e-6 | 1e-6 |
| Epochs | 1000 | 1000 | 1000 | 1000 | 1000 | 1000 | 1000 | 1000 |
| Early Stop | 200 | 200 | 200 | 200 | 200 | 200 | 200 | 200 |
| SFT Learning Rate | 1e-8 | 1e-7 | 1e-5 | 1e-7 | 1e-8 | 1e-4 | 1e-8 | 1e-7 |
| SFT Epochs | 100 | 100 | 100 | 100 | 100 | 100 | 100 | 100 |

Table 12: The hyper-parameters used in fine-tuning on molecule graphs.

| Molecule | BBBP | BACE | TOXCAST | TOX21 | CYP450 | HIV | MIV | PCBA |
|---|---|---|---|---|---|---|---|---|
| Normalize | BN | BN | BN | BN | BN | BN | None | BN |
| Learning Rate | 1e-3 | 1e-4 | 1e-4 | 1e-4 | 1e-4 | 1e-5 | 1e-4 | 1e-5 |
| Weight Decay | 1e-6 | 0 | 1e-6 | 1e-6 | 1e-6 | 0 | 0 | 0 |
| Epochs | 300 | 300 | 300 | 300 | 300 | 300 | 300 | 300 |
| Early Stop | 30 | 30 | 30 | 30 | 30 | 30 | 30 | 30 |
| SFT Learning Rate | 1e-7 | 1e-6 | 1e-8 | 1e-7 | 1e-7 | 1e-7 | 1e-6 | 1e-7 |
| SFT Epochs | 10 | 10 | 10 | 10 | 10 | 10 | 10 | 10 |

Table 13: The hyper-parameters used in fine-tuning on temporal graphs.

| Temporal | Enron | Googlemap CT |
|---|---|---|
| Normalize | None | None |
| Learning Rate | 1e-3 | 1e-3 |
| Weight Decay | 1e-6 | 1e-6 |
| Epochs | 1000 | 1000 |
| Early Stop | 200 | 200 |
| SFT Learning Rate | 1e-6 | 1e-6 |
| SFT Epochs | 100 | 100 |

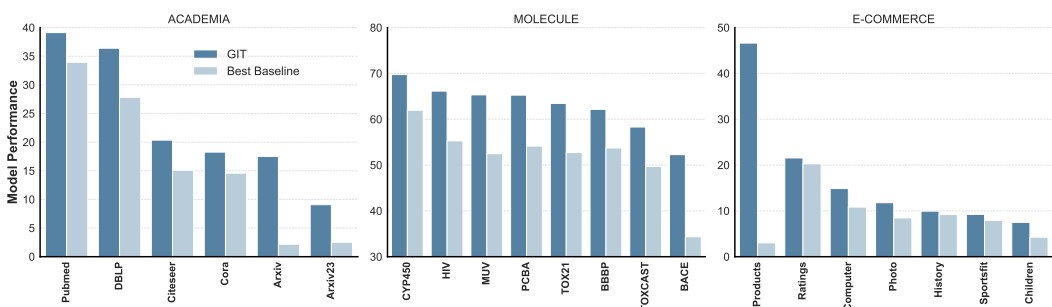

Figure 11: The model performance on all datasets in the zero-shot setting.

**GIT.** The model architecture and pretraining parameters of our GIT are presented in Table 8. The specific fine-tuning hyperparameters, categorized by domain, are shown in Tables 9, 10, 11, 12, and 13. For in-context learning and zero-shot learning results without fine-tuning, the general model does not involve any hyperparameters. For the specialized model, we tune the hyperparameters of SFT epochs from 10 to 500, in steps of 10, the SFT learning rate from 1e-4, 1e-5, 1e-6, 1e-7, 1e-8, and the normalization method from None, BN.

# F    ADDITIONAL RESULTS

## F.1    MAIN RESULTS

**In-context Learning.** The model performance in in-context learning, along with the best baselines, is shown in Figure 3. The results are presented from best to worst. It is evident that the specialized version significantly improves the general model, particularly in molecule and knowledge graphs. An interesting observation is that, while specialization enhances performance across datasets within the domain, it does not necessarily lead to a significant improvement on the dataset used for specialization. We hypothesize this is due to the use of prototypes for predictions, where the prototypes are constructed via random sampling, which may not be directly influenced by the supervised fine-tuning process. Detailed experimental results are presented in the following sections.

**Zero-shot Learning.** The zero-shot learning results across all datasets are shown in Figure 11. Compared to basic fine-tuning and in-context learning, the improvements in zero-shot performance are the most pronounced. Unlike the observations in in-context learning, the model performance on SFT datasets (`Arxiv`, `Products`, `PCBA`) is significantly higher. This is likely because the model is directly fine-tuned using label description embeddings, which are also used as prototypes during predictions. To reduce the influence of SFT data on zero-shot learning, we report the average performance on held-out graphs (i.e., graphs not used in pretraining or specialization) in Table 1 of the main paper.

## F.2    DOMAIN: ACADEMIA

**Node Classification.** We perform node classification on academic networks across three settings: basic fine-tuning, 3-shot in-context learning, and zero-shot learning. The comprehensive node classification results on academic networks, measured in terms of accuracy, are presented in Table 14. Notably, the specialized model (GIT-S) does not always outperform the general model (GIT-G). This may be because the manually selected SFT data does not adequately capture the underlying distribution of the domain. It would be valuable to explore dataset selection or instance selection methods to better optimize the choice of SFT data.

**Link Prediction.** We present the link prediction results, measured by AUC, on academic networks in Table 15. The train/val/test sets are randomly split in a 70%/15%/15% ratio. GIT outperforms all baselines across all settings. Additionally, the specialized GIT surpasses the general GIT, highlighting the potential of specialization to enhance performance on other tasks within the same domain. This finding underscores the cross-task transferability of the proposed specialization process.

Table 14: Node classification results on academic networks in terms of accuracy.

|  |  | Cora | Citeseer | Pubmed | Arxiv23 | DBLP | Arxiv | **Avg.** |
|---|---|---|---|---|---|---|---|---|
| **0-shot** | GCN | - | - | - | - | - | - | - |
|  | BGRL | 14.37 ± 0.38 | 15.09 ± 0.40 | 33.94 ± 0.46 | 2.44 ± 0.23 | 25.53 ± 0.27 | 2.10 ± 0.14 | 15.58 |
|  | GraphMAE | 13.88 ± 0.41 | 13.48 ± 0.83 | 32.62 ± 0.67 | 2.51 ± 0.37 | 27.83 ± 0.40 | 2.17 ± 0.26 | 15.42 |
|  | OFA | 14.58 ± 0.43 | 13.28 ± 0.12 | 30.89 ± 0.10 | 2.08 ± 0.03 | 21.00 ± 0.27 | 2.05 ± 0.18 | 13.98 |
|  | GIT - G | 15.31 ± 0.27 | 16.04 ± 0.31 | 29.66 ± 0.60 | 2.89 ± 0.25 | 21.80 ± 0.35 | 3.57 ± 0.18 | 14.88 |
|  | GIT - S | **18.26 ± 0.29** | **20.35 ± 0.29** | **39.12 ± 0.55** | **9.08 ± 0.32** | **36.40 ± 0.58** | **17.50 ± 0.66** | **23.45** |
| **3-shot** | GCN | - | - | - | - | - | - | - |
|  | BGRL | 61.24 ± 0.50 | 44.97 ± 0.43 | 54.55 ± 0.81 | 43.17 ± 0.93 | 42.89 ± 0.61 | 59.09 ± 0.24 | 50.99 |
|  | GraphMAE | 62.02 ± 0.58 | 44.08 ± 0.59 | 55.98 ± 0.68 | 31.64 ± 0.28 | 38.16 ± 0.54 | 63.62 ± 0.79 | 49.25 |
|  | OFA | 55.92 ± 0.40 | 41.57 ± 0.32 | 40.89 ± 0.79 | 37.01 ± 0.41 | 43.08 ± 0.51 | 57.08 ± 0.48 | 45.93 |
|  | GIT - G | 60.93 ± 0.47 | 48.32 ± 0.53 | **60.30 ± 0.76** | 45.62 ± 0.35 | 44.76 ± 0.54 | **64.07 ± 0.50** | 54.00 |
|  | GIT - S | **63.23 ± 0.29** | **49.55 ± 0.33** | 59.62 ± 0.54 | **47.21 ± 0.31** | **47.40 ± 0.43** | 64.06 ± 0.58 | **55.18** |
| **Finetune** | GCN | 77.40 ± 1.36 | 80.19 ± 1.30 | 72.44 ± 2.08 | 71.61 ± 0.02 | 68.15 ± 0.14 | 71.65 ± 0.02 | 73.57 |
|  | BGRL | 71.06 ± 2.84 | 80.56 ± 1.59 | 68.75 ± 3.69 | 69.23 ± 0.19 | 55.66 ± 2.00 | 67.62 ± 0.19 | 68.81 |
|  | GraphMAE | 76.34 ± 1.49 | 79.19 ± 1.32 | 73.88 ± 1.16 | 70.46 ± 0.04 | 71.18 ± 0.13 | 71.82 ± 0.05 | 73.81 |
|  | OFA | 70.63 ± 1.03 | 79.13 ± 2.53 | 70.95 ± 1.02 | 70.43 ± 0.12 | 70.67 ± 0.21 | 71.28 ± 0.24 | 72.18 |
|  | GIT - G | 78.74 ± 1.12 | 81.03 ± 0.78 | 75.26 ± 2.81 | **72.49 ± 0.07** | **74.42 ± 0.15** | 72.99 ± 0.10 | 75.82 |
|  | GIT - S | **78.90 ± 1.44** | **81.97 ± 0.80** | **76.17 ± 1.70** | 71.50 ± 0.08 | 73.59 ± 0.08 | **73.13 ± 0.11** | **75.88** |

Table 15: Link prediction results on academic networks in terms of AUC.

|  | Cora | Citeseer | Pubmed | Arxiv23 | DBLP | Arxiv | **Avg.** |
|---|---|---|---|---|---|---|---|
| GCN | 87.34 ± 0.88 | 87.52 ± 0.98 | 84.41 ± 0.17 | 89.67 ± 0.24 | **98.29 ± 0.07** | **97.50 ± 0.08** | 90.79 |
| BGRL | 83.96 ± 0.36 | 81.51 ± 0.85 | 84.01 ± 0.60 | 86.42 ± 0.08 | 97.24 ± 0.06 | 96.80 ± 0.04 | 88.32 |
| GraphMAE | 85.57 ± 0.27 | 84.55 ± 0.69 | **89.83 ± 0.35** | 91.45 ± 0.44 | 98.05 ± 0.06 | 96.31 ± 0.02 | 90.96 |
| OFA | 82.82 ± 0.72 | 81.52 ± 1.16 | 84.78 ± 1.08 | 85.40 ± 0.62 | 97.23 ± 0.14 | 96.46 ± 0.05 | 88.04 |
| GIT - G | 87.79 ± 2.07 | 87.59 ± 0.96 | 84.35 ± 0.26 | 91.47 ± 0.46 | 98.25 ± 0.09 | 97.14 ± 0.06 | 91.10 |
| GIT - S | **88.58 ± 1.88** | **88.50 ± 1.15** | 87.78 ± 0.13 | **91.86 ± 0.38** | 98.27 ± 0.05 | 97.30 ± 0.05 | **92.05** |

Table 16: The few-shot performance on Arxiv, comparing to few-shot learning methods.

|  | **5-way** |  |  | **3-way** |  |  |
|---|---|---|---|---|---|---|
|  | 5-shot | 3-shot | 1-shot | 5-shot | 3-shot | 1-shot |
| GPN | 50.53 ± 3.07 | 48.32 ± 3.80 | 38.58 ± 1.61 | 62.25 ± 4.94 | 58.52 ± 3.00 | 48.45 ± 5.60 |
| TENT | 60.83 ± 7.45 | 56.03 ± 8.90 | 45.62 ± 10.70 | 74.20 ± 9.93 | 70.48 ± 11.50 | 59.38 ± 13.55 |
| GLITTER | 56.00 ± 4.40 | 57.44 ± 4.90 | 47.12 ± 2.73 | 62.13 ± 10.85 | 60.93 ± 12.12 | 59.20 ± 5.48 |
| TLP-BGRL | 50.13 ± 8.78 | 46.21 ± 7.92 | 35.81 ± 8.58 | 62.93 ± 11.74 | 58.37 ± 11.34 | 46.30 ± 10.83 |
| TLP-SURGL | 77.89 ± 6.46 | 74.19 ± 7.55 | 61.75 ± 10.07 | 86.27 ± 7.54 | 83.75 ± 8.86 | 73.46 ± 12.68 |
| Prodigy | 61.09 ± 5.85 | 58.64 ± 5.84 | 48.23 ± 6.18 | 73.64 ± 6.93 | 71.43 ± 7.28 | 61.59 ± 8.53 |
| GIT - G | 70.50 ± 0.47 | **64.07 ± 0.50** | 49.18 ± 0.56 | 80.20 ± 0.67 | 74.65 ± 0.54 | 61.93 ± 0.18 |
| GIT - S | **70.70 ± 0.28** | 64.06 ± 0.58 | **50.94 ± 0.57** | **80.51 ± 0.68** | **76.05 ± 0.53** | **63.42 ± 0.46** |

**Comparing to Few-shot Learning Methods.** We compare the performance of GIT with methods designed for few-shot learning on Arxiv in Table 16. It is important to note that we implement an in-context version of few-shot learning without fine-tuning GIT on the few-shot task, whereas the other methods require fine-tuning. Despite this, GIT achieves the second-best performance across all settings, outperforming 5 out of 6 few-shot learning methods. This highlights GIT's strong potential in scenarios with limited instances and labels.

**Ablation Study on Training Strategy.** We perform an ablation study on academic networks to assess the impact of different training strategies. Specifically, we evaluate four strategies: (1) base model: pretraining and fine-tuning on the same graph, (2) domain expert: pretraining on all academic networks, (3) general model: pretraining on the default datasets used in this paper, and (4) specialized model: pretraining on the default datasets followed by specialization on the academic domain. The results, averaged across all academic graphs, are presented in Table 17. We make the following observations: (1) When pretrained on cross-domain and cross-task datasets (general model), GIT achieves the best performance, highlighting the advantages of using task-trees as the

Table 17: Ablation study on the academic domain, evaluating different training strategies. We report the average performance across all academic graphs. The **base model** is pretrained on the target graph, the **domain expert** is pretrained on all academic graphs, the **general model** is pretrained on the default pretraining datasets, and the **specialized model** is fine-tuned via instruction tuning on the `Arxiv` dataset, based on the general model.

| | **Base Model** | | | **Domain Expert** | | |
|---|---|---|---|---|---|---|
| | 0-shot | 3-shot | Finetune | 0-shot | 3-shot | Finetune |
| GraphMAE | 15.30 | 51.51 | **75.57** | 17.89 | **55.88** | 75.26 |
| OFA | 14.19 | 50.15 | 75.12 | 17.26 | 54.88 | **75.97** |
| GIT | **15.36** | **53.31** | 75.53 | **18.38** | 55.10 | 75.47 |
| | **General Model** | | | **Specialized Model** | | |
| | 0-shot | 3-shot | Finetune | 0-shot | 3-shot | Finetune |
| GraphMAE | **15.42** | 49.25 | 73.81 | 20.31 | 51.21 | 74.05 |
| OFA | 13.98 | 45.93 | 72.18 | 20.05 | 46.87 | 73.04 |
| GIT | 14.88 | **54.00** | **75.82** | **23.45** | **55.18** | **75.88** |

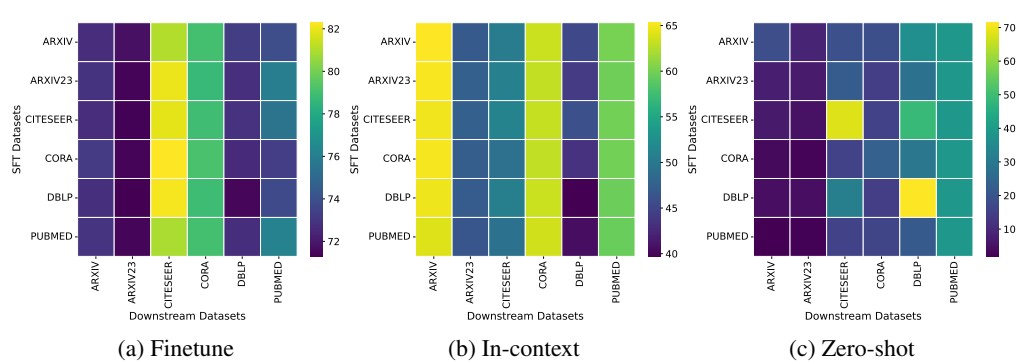

(a) Finetune        (b) In-context        (c) Zero-shot

Figure 12: The impact of different SFT datasets used for specialization in academic networks.

basic learning instance. However, the model does not consistently outperform in the base model and domain expert settings, possibly due to its model simplicity. (2) The general version of GIT maintains consistent performance compared to the base and expert models, while GraphMAE and OFA show a performance drop when transitioning from the base and expert models to the general model. This demonstrates the potential of task-trees in mitigating negative transfer in cross-domain and cross-task settings. (3) Interestingly, specialization enhances the performance of all baselines, showcasing the potential of instruction tuning to improve model capabilities in specific scenarios.

**Ablation Study on SFT Data used for Specialization.** We analyze the impact of SFT data in the experiments, as shown in Figure 12. The results show that changes in SFT data do not significantly affect model performance, particularly in fine-tuning and in-context learning settings. Even when the SFT and downstream data are the same, the model does not necessarily outperform models fine-tuned on other SFT datasets. This observation supports the motivation behind our proposed specialization method, which aims to shift the pretraining distribution $\mathcal{P}$ toward the distribution of target domains. It also highlights the importance of designing an instance selection method to identify the most effective SFT data.

### F.3 DOMAIN: E-COMMERCE

**Node Classification.** The comprehensive node classification results on e-commerce datasets are presented in Table 18. Our proposed GIT model outperforms the baselines in most settings, particularly for the specialized version. Specialization significantly improves performance in zero-shot and in-context learning, highlighting the advantages of using task-trees as the basic learning instances. In the basic fine-tuning setting, we also observe that supervised methods (GCN and GAT) generally

Table 18: Node classification results on e-commerce networks in terms of accuracy.

| | | History | Children | Computer | Photo | Sportsfit | Ratings | Products | **Avg.** |
|---|---|---|---|---|---|---|---|---|---|
| **0-shot** | GCN | - | - | - | - | - | - | - | - |
| | GAT | - | - | - | - | - | - | - | - |
| | BGRL | 6.76 ± 0.18 | 4.26 ± 0.14 | 9.70 ± 0.39 | 6.32 ± 0.20 | 7.91 ± 0.31 | 17.50 ± 0.65 | 0.58 ± 0.19 | 7.58 |
| | GraphMAE | 9.20 ± 0.23 | 4.25 ± 0.13 | 7.86 ± 0.23 | 8.02 ± 0.40 | 7.70 ± 0.34 | 20.26 ± 0.16 | 0.07 ± 0.03 | 8.19 |
| | OFA | 8.84 ± 0.52 | 4.22 ± 0.19 | 10.83 ± 0.32 | 8.46 ± 0.34 | 7.28 ± 0.52 | 18.43 ± 0.50 | 3.02 ± 0.37 | 8.73 |
| | GIT - G | 4.72 ± 0.31 | 4.34 ± 0.19 | 8.85 ± 0.30 | **11.78 ± 0.26** | 7.20 ± 0.18 | 21.00 ± 0.06 | 3.64 ± 0.25 | 8.79 |
| | GIT - S | **9.94 ± 0.54** | **7.49 ± 0.12** | **14.87 ± 0.40** | 9.69 ± 0.22 | **9.23 ± 0.65** | **21.55 ± 0.31** | **46.62 ± 1.06** | **17.06** |
| **3-shot** | GCN | - | - | - | - | - | - | - | - |
| | GAT | - | - | - | - | - | - | - | - |
| | BGRL | 38.35 ± 0.51 | 32.93 ± 0.75 | 50.90 ± 0.82 | 61.64 ± 0.51 | 42.99 ± 0.48 | **21.67 ± 0.21** | 71.71 ± 0.23 | 45.74 |
| | GraphMAE | 42.28 ± 0.38 | 38.71 ± 0.49 | 58.24 ± 0.79 | 59.47 ± 0.25 | 46.57 ± 0.46 | 21.11 ± 0.56 | 71.01 ± 0.67 | 48.20 |
| | OFA | 48.87 ± 0.26 | 47.13 ± 0.32 | 68.14 ± 0.49 | 75.73 ± 0.24 | 63.56 ± 0.57 | 21.38 ± 0.16 | 74.58 ± 0.33 | 57.06 |
| | GIT - G | 50.78 ± 0.41 | 47.55 ± 0.26 | 66.64 ± 0.50 | 75.43 ± 0.26 | 64.56 ± 0.43 | 21.21 ± 0.37 | 74.35 ± 0.48 | 57.22 |
| | GIT - S | **50.99 ± 0.64** | **47.65 ± 0.36** | **69.29 ± 0.48** | **76.32 ± 0.55** | **65.84 ± 0.53** | 21.17 ± 0.40 | **74.80 ± 0.54** | **58.01** |
| **Finetune** | GCN | 84.62 ± 0.06 | 58.08 ± 0.08 | 88.41 ± 0.06 | 86.39 ± 0.11 | 92.07 ± 0.02 | 50.99 ± 0.23 | 86.91 ± 0.05 | 78.21 |
| | GAT | 84.54 ± 0.07 | 59.09 ± 0.05 | **89.00 ± 0.04** | **86.70 ± 0.07** | 91.12 ± 0.05 | 51.19 ± 0.15 | 87.22 ± 0.05 | 78.41 |
| | GraphMAE | 82.51 ± 0.05 | 56.76 ± 0.09 | 84.31 ± 0.06 | 83.26 ± 0.06 | 90.47 ± 0.03 | 52.39 ± 0.29 | 86.30 ± 0.07 | 76.57 |
| | OFA | 82.81 ± 0.11 | 55.43 ± 0.08 | 85.78 ± 0.13 | 83.21 ± 0.25 | 91.23 ± 0.07 | 51.79 ± 0.18 | 86.23 ± 0.07 | 76.64 |
| | GIT - G | 84.94 ± 0.10 | 59.09 ± 0.15 | 87.81 ± 0.10 | 85.66 ± 0.06 | 92.17 ± 0.06 | 52.45 ± 0.26 | 87.75 ± 0.04 | 78.61 |
| | GIT - S | **85.18 ± 0.11** | **59.73 ± 0.12** | 88.05 ± 0.18 | 85.66 ± 0.05 | **92.44 ± 0.02** | **52.56 ± 0.29** | **88.20 ± 0.05** | **78.83** |

Table 19: Link prediction results on e-commerce networks in terms of AUC.

| | History | Photo | Ratings | **Avg.** |
|---|---|---|---|---|
| GCN | **97.87 ± 0.06** | 97.37 ± 0.03 | 97.77 ± 0.07 | 97.67 |
| BGRL | 96.40 ± 0.08 | 97.58 ± 0.04 | 98.05 ± 0.04 | 97.34 |
| GraphMAE | 97.59 ± 0.06 | 98.09 ± 0.05 | 95.35 ± 0.15 | 97.01 |
| OFA | 95.86 ± 0.09 | 97.05 ± 0.06 | 97.79 ± 0.12 | 96.90 |
| GIT - G | 96.55 ± 0.07 | 96.24 ± 0.05 | 98.45 ± 0.07 | 97.08 |
| GIT - S | 97.08 ± 0.05 | **97.80 ± 0.06** | **98.49 ± 0.10** | **97.79** |

outperform self-supervised methods, such as GraphMAE (Hou et al., 2022) and OFA (Liu et al., 2024), indicating the occurrence of negative transfer. However, GIT surpasses these supervised methods on 5 out of 7 datasets, further demonstrating the benefits of task-trees as basic learning instances. It it important to note that we consider `Children` and `Ratings` as heterophily graphs (Chen et al., 2024b) due to their low homophily ratio.

**Link Prediction.** The link prediction results on e-commerce networks (`History`, `Photo`, `Ratings`) are presented in Table 19. We randomly select 70% of the edges for training, 15% for validation, and the remaining 15% for testing. Our GIT model achieves the best average performance across these three e-commerce graphs. However, other baselines like BGRL, GraphMAE, and OFA fail to outperform the basic GCN. This may be because they struggle to acquire useful knowledge during pretraining for tasks that require structural insight, such as link prediction. These results underscore the advantages of using task-trees as the basic learning instances.

### F.4 DOMAIN: KNOWLEDGE BASE

**Edge Classification.** The edge classification results on knowledge graphs are presented in Table 20. In this domain, our GIT model significantly outperforms the existing baselines, demonstrating the advantages of using task-trees as the basic learning instances for knowledge bases, even though these KGs represent different scenarios. We hypothesize that this improvement stems from the nature of relation triplets in KGs, where each relation inherently describes the aggregation of the head and tail nodes, aligning with the concept of task-trees.

**Comparison to Domain Experts.** In addition to comparing GIT to standard baselines applicable for all graphs, we also evaluate it against ULTRA, a foundation model specifically designed for knowledge graphs. As a domain expert, ULTRA is compared to Expert GIT (pretrained on all KGs) and Specialized GIT (pretrained on default datasets and fine-tuned on `FB15K237`), with the results presented in Table 21. We find that the two domain experts, ULTRA and Expert GIT, achieve comparable performance, though ULTRA significantly outperforms Expert GIT in certain settings.

Table 20: Edge classification results on knowledge graphs in terms of accuracy.

| | | WN18RR | Codex-S | Codex-M | Codex-L | NELL995 | GDELT | ICEWS1819 | FB15K237 | Avg. |
|---|---|---|---|---|---|---|---|---|---|---|
| **3-shot** | GCN | - | - | - | - | - | - | - | - | - |
| | GraphMAE | 55.20 ± 0.52 | 61.41 ± 0.86 | 54.30 ± 0.42 | 61.01 ± 0.55 | 86.42 ± 0.53 | 32.43 ± 0.48 | 31.58 ± 0.39 | 70.15 ± 0.75 | 56.56 |
| | OFA | 55.27 ± 0.64 | 55.14 ± 0.34 | 50.20 ± 0.68 | 62.40 ± 0.46 | 88.41 ± 0.38 | 30.23 ± 0.50 | 34.94 ± 0.32 | 79.15 ± 0.45 | 56.97 |
| | GIT - G | 55.80 ± 0.32 | 76.96 ± 0.43 | 73.79 ± 0.43 | 78.54 ± 0.51 | 89.13 ± 0.48 | 34.30 ± 0.68 | **42.07 ± 0.75** | 89.78 ± 0.46 | 67.55 |
| | GIT - S | **57.90 ± 0.97** | **77.19 ± 0.32** | 72.14 ± 0.84 | 76.99 ± 0.72 | **90.80 ± 0.51** | **34.85 ± 0.69** | 42.02 ± 0.65 | **90.49 ± 0.32** | **67.80** |
| **Finetune** | GCN | 86.77 ± 0.30 | 93.56 ± 2.11 | 85.73 ± 1.84 | 84.45 ± 0.18 | 79.06 ± 0.32 | 11.72 ± 0.05 | 27.53 ± 0.06 | 66.07 ± 0.26 | 66.86 |
| | GraphMAE | 93.87 ± 0.35 | 97.09 ± 0.72 | 94.07 ± 0.60 | 94.18 ± 0.19 | 86.10 ± 0.42 | 13.12 ± 0.04 | 28.91 ± 0.06 | 73.52 ± 0.12 | 72.61 |
| | OFA | 93.10 ± 0.31 | 90.78 ± 5.46 | 93.83 ± 3.28 | 93.26 ± 0.59 | 86.91 ± 1.50 | 14.48 ± 0.03 | 30.60 ± 0.63 | 76.08 ± 1.95 | 72.38 |
| | GIT - G | 94.16 ± 0.11 | 98.08 ± 0.08 | 97.89 ± 0.04 | **96.85 ± 0.03** | 90.10 ± 0.23 | 14.86 ± 0.12 | **33.49 ± 0.06** | **80.39 ± 0.13** | 75.73 |
| | GIT - S | **95.15 ± 0.07** | **99.19 ± 0.04** | **97.92 ± 0.04** | 96.83 ± 0.04 | **91.28 ± 0.41** | **14.89 ± 0.05** | 33.61 ± 0.10 | 80.32 ± 0.07 | **76.15** |

Table 21: Comparison between GIT and ULTRA, a foundation model designed for knowledge graphs. The Expert GIT is pretrained on all KGs used in the paper.

| | **3-shot** | | | **Finetune** | | |
|---|---|---|---|---|---|---|
| | ULTRA | Expert GIT | Specialized GIT | ULTRA | Expert GIT | Specialized GIT |
| WN18RR | **60.69 ± 0.82** | 55.83 ± 0.44 | 57.90 ± 0.97 | **96.35 ± 0.22** | 95.12 ± 0.05 | 95.15 ± 0.07 |
| Codex-S | **82.45 ± 0.53** | 76.07 ± 0.41 | 77.19 ± 0.32 | 98.27 ± 0.36 | 99.14 ± 0.07 | **99.19 ± 0.04** |
| Codex-M | **74.35 ± 0.23** | 73.54 ± 0.46 | 72.14 ± 0.84 | 96.90 ± 0.11 | 97.90 ± 0.06 | **97.92 ± 0.04** |
| Codex-L | 75.98 ± 0.48 | **78.13 ± 0.36** | 76.99 ± 0.72 | 96.22 ± 0.04 | **96.84 ± 0.04** | 96.83 ± 0.04 |
| NELL995 | 90.22 ± 0.46 | 89.99 ± 0.24 | **90.80 ± 0.51** | 89.46 ± 0.28 | 90.55 ± 0.59 | **91.28 ± 0.41** |
| GDELT | 33.89 ± 0.33 | **34.92 ± 0.55** | 34.85 ± 0.69 | 14.63 ± 0.02 | **14.91 ± 0.10** | 14.89 ± 0.05 |
| ICEWS1819 | 41.37 ± 0.53 | **42.42 ± 0.64** | 42.02 ± 0.65 | **35.95 ± 0.03** | 33.62 ± 0.13 | 33.61 ± 0.10 |
| FB15K237 | 89.29 ± 0.40 | **90.83 ± 0.30** | 90.49 ± 0.32 | **82.28 ± 0.08** | 80.18 ± 0.29 | 80.32 ± 0.07 |
| **Average** | **68.53** | 67.72 | 67.80 | **76.26** | 76.03 | 76.15 |

Table 22: Graph classification results on molecule graphs in terms of AUC.

| | | HIV | BBBP | BACE | TOXCAST | CYP450 | TOX21 | MUV | PCBA | Avg. |
|---|---|---|---|---|---|---|---|---|---|---|
| **0-shot** | GIN | - | - | - | - | - | - | - | - | - |
| | BGRL | 55.27 | 53.72 | 33.74 | 49.00 | 60.99 | 46.40 | 39.90 | 42.39 | 47.68 |
| | GraphMAE | 46.48 | 49.08 | 30.76 | 48.22 | 60.55 | 49.17 | 48.17 | 45.10 | 47.19 |
| | OFA | 47.96 | 50.61 | 34.35 | 49.70 | 61.96 | 52.73 | 52.48 | 54.14 | 50.49 |
| | GIT - G | 56.76 | 54.76 | 33.66 | 51.55 | 63.21 | 56.83 | 53.71 | 56.25 | 53.34 |
| | GIT - S | **66.14** | **62.16** | **52.27** | **58.30** | **69.75** | **63.45** | **65.32** | **65.26** | **62.83** |
| **3-shot** | GIN | - | - | - | - | - | - | - | - | - |
| | BGRL | 52.72 ± 1.84 | 49.12 ± 0.78 | 59.58 ± 0.89 | **57.27 ± 0.05** | 67.49 ± 0.56 | 59.26 ± 0.19 | 52.61 ± 0.23 | 51.48 ± 0.22 | 56.19 |
| | GraphMAE | 54.40 ± 1.04 | 48.41 ± 1.34 | 60.78 ± 1.01 | 56.99 ± 0.06 | 66.93 ± 0.91 | 58.40 ± 0.22 | 51.95 ± 0.18 | 50.24 ± 0.23 | 56.01 |
| | OFA | **56.04 ± 1.49** | 50.69 ± 1.36 | 60.21 ± 0.64 | 56.40 ± 0.05 | 68.76 ± 0.16 | 57.18 ± 0.29 | 56.17 ± 0.23 | 50.77 ± 0.30 | 57.03 |
| | GIT - G | 52.42 ± 1.74 | 48.22 ± 1.14 | 59.32 ± 0.91 | 56.32 ± 0.04 | 66.77 ± 0.45 | 58.53 ± 0.36 | 55.98 ± 0.19 | 50.09 ± 0.30 | 55.96 |
| | GIT - S | 54.12 ± 1.66 | **66.74 ± 1.34** | **61.76 ± 0.92** | 55.53 ± 0.03 | **81.50 ± 0.23** | **65.16 ± 0.27** | **66.14 ± 0.30** | **51.58 ± 0.30** | **62.82** |
| **Finetune** | GIN | **76.83 ± 1.32** | 67.36 ± 1.39 | 75.55 ± 2.91 | 62.92 ± 0.42 | 85.82 ± 0.77 | 72.26 ± 0.24 | 70.12 ± 0.39 | 78.34 ± 0.51 | 73.65 |
| | BGRL | 72.18 ± 1.24 | 67.40 ± 1.45 | 73.75 ± 3.69 | 62.52 ± 0.10 | 83.10 ± 0.26 | 72.97 ± 0.54 | 68.46 ± 0.63 | 76.69 ± 1.40 | 72.13 |
| | GraphMAE | 69.54 ± 2.59 | 66.43 ± 2.48 | 66.56 ± 4.73 | 62.52 ± 0.14 | 86.64 ± 0.27 | **74.13 ± 0.41** | 70.12 ± 0.40 | 75.34 ± 1.33 | 71.41 |
| | OFA | 76.48 ± 2.11 | 65.79 ± 0.96 | 77.88 ± 1.08 | 63.49 ± 0.61 | 85.77 ± 0.32 | 73.00 ± 0.67 | 69.53 ± 0.56 | 80.31 ± 1.20 | 74.03 |
| | GIT - G | 73.63 ± 0.77 | 68.33 ± 1.06 | 79.28 ± 2.71 | 63.00 ± 0.43 | 86.86 ± 0.22 | 73.81 ± 0.33 | 70.49 ± 0.51 | 81.13 ± 0.53 | 74.57 |
| | GIT - S | 74.75 ± 0.42 | **68.72 ± 1.13** | **81.10 ± 0.61** | **63.63 ± 0.61** | **87.00 ± 0.37** | 73.78 ± 0.77 | **71.16 ± 0.51** | **81.43 ± 0.34** | **75.20** |

This may be due to ULTRA learning more fine-grained relational information within KGs. Notably, Specialized GIT also performs comparably to both domain experts, highlighting the potential of specialization. We believe this is because the distributions of KGs are more similar to each other compared to graphs from other domains.

## F.5 DOMAIN: MOLECULE

**Experimental Settings.** We evaluate fine-tuning, in-context learning, and zero-shot learning in this domain. The fine-tuning and in-context learning settings are consistent with those used in previous domains. For zero-shot learning, however, we follow the approach of Zhao et al. (2023) by assessing zero-shot performance on the original test set.

**Graph Classification.** The graph classification results are presented in Table 22. Our GIT model achieves the best average performance across the three evaluated settings. We also observe that spe-

Table 23: Comparison between our GIT and domain experts of molecule graphs in zero-shot setting.

| | HIV | BBBP | BACE | TOXCAST | CYP450 | TOX21 | MUV | PCBA | **Avg.** |
|---|---|---|---|---|---|---|---|---|---|
| KVPLM* | 61.20 | 60.20 | 51.26 | 50.96 | 59.22 | 49.17 | 61.72 | 48.11 | 55.23 |
| MoMu* | 50.26 | 49.81 | 66.56 | 52.38 | 57.98 | 57.57 | 60.51 | 51.50 | 55.82 |
| Galactica-1.3B* | 33.85 | 53.94 | 56.48 | 51.23 | 46.86 | 49.46 | 57.15 | 52.02 | 50.12 |
| GIMLET* | **66.24** | **59.39** | **69.57** | **59.04** | **71.25** | **61.19** | **64.39** | **62.11** | **64.15** |
| GIT - G | 56.76 | 54.76 | 33.66 | 51.55 | 63.21 | 56.83 | 53.71 | 56.25 | 53.34 |
| GIT - S | **66.14** | **62.16** | **52.27** | **58.30** | **69.75** | **63.45** | **65.32** | **65.26** | **62.83** |

* indicates the results from paper (Zhao et al., 2023).

Table 24: The impact of SFT datasets in zero-shot setting.

| **SFT Data** | HIV | BBBP | BACE | TOXCAST | CYP450 | TOX21 | MUV | PCBA | **Avg.** |
|---|---|---|---|---|---|---|---|---|---|
| PCBA | 66.14 | 62.16 | 52.27 | 58.30 | 69.75 | 63.45 | 65.32 | 65.26 | 62.83 |
| HIV | 66.28 | 45.97 | 43.35 | 52.78 | 64.50 | 57.86 | 53.46 | 46.57 | 53.85 |

Table 25: Edge classification results on temporal graph Enron.

| | | Enron 1 | Enron 2 | Enron 3 | Enron 4 | Enron 5 | Enron 6 | Enron 7 | Enron 8 | Enron 9 | Enron 10 | Avg. |
|---|---|---|---|---|---|---|---|---|---|---|---|---|
| **Finetune** | GAT | 81.36 ± 0.08 | 60.60 ± 0.88 | 62.40 ± 1.83 | 83.49 ± 0.25 | 45.88 ± 0.34 | 65.97 ± 1.07 | 48.14 ± 0.23 | 59.15 ± 0.65 | 82.39 ± 1.98 | 45.35 ± 0.43 | 63.47 |
| | GraphMAE | 81.29 ± 0.01 | 59.52 ± 0.10 | 66.13 ± 1.42 | 82.84 ± 0.67 | 50.01 ± 0.34 | 64.46 ± 0.75 | 45.16 ± 0.15 | 67.25 ± 0.21 | 72.05 ± 3.27 | 48.00 ± 0.01 | 63.67 |
| | GIT - G | **81.48 ± 0.28** | 61.25 ± 0.25 | 67.56 ± 2.16 | 84.50 ± 0.21 | **52.52 ± 0.80** | **67.69 ± 0.54** | **50.32 ± 0.17** | 68.35 ± 0.51 | 76.92 ± 1.16 | 48.28 ± 0.15 | 65.89 |
| | GIT - S | 81.27 ± 0.12 | **61.42 ± 0.12** | **69.15 ± 0.43** | **84.51 ± 0.17** | 51.93 ± 0.48 | 66.74 ± 1.24 | 50.12 ± 0.32 | **68.89 ± 0.64** | 77.03 ± 2.05 | **48.35 ± 0.02** | **65.94** |
| **3-shot** | GAT | - | - | - | - | - | - | - | - | - | - | - |
| | OFA | 68.91 ± 0.31 | 58.27 ± 0.41 | **62.43 ± 0.60** | 55.48 ± 0.59 | 61.46 ± 0.22 | 50.35 ± 0.75 | 53.44 ± 0.37 | 49.01 ± 0.55 | 56.43 ± 0.70 | 59.01 ± 0.19 | 57.48 |
| | GraphMAE | 73.23 ± 0.76 | 58.53 ± 0.66 | 61.66 ± 0.61 | 58.15 ± 0.52 | 59.81 ± 0.50 | **50.59 ± 0.60** | **56.89 ± 0.74** | **56.08 ± 0.45** | 59.69 ± 0.44 | 63.63 ± 0.62 | 59.83 |
| | GIT - G | 71.67 ± 0.43 | **60.31 ± 0.49** | 61.46 ± 0.59 | 57.62 ± 0.56 | 59.60 ± 0.93 | 50.82 ± 0.40 | 54.02 ± 0.58 | 52.22 ± 0.32 | 60.61 ± 0.29 | 62.17 ± 0.34 | 59.05 |
| | GIT - S | **73.73 ± 0.50** | 58.96 ± 0.43 | 60.08 ± 0.45 | **59.38 ± 0.56** | **61.84 ± 0.78** | 50.43 ± 0.72 | 54.92 ± 0.22 | 56.03 ± 0.43 | **61.99 ± 0.80** | **64.85 ± 0.50** | **60.22** |

cialization consistently improves performance across different graphs, aligning with our theoretical analysis.

**Comparison to Domain Experts.** In addition to general GNN baselines applicable across various graphs, we compare our GIT model to domain experts specifically designed for molecules, including KVPLM (Zeng et al., 2022), MoMu (Su et al., 2022), Galactica (Taylor et al., 2022), and the recent SOTA model, GIMLET (Zhao et al., 2023). The results are presented in Table 23. We find that the general model pretrained on large-scale graphs generally underperforms compared to these domain experts. However, after specialization, the specialized model surpasses 3 out of 4 domain experts on average and outperforms the best expert model, GIMLET, on 4 out of 8 datasets. This observation demonstrates that post-training the general model with a reasonable number of domain-specific instances can enable it to match or even surpass expert models designed for that domain. These results strongly support the effectiveness of task-trees in designing graph foundation models.

**Ablation Study on SFT Data used for Specialization.** We also evaluate the impact of the SFT dataset used for specialization. The model's zero-shot performance is reported in Table 24, comparing the default SFT dataset PCBA with another SFT dataset, HIV. We find that the model performance with HIV as the SFT data is lower than with PCBA. We hypothesize that this is due to HIV having fewer graphs and tasks, which may provide less information for reducing the distribution discrepancy. Nevertheless, HIV still improves the model's performance over the general model on 5 out of 8 datasets.

## F.6 DOMAIN: TEMPORAL E-COMMERCE

**Edge Classification.** We report the experimental results on two temporal graphs, Enron and Googlemap CT, in Table 25 and Table 26, respectively. The original graph is split into ten snapshots based on timestamps, and the model performance is evaluated separately on each snapshot. Since we fine-tuned the pretrained model on Products, these experiments assess the model's robustness to temporal distribution shifts. The results demonstrate GIT's capability to effectively handle temporal information in graphs.

Table 26: Edge classification results on temporal graph `Googlemap CT`.

| | | GCT 1 | GCT 2 | GCT 3 | GCT 4 | GCT 5 | GCT 6 | GCT 7 | GCT 8 | GCT 9 | GCT 10 | **Avg.** |
|---|---|---|---|---|---|---|---|---|---|---|---|---|
| **Finetune** | GAT | 61.29 ± 0.04 | 56.29 ± 0.03 | 56.13 ± 0.08 | 57.32 ± 0.06 | 60.12 ± 0.08 | 61.65 ± 0.13 | 63.37 ± 0.06 | 64.71 ± 0.06 | 67.08 ± 0.06 | 69.46 ± 0.04 | 61.74 |
| | GraphMAE | 64.60 ± 0.42 | 57.61 ± 0.32 | 55.63 ± 0.24 | 57.08 ± 0.25 | 60.36 ± 0.19 | 60.99 ± 0.08 | 62.90 ± 0.06 | 63.83 ± 0.09 | 66.89 ± 0.12 | 68.35 ± 0.06 | 61.82 |
| | GIT - G | 64.21 ± 1.10 | 59.06 ± 0.20 | **57.12 ± 0.23** | **59.85 ± 0.20** | 61.92 ± 0.11 | **62.91 ± 0.10** | 64.02 ± 0.04 | **65.62 ± 0.14** | **67.66 ± 0.11** | 70.51 ± 0.10 | 63.29 |
| | GIT - S | **66.52 ± 0.29** | **58.63 ± 0.69** | 56.82 ± 0.23 | 59.77 ± 0.46 | **61.93 ± 0.22** | 62.72 ± 0.18 | **64.08 ± 0.07** | 65.56 ± 0.11 | 67.49 ± 0.18 | **70.62 ± 0.11** | **63.41** |
| **3-shot** | GAT | - | - | - | - | - | - | - | - | - | - | - |
| | OFA | 20.62 ± 0.34 | 21.22 ± 0.56 | 20.10 ± 0.32 | 20.16 ± 0.21 | 20.25 ± 0.36 | 20.39 ± 0.50 | 20.13 ± 0.14 | 20.21 ± 0.25 | 19.90 ± 0.30 | 20.59 ± 0.22 | 20.36 |
| | GraphMAE | 21.15 ± 0.44 | 21.03 ± 0.39 | 21.73 ± 0.23 | 21.60 ± 0.53 | 19.73 ± 0.41 | 20.38 ± 0.28 | 20.62 ± 0.22 | 20.51 ± 0.27 | 19.63 ± 0.43 | 21.38 ± 0.35 | 20.78 |
| | GIT - G | 21.81 ± 0.29 | 21.94 ± 0.23 | 20.78 ± 0.31 | 20.61 ± 0.40 | 20.73 ± 0.37 | 20.33 ± 0.56 | 20.90 ± 0.32 | 20.57 ± 0.46 | 20.58 ± 0.33 | 20.28 ± 0.31 | 20.85 |
| | GIT - S | **25.21 ± 0.53** | **24.21 ± 0.43** | **23.43 ± 0.44** | **22.41 ± 0.14** | **21.83 ± 0.59** | **21.65 ± 0.33** | **21.41 ± 0.50** | **21.76 ± 0.57** | **21.72 ± 0.22** | **21.72 ± 0.50** | **22.54** |

Table 27: The in-context learning performance of GIT with different specialization datasets (SFT Data) on four domains. The results of each domain is the average of all datasets within the domain.

| | SFT Data | Academia | E-commerce | KG | Molecule |
|---|---|---|---|---|---|
| **General Model** | - | 54.00 | 57.22 | 67.55 | 55.96 |
| **Specialized Model** | `Arxiv` (academia) | **55.18** | 57.63 | 66.80 | 54.42 |
| | `Products` (E-com) | 50.09 | **58.01** | 65.06 | 55.75 |
| | `FB15K237` (KG) | 54.70 | 56.57 | **67.80** | 55.37 |
| | `PCBA` (Mol) | 50.49 | 56.87 | 61.49 | **62.82** |

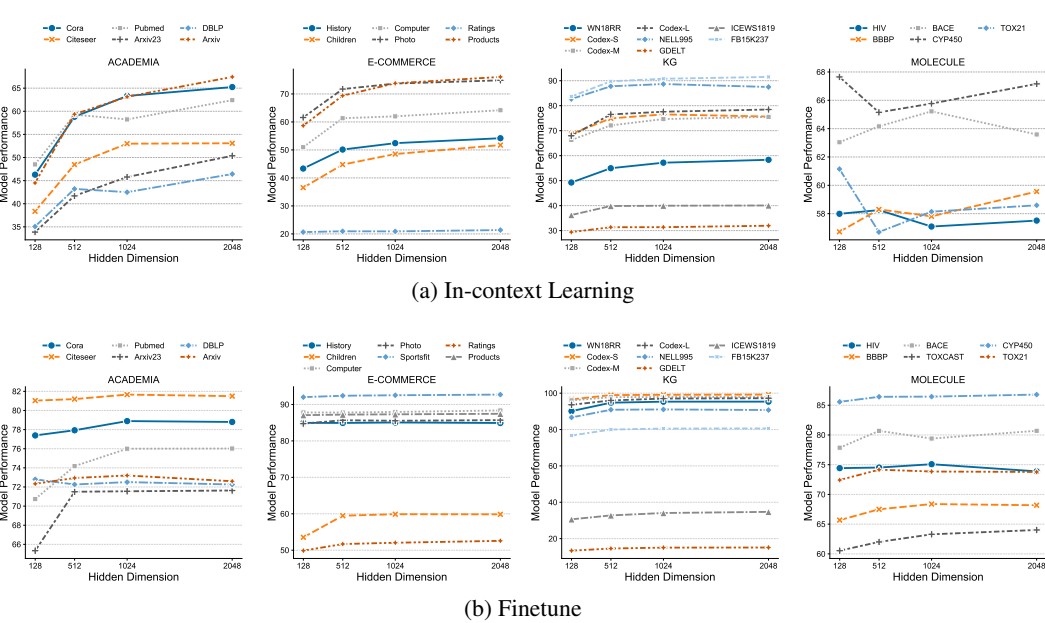

(a) In-context Learning

(b) Finetune

Figure 13: The comprehensive results of the impact of hidden dimensions on model performance.

## F.7 GENERAL REASONING CAPABILITY OF SPECIALIZED MODELS

We further analyze the performance of specialized models on general reasoning tasks beyond their specific domains. We assess the model's performance on other domains as a measure of its general reasoning ability. For example, if a model is specialized for academic networks, its general reasoning capability refers to its performance on graphs from other domains, such as e-commerce networks, knowledge graphs, and molecular graphs. The results are presented in Table 27. We report in-context learning performance rather than basic fine-tuning due to computational efficiency. Additionally, we include the performance of the pretrained general model without specialization as a baseline. If the specialized model performs worse than the general model, it suggests that specialization may diminish GIT's general reasoning capability. From the table, it is clear that while specialized models excel in their specific domains, they struggle in other domains. This degradation of general inference capability, often referred to as the *specialization tax*, is a common challenge in building specialized large language models. The specialization tax can limit the model's practicality in scenarios requiring both domain-specific knowledge and the ability to handle general tasks.

Thus, balancing domain-specific performance with maintaining general reasoning capability is an important research direction.

## F.8 More Parameters Enhance Model Performance

We present comprehensive results of general GIT with different hidden dimensions in Figure 13. For computational efficiency, we does not report results on datasets needing intensive computing resources. We observe that increasing the number of model parameters consistently improves performance across both basic fine-tuning and in-context learning settings. Notably, the performance improvement is more pronounced in in-context learning as the model size increases. This may be because, in in-context learning, the model is not fine-tuned on downstream tasks, making the knowledge retained in the original model more crucial. Larger hidden dimensions allow the model to preserve more knowledge. However, when the model is fine-tuned on downstream tasks, the pre-training knowledge is adapted to the specific task, reducing the reliance on the original model's knowledge and leading to a relatively smaller performance gap.

