# OpenReview forum: "Towards Graph Foundation Models: Learning Generalities Across Graphs via Task-trees"
_ICLR.cc/2025/Conference — Submitted to ICLR 2025_

### Official Review · Reviewer_BMnX · 2024-10-27

**Soundness:** 3
**Presentation:** 4
**Contribution:** 2
**Rating:** 3
**Confidence:** 4

**Summary:**

The paper centers on learning shared structures, or “generalities,” across various graphs using a unique construct called "task-trees." The proposed model, Graph Generality Identifier on task-Trees (GIT), is designed for broad applicability across graph-related tasks, such as node, edge, and graph classification.

**Strengths:**

1. The presentation is generally clear.
2. Both theoretical and experimental sections are well-executed, with experiments spanning a range of graph-based tasks across multiple domains.

**Weaknesses:**

1. My primary question is: what exactly differentiates the proposed Task-Tree Space from existing concepts like prompt graphs, contextual graphs, anchor nodes, or nodes of interest[1][2][3][4]? We already know that subgraphs can be used to unify node-, edge-, and graph-level tasks, and this paper seems to be doing the same thing, just under a different name.
2. While task-trees provide a generalized approach, there is a potential trade-off in terms of performance on domain-specific tasks, as noted in the specialization experiments.
3. Could the authors clarify the computational impact of task-tree transformations, particularly for large-scale graphs?

[1]Sun, X., Cheng, H., Li, J., Liu, B., & Guan, J. (2023, August). All in one: Multi-task prompting for graph neural networks. In Proceedings of the 29th ACM SIGKDD Conference on Knowledge Discovery and Data Mining (pp. 2120-2131).

[2]Liu, H., Feng, J., Kong, L., Liang, N., Tao, D., Chen, Y., & Zhang, M. (2023). One for all: Towards training one graph model for all classification tasks. arXiv preprint arXiv:2310.00149.

[3]He Y, Hooi B. UniGraph: Learning a Cross-Domain Graph Foundation Model From Natural Language[J]. arXiv preprint arXiv:2402.13630, 2024.

[4]Huang, Q., Ren, H., Chen, P., Kržmanc, G., Zeng, D., Liang, P. S., & Leskovec, J. (2024). Prodigy: Enabling in-context learning over graphs. Advances in Neural Information Processing Systems, 36.

**Questions:**

See weaknesses.

---

> ### Author Response · Authors · 2024-11-17
> **Part 1**
>
> Dear Reviewer BMnX,
>
> We sincerely appreciate your detailed feedback and constructive criticism, which will help us improve our work. Below, we address your concerns in detail:
>
> > W1: Differentiation of Task-Tree Space from Existing Concepts
> >
>
> Thank you for highlighting this concern. We would like to clarify that all the referenced works [1,2,3,4] are fundamentally based on **subgraphs**, whereas our proposed approach leverages **task-trees**. We will provide a thorough discussion on the distinctions between subgraphs and task-trees. In summary, our task-tree approach is both more efficient and effective in unifying graph-related tasks and preserving generalities across different graphs.
>
> **The indicated methods are based on subgraphs:** Here is a more detailed breakdown of why these methods are subgraph-based:
>
> - **[1]**: This work uses graph prompt learning for handling different graph-related tasks by extracting subgraphs around task-relevant nodes (e.g., ego-graphs around nodes for node classification). GNNs are then applied to the induced subgraphs to obtain embeddings for predictions.
> - **[2]**: This method enables in-context learning for node classification using what it refers to as "data graphs". However, these are essentially k-hop ego-graphs extracted around each node, which are then integrated into a graph prompt learning framework to learn subgraph embeddings.
> - **[3]**: Building on [1] and [2], this work extends the framework to a cross-domain, cross-task graph foundation model. Here, it uses the concept of NOI (Nodes of Interest) graphs to unify graph tasks, but the process is fundamentally the same as extracting subgraphs.
> - **[4]**: This work builds upon the subgraph sampling methods of [1], employing k-hop ego-graphs and using techniques like personalized PageRank to select subgraphs more effectively.
>
> Following, we will present the detailed comparison between subgraphs and task-trees.
>
> **Subgraphs**: Some existing methods utilize subgraphs (typically k-hop subgraphs) as the basic learning instances [1,2,3]. For example, in node classification, these approaches extract ego-graphs around each node and assign the labels of the central nodes as labels for the induced subgraphs, transforming node-level classification into subgraph-level classification. This approach can also be adapted for link-level or graph-level tasks [1]. In general, these methods involve: (1) Extracting subgraphs around task-relevant nodes [1] and (2) Applying GNNs on each subgraph to obtain subgraph embeddings for predictions. However, these methods have two significant limitations: (1) **Efficiency Issue**: Extracting subgraphs incurs additional computational overhead, leading to increased time and memory usage due to the need to process and store these subgraphs. (2) **Learnability Issue**: The information preserved in subgraphs may not always be learnable by standard GNNs. Given the expressiveness limitations of message passing (bounded by the 1-WL test), these methods often struggle to capture essential substructures within subgraphs.
>
> **Task-Trees**: Compared to subgraphs, task-trees are both more efficient and learnable. The encoding of task-trees for a node/link/graph involves: (1) Adding virtual nodes to the original graphs and connecting them to task-relevant nodes. (2) Using GNNs over the augmented graph to encode the embeddings of these virtual nodes for predictions. For instance, in node classification, we first augment the original graph by adding virtual nodes connected to each node and perform GNNs on the new graph to learn virtual node embeddings for predictions. Thus, our method requires only the addition of nodes and edges to the original graph, making it **significantly more efficient than extracting and storing the subgraph**. Additionally, encoding task-trees is equivalent to directly encoding the virtual nodes via message passing, ensuring that the **information in task-trees is fully learnable by standard GNNs**. In conclusion, our task-tree approach offers both superior efficiency and learnability.

---

> > ### Author Response · Authors · 2024-11-17
> > **Part 2**
> >
> > **Efficiency Comparison (Section 5.4 of the Original Submission):** We conducted an empirical comparison between subgraphs and task-trees. For this, we implemented a subgraph version of our model, **GIT-SubG**, by replacing task-trees with subgraphs. Below is a comparison of time and memory usage (on a 48GB GPU) between GIT and GIT-SubG during pretraining (about 1,700,000 task-trees). To give a better understanding, using a batch size of 2048, **GraphMAE** requires 193 seconds per epoch with a memory allocation of 35%.
> >
> > | Batch Size | 512 | 1024 | 2048 | 4096 | 8192 | 16384 | 32768 |
> > | --- | --- | --- | --- | --- | --- | --- | --- |
> > | *GIT - Task-Tree* |  |  |  |  |  |  |  |
> > | Time / Epoch (s) | 208 | 180 | 176 | 172 | 172 | 163 | 162 |
> > | Memory Allocation (%) | 6 | 8 | 18 | 41 | 75 | 93 | 98 |
> > | *GIT - SubG* |  |  |  |  |  |  |  |
> > | Time / Epoch (s) | 280 | 243 | 234 | 223 | OOM | OOM | OOM |
> > | Memory Allocation (%) | 21 | 39 | 74 | 97 | OOM | OOM | OOM |
> >
> > **Effectiveness Comparison (Section 5.4 of the Original Submission):** We also evaluated model performance across three domains: academic networks (node classification), knowledge graphs (edge classification), and molecule graphs (graph classification). Additionally, we compared our model against another popular subgraph-based method, **OFA** [3], which employs graph prompt learning for downstream tasks. The results (averaged over all graphs in each domain) are as follows.
> >
> > |  | OFA | GIT - SubG | **GIT** |
> > | --- | --- | --- | --- |
> > | Academia (node classification) | 72.18 | 73.48 | **75.82** |
> > | KG (edge classification) | 72.38 | 73.59 | **75.73** |
> > | Molecule (graph classification) | 74.03 | 72.67 | **75.73** |
> >
> > > W2: Potential Trade-off in Domain-Specific Performance
> > >
> >
> > You are correct that a model pretrained across multiple domains may not achieve state-of-the-art (SOTA) performance compared to a model designed for a single domain. However, one of our key contributions is to demonstrate that **this trade-off can be effectively mitigated through a lightweight specialization process**. This process enables the pretrained model to adapt to a specific domain with minimal computational overhead. We have demonstrated the effectiveness of this specialization process in the original submission by comparing it to domain expert models.
> >
> > Moreover, we show that the **general model (GIT-G) still achieves competitive performance on various domains**. Below is a comparison between the general model (GIT-G) and the specialized model (GIT-S) across five domains, with each value representing the average performance over all graphs within the domain (Table 1 in the original submission):
> >
> > |  | **Academia** | **E-commerce** | **KG** | **Molecule** | **Temporal** |
> > | --- | --- | --- | --- | --- | --- |
> > | **GIT - G** | 75.82 | 78.55 | 75.73 | 74.57 | 64.59 |
> > | **GIT - S** | 75.88 | 78.83 | 76.15 | 75.2 | 64.68 |

---

> > > ### Author Response · Authors · 2024-11-17
> > > **Part 3**
> > >
> > > > W3: Computational Impact of Task-Tree Transformations for Large-Scale Graphs
> > > >
> > >
> > > We appreciate your concern regarding the computational efficiency of task-tree transformations, especially for large-scale graphs. Unlike subgraph-based methods, our approach does not require explicit extraction of task-trees. Instead, we use message passing directly on augmented graphs to learn the embedding of task-trees, **making it extremely efficient**. We provide some notations for the following time complexity analysis: given a graph dataset, N is the number of nodes, E is the number of edges, and G is the number of graphs.
> > >
> > > **Complexity analysis:**
> > >
> > > - For **node classification**, we associate a virtual node with each node in the graph and learn the embeddings of these virtual nodes using GNNs as the embeddings of node-induced task-trees.
> > >
> > >     Complexity: Adding N virtual nodes O(N); adding N edges to connect each virtual node to its corresponding original node O(N). The overall time complexity of O(N).
> > >
> > > - For **edge classification**, we associate a virtual node with each edge by connecting it to the start and end nodes, allowing GNNs to learn the embeddings of these virtual nodes, as the embeddings of edge-induced task-trees.
> > >
> > >     Complexity: Adding E virtual nodes O(E), adding 2E edges to connect each virtual node to edges O(2E). The overall time complexity O(E)
> > >
> > > - For **graph classification**, we similarly add a virtual node for each graph, connecting it to all nodes on the graph, and use GNNs to learn the embedding of these virtual nodes, as the embeddings as graph-induced task-trees.
> > >
> > >     Complexity: Adding G virtual nodes O(G), adding N edges to connect virtual nodes to all nodes on the graph O(N). As the number of nodes in the dataset is much more than the number of graphs, the overall complexity is O(N).
> > >
> > >
> > > **Empirical Efficiency**: This approach only requires adding nodes and edges to the original graph, regardless of its size, making it computationally efficient. An empirical efficiency analysis of task-trees can be found in the general response. When the batch size is set to 1024 (~1,700,000 task-trees in total), the training time consumption per epoch is around 180s.
> > >
> > > ---
> > >
> > > Thank you once again for your thoughtful feedback. We hope these clarifications address your concerns. Please feel free to reach out if you have further questions or require additional details.
> > >
> > > ---
> > >
> > > Reference:
> > >
> > > [1] All in one: Multi-task prompting for graph neural networks, KDD 23.
> > >
> > > [2] Prodigy: Enabling in-context learning over graphs, NeurIPS 23.
> > >
> > > [2] One for all: Towards training one graph model for all classification tasks, ICLR 24.
> > >
> > > [4] UniGraph: Learning a Cross-Domain Graph Foundation Model From Natural Language, Arxiv 24.

---

### Official Review · Reviewer_hTAG · 2024-10-30

**Soundness:** 3
**Presentation:** 4
**Contribution:** 3
**Rating:** 6
**Confidence:** 4

**Summary:**

The paper presents GIT, learning general patterns using task-trees. By pretraining on task-trees, GIT captures transferable knowledge, validated through experiments on 30 graphs across five domains. GIT demonstrates strong performance in fine-tuning, in-context, and zero-shot learning, often outperforming domain-specific models.

**Strengths:**

1. The paper is well-written and easy to follow.
2. The paper provides a theoretical analysis of the effectiveness of task-trees.

**Weaknesses:**

1. **Inaccurate Statements**: The authors make some inaccurate statements. For example, in Appendix A, they claim that "most of these approaches rely on subgraphs as the primary learning instances, which can result in inefficient training and reduced expressiveness, as discussed in the main paper." However, many foundation models [1,2] do not rely on subgraphs.
2. **Limited Novelty in Task Trees as Unified Instances**: The novelty of leveraging task trees as unified instances is limited. Building task trees based on task-relevant nodes as unified instances is quite similar to using subgraphs as instances [3,4]. Subgraphs are a versatile structure that can represent node-, edge-, and graph-level instances [3].
3. **Limitations in Building a Graph Foundation Model**: I believe the paper is far from building a graph foundation model. Although a unified instance is proposed to unify various tasks, the semantic and structural gaps between various domains [1,2] remain unaddressed. These are important challenges in building a foundation model.

[1] Zhao et al. "All in one and one for all: A simple yet effective method towards cross-domain graph pretraining." Proceedings of the 30th ACM SIGKDD Conference on Knowledge Discovery and Data Mining. 2024.\
[2] Yu et al. "Text-Free Multi-domain Graph Pre-training: Toward Graph Foundation Models." arXiv preprint arXiv:2405.13934 (2024).\
[3] Liu et al. Graphprompt: Unifying pre-training and downstream tasks for graph neural networks[C]//Proceedings of the ACM Web Conference 2023. 2023: 417-428.\
[4] Hao et al. Motif-based prompt learning for universal cross-domain recommendation[C]//Proceedings of the 17th ACM International Conference on Web Search and Data Mining. 2024: 257-265.

**Questions:**

See weaknesses.

---

> ### Author Response · Authors · 2024-11-17
> **Part 1**
>
> Dear Reviewer hTAG,
>
> We sincerely appreciate your thorough review and constructive feedback, which will help us enhance our work. Below, we address your concerns in detail:
>
> > W1: Inaccurate Statements
> >
>
> Thank you for pointing this out. We apologize for any confusion and would like to clarify the statement in question. In Appendix A, we stated: “… Other efforts to resolve feature heterogeneity include methods like singular vector decomposition (SVD) [1,2] and non-parametric encoders [4]. However, most of these approaches rely on subgraphs as the primary learning instances, which can result in inefficient training and reduced expressiveness, as discussed in the main paper.”
>
> Our intention was to convey that these methods [1,2,4] can handle feature heterogeneity; however, **we did not mean to imply that [1,2,4] are based on subgraphs for task unification**. In fact, these methods do not explicitly address the unification of node, link, and graph tasks but instead directly apply GNNs to learn node embeddings, which are subsequently used for model training. We acknowledge that the wording may have been misleading, and we have revised this section in the updated version to avoid any further misunderstandings.
>
> > W2: Limited Novelty in Task Trees as Unified Instances
> >
>
> Regarding your concern about the novelty of task-trees as unified instances, we’d like to provide a detailed comparison between subgraphs and task-trees. In summary, our proposed task-trees are more efficient and learnable compared to subgraphs, particularly in unifying various graph-related tasks and preserving generalities across different domains.
>
> **Subgraphs**: Some existing methods utilize subgraphs (typically k-hop subgraphs) as the basic learning instances [3,5,6]. For example, in node classification, these approaches extract ego-graphs around each node and assign the labels of the central nodes as labels for the induced subgraphs, transforming node-level classification into subgraph-level classification. This approach can also be adapted for link-level or graph-level tasks [5]. In general, these methods involve: (1) Extracting subgraphs around task-relevant nodes [5] and (2) Applying GNNs on each subgraph to obtain subgraph embeddings for predictions. However, these methods have two significant limitations: (1) **Efficiency Issue**: Extracting subgraphs incurs additional computational overhead, leading to increased time and memory usage due to the need to process and store these subgraphs. (2) **Learnability Issue**: The information preserved in subgraphs may not always be learnable by standard GNNs. Given the expressiveness limitations of message passing (bounded by the 1-WL test), these methods often struggle to capture essential substructures within subgraphs.
>
> **Task-Trees**: Compared to subgraphs, task-trees are both more efficient and learnable. The encoding of task-trees for a node/link/graph involves: (1) Adding virtual nodes to the original graphs and connecting them to task-relevant nodes. (2) Using GNNs over the augmented graph to encode the embeddings of these virtual nodes for predictions. For instance, in node classification, we first augment the original graph by adding virtual nodes connected to each node and perform GNNs on the new graph to learn virtual node embeddings for predictions. Thus, our method requires only the addition of nodes and edges to the original graph, making it **significantly more efficient than extracting and storing the subgraph**. Additionally, encoding task-trees is equivalent to directly encoding the virtual nodes via message passing, ensuring that the **information in task-trees is fully learnable by standard GNNs**. In conclusion, our task-tree approach offers both superior efficiency and learnability.
>
> **Efficiency Comparison (Section 5.4 of the Original Submission):** We conducted an empirical comparison between subgraphs and task-trees. For this, we implemented a subgraph version of our model, **GIT-SubG**, by replacing task-trees with subgraphs. Below is a comparison of time and memory usage (on a 48GB GPU) between GIT and GIT-SubG during pretraining (about 1,700,000 task-trees). To give a better understanding, using a batch size of 2048, **GraphMAE** requires 193 seconds per epoch with a memory allocation of 35%.
>
> | Batch Size | 512 | 1024 | 2048 | 4096 | 8192 | 16384 | 32768 |
> | --- | --- | --- | --- | --- | --- | --- | --- |
> | *GIT - Task-Tree* |  |  |  |  |  |  |  |
> | Time / Epoch (s) | 208 | 180 | 176 | 172 | 172 | 163 | 162 |
> | Memory Allocation (%) | 6 | 8 | 18 | 41 | 75 | 93 | 98 |
> | *GIT - SubG* |  |  |  |  |  |  |  |
> | Time / Epoch (s) | 280 | 243 | 234 | 223 | OOM | OOM | OOM |
> | Memory Allocation (%) | 21 | 39 | 74 | 97 | OOM | OOM | OOM |

---

> > ### Author Response · Authors · 2024-11-17
> > **Part 2**
> >
> > **Effectiveness Comparison (Section 5.4 of the Original Submission):** We also evaluated model performance across three domains: academic networks (node classification), knowledge graphs (edge classification), and molecule graphs (graph classification). Additionally, we compared our model against another popular subgraph-based method, **OFA** [3], which employs graph prompt learning for downstream tasks. The results (averaged over all graphs in each domain) are as follows.
> >
> > |  | OFA | GIT - SubG | **GIT** |
> > | --- | --- | --- | --- |
> > | Academia (node classification) | 72.18 | 73.48 | **75.82** |
> > | KG (edge classification) | 72.38 | 73.59 | **75.73** |
> > | Molecule (graph classification) | 74.03 | 72.67 | **75.73** |
> >
> > > W3: Semantic and Structural Gaps between Various Domains Remain Unaddressed
> > >
> >
> > Thank you for this observation. We would like to clarify that the primary objective of our work is not to fully address the semantic and structural gaps across different domains. Instead, **our focus is on providing a theoretical framework for defining a basic learning instance on graphs** that can potentially preserve generalities across various domains. This is a crucial and challenging step in designing graph foundation models, and to the best of our knowledge, our work is the first to investigate this problem from a theoretical perspective.
> >
> > While addressing the semantic and structural gaps is not the central focus of our paper, **we have nonetheless incorporated some techniques to mitigate these challenges**:
> >
> > - **Semantic Gap**: We adopt techniques from [3] by using textual encoders to unify node features for text-attributed graphs. For graphs without textual information (but with vector features), approaches like [1,2] can be seamlessly integrated with our GIT model to address semantic issues.
> > - **Structural Gap**: We acknowledge that different domains may exhibit varying structural patterns, such as triangles in social networks indicating strong social ties. To bridge structural distribution differences, we employ two strategies: (1) **Domain Regularizer**: We introduce a regularizer to minimize the embedding distance between different domains, which helps capture shared patterns across domains. The effectiveness of this regularizer is thoroughly evaluated in Appendix C.2. (2) **Specialization Process**: We design a lightweight specialization process that rapidly adapts the pretrained model to a specific domain, allowing it to better understand and leverage the unique structural patterns of that domain.
> >
> > ---
> >
> > We genuinely appreciate your feedback and hope these clarifications address your concerns. Please let us know if you have any further questions or require additional details.
> >
> > ---
> >
> > Reference:
> >
> > [1] All in one and one for all: A simple yet effective method towards cross-domain graph pretraining, KDD 24.
> >
> > [2] Text-Free Multi-domain Graph Pre-training: Toward Graph Foundation Models, Arxiv 24.
> >
> > [3] One for All: Towards Training One Graph Model for All Classification Tasks, ICLR 24.
> >
> > [4] GraphAny: A Foundation Model for Node Classification on Any Graph, Arxiv 24.
> >
> > [5] All in One: Multi-task Prompting for Graph Neural Networks, KDD 23.
> >
> > [6] PRODIGY: Enabling In-context Learning Over Graphs, NeurIPS 23.

---

### Official Review · Reviewer_WQqF · 2024-11-03

**Soundness:** 2
**Presentation:** 1
**Contribution:** 3
**Rating:** 6
**Confidence:** 2

**Summary:**

This paper proposes GIT, which learns generalities across graphs via a novel concept of task trees. The task trees are defined as the basic learning instances in graphs and they can preserve the generalities shared across graphs. The extensive experiments demosntrate the effectiveness of GIT in fine-tuning, in-context learning, and zero-shot learning.

**Strengths:**

1. This paper proposes a novel concept of task trees.
2. The authors show that task trees can preserve the generalities shared across graphs by several theorems and extensive experiments.

**Weaknesses:**

1. As pointed out by Line 90, message-passing GNNs---which have been used in many existing graph prompt methods---can fully capture task trees. Thus, the existing graph prompt methods may also implicitly use task trees (the definition may be different). Please compare GIT with the existing graph prompt methods in detail and explain why GIT can capture the task generalities better than the existing graph prompt methods.
2. I suggest moving the related work into the main body.

**Questions:**

See Weaknesses.

---

> ### Author Response · Authors · 2024-11-17
>
> Dear Reviewer WQqF,
>
> We sincerely appreciate your recognition of our contributions and your insightful feedback, which will help us further improve our work. Below, we address your concerns in detail:
>
> > W1: Comparison Between GIT and Existing Graph Prompt Methods
> >
>
> Thank you for pointing this out. We acknowledge that some concepts in existing graph prompt learning methods, such as induced graphs [1], data graphs [2], or NOI graphs [3], may appear to have similarities to our proposed task-trees. However, these methods fundamentally rely on **subgraphs**, not **task-trees**. In particular, these approaches involve: Extracting subgraphs (typically k-hop ego-graphs) around task-relevant nodes. Applying GNNs and graph prompt learning to each induced subgraph to learn subgraph representations, which are then used for predictions. For example, in node classification, existing methods extract ego-graphs around each node and assign labels to the induced subgraphs based on the central node, effectively transforming node-level classification into subgraph-level classification. However, subgraph-based methods have two inherent limitations: (1) **Efficiency Issue**: Extracting subgraphs introduces significant computational overhead, increasing both time and memory usage due to the need to process and store numerous subgraphs. (2) **Learnability Issue**: The information preserved in subgraphs may not always be fully captured by standard GNNs. Due to the expressiveness limitations of message passing (restricted by the 1-WL test), these methods often struggle to capture essential substructures within the subgraphs.
>
> Our proposed **task-tree** method fundamentally differs from the subgraph-based approaches. Task-trees offer both higher efficiency and improved learnability. The encoding process for task-trees involves: (1) **Virtual node augmentation**: Adding virtual nodes to the original graph and connecting them to task-relevant nodes. (2) **GNN encoding**: Using GNNs on the augmented graph to generate embeddings for these virtual nodes, which are then used for predictions. For instance, in node classification, we augment the graph by adding virtual nodes linked to each task-relevant node and use GNNs to learn embeddings for these virtual nodes. This approach is significantly more efficient since it only requires augmenting the original graph without extracting subgraphs. Moreover, since encoding task-trees is equivalent to encoding virtual nodes through message passing, the task-tree structure is fully learnable by standard GNNs.
>
> Conceptually, task-trees are similar to computation trees used in GFT [4], which defines transferable vocabularies for graph foundation models. However, our work goes further by providing a theoretical foundation that explains why tree structures (such as our task-trees and the computation trees in [4]) can serve as fundamental learning instances for graphs.
>
> **Effectiveness Comparison (Section 5.4 of the Original Submission):** We conducted an empirical comparison between subgraphs and task-trees. For this, we implemented a subgraph version of our model, **GIT-SubG**, by replacing task-trees with subgraphs. We evaluated model performance across three domains: academic networks (node classification), knowledge graphs (edge classification), and molecule graphs (graph classification). Additionally, we compared our model against another popular subgraph-based method, **OFA** [3], which employs graph prompt learning for downstream tasks. The results (averaged over all graphs in each domain) are as follows.
>
> |  | OFA | GIT - SubG | **GIT** |
> | --- | --- | --- | --- |
> | Academia (node classification) | 72.18 | 73.48 | **75.82** |
> | KG (edge classification) | 72.38 | 73.59 | **75.73** |
> | Molecule (graph classification) | 74.03 | 72.67 | **75.73** |
>
> > W2: Move Related Work into Main Body
> >
>
> Thank you for the suggestion. We will incorporate a section on related work into the main paper.
>
> ---
>
> Thank you once again for your valuable feedback. Please let us know if you have any further questions or need additional clarification.
>
> ---
>
> Reference:
>
> [1] All in One: Multi-task Prompting for Graph Neural Networks, KDD 23.
>
> [2] PRODIGY: Enabling In-context Learning Over Graphs, NeurIPS 23.
>
> [3] One for All: Towards Training One Graph Model for All Classification Tasks, ICLR 24.
>
> [4] GFT: Graph Foundation Model with Transferable Tree Vocabulary, NeurIPS 24.

---

> > ### Comment · Reviewer_WQqF · 2024-12-03
> >
> > Thanks for your rebuttal. Unfortunately, my concerns remain unaddressed. The suggestions and questions are as follows.
> >
> > 1. **Comparison in terms of Efficiency.** Please provide the computational complexity. Moreover,  I am not sure whether it is the main contribution of the proposed method.
> > 2. **Comparison in terms of Learnability.** The authors claim that the expressiveness limitations of message passing struggle to capture essential substructures within the subgraphs, while the authors use message passing-based GraphSage as the encoder. Can GraphSage capture essential information within the whole graph?
> > 3. I do not understand the key difference between the subgraphs and task-trees. In my opinion, the computation graph of the GNN always forms a tree structure as shown in Figure 5.1 in [Ref1], even if the computation process is restricted in a subgraph. 4. Can you provide a motivating example such that the performance proposed task-trees outperforms other methods?
> >
> >
> > [Ref1] Graph Representation Learning.

---

> > > ### Author Response · Authors · 2024-12-03
> > > **Additional Clarification between Task-trees and Subgraphs [P1]**
> > >
> > > Dear Reviewer WQqF,
> > >
> > > Thank you for your thoughtful response. As the rebuttal phase nears its conclusion, we will do our best to provide a quick yet comprehensive reply.
> > >
> > > Before diving into the point-by-point response, we would like to re-emphasize our main contribution: **introducing task-trees as a novel learning instance for graph foundation models (GFMs), unifying node-level, edge-level, and graph-level tasks.** To the best of our knowledge, this is **the first work to provide a theoretical foundation** demonstrating the potential existence of a basic learning instance capable of preserving cross-domain and cross-task generality on graphs. This framework lays a strong foundation for building scalable and transferable GFMs. While efficiency and learnability are important aspects of our work, **our primary contribution lies in establishing this theoretical framework.**
> > >
> > > For better readability, we have slightly reorganized the questions and responses.
> > >
> > > > W1: The key difference between the subgraphs and task-trees
> > > >
> > >
> > > Let us use node classification as an example to illustrate the distinction. **Subgraph-Based Methods** rely on subgraph-level embeddings for prediction, which involves: (1) Sampling k-hop subgraphs (in general) for each node and assigning the central node's label as the subgraph label. (2) Performing message passing on each subgraph to learn subgraph-level embeddings for prediction. **Task-Tree-Based Methods** use tree-level embeddings (technically node-level embeddings) for prediction, which involves: (1) Appending virtual nodes for each node (since we are focusing on node classification) on the original graph. (2) Using GNNs to learn the task-tree embeddings (technically virtual node embeddings) for predictions.
> > >
> > > Based on this distinction, the method referenced by the reviewer, stating that “the computation graph of the GNN always forms a tree structure as shown in Figure 5.1 in [Ref1], even if the computation process is restricted to a subgraph,” aligns more closely with our proposed task-tree approach rather than with existing subgraph-based methods. This highlights the core difference in how embeddings are generated and utilized in each framework.
> > >
> > > > W2: Comparison in terms of Learnability
> > > >
> > >
> > > Subgraph-based methods use GNNs to learn subgraph embeddings by pooling the embeddings of all nodes within extracted subgraphs. However, numerous studies [1,2,3] have demonstrated that subgraph-level (or graph-level) embeddings learned through message-passing GNNs are limited in their ability to capture certain fundamental substructures. This limitation stems from the well-known conclusion that message-passing GNNs are constrained by the expressiveness of the 1-WL test [4].
> > >
> > > In contrast, we observe that tasks on graphs often exhibit inherent tree structures, which are naturally learnable by message-passing GNNs [1,3]. Specifically, GraphSAGE can effectively capture the essential information preserved in task-trees, as its message-passing mechanism can naturally unfold into tree structures [3,5]. This property underpins the learnability of our proposed task-trees and distinguishes them from subgraph-based methods.
> > >
> > > [1] Generalization and representational limits of graph neural networks. In ICML, 2020
> > >
> > > [2] Can graph neural networks count substructures? In NeurIPS, 2020.
> > >
> > > [3] Beyond weisfeiler-lehman: A quantitative framework for gnn expressiveness. In ICLR, 2024.
> > >
> > > [4] How Powerful are Graph Neural Networks? In ICLR, 2019.
> > >
> > > [5] Tree Mover's Distance: Bridging Graph Metrics and Stability of Graph Neural Networks. In NeurIPS, 2022.
> > >
> > > > W3: Comparison in terms of Efficiency
> > > >
> > >
> > > **Complexity analysis:**
> > >
> > > - For **node classification**, we associate a virtual node with each node in the graph and learn the embeddings of these virtual nodes using GNNs as the embeddings of node-induced task-trees.
> > >
> > >     Complexity: Adding N virtual nodes O(N); adding N edges to connect each virtual node to its corresponding original node O(N). The overall time complexity of O(N).
> > >
> > > - For **edge classification**, we associate a virtual node with each edge by connecting it to the start and end nodes, allowing GNNs to learn the embeddings of these virtual nodes, as the embeddings of edge-induced task-trees.
> > >
> > >     Complexity: Adding E virtual nodes O(E), adding 2E edges to connect each virtual node to edges O(2E). The overall time complexity O(E)
> > >
> > > - For **graph classification**, we similarly add a virtual node for each graph, connecting it to all nodes on the graph, and use GNNs to learn the embedding of these virtual nodes, as the embeddings as graph-induced task-trees.
> > >
> > >     Complexity: Adding G virtual nodes O(G), adding N edges to connect virtual nodes to all nodes on the graph O(N). As the number of nodes in the dataset is much more than the number of graphs, the overall complexity is O(N).
> > >
> > > ---
> > >
> > > *To be continued*

---

> ### Author Response · Authors · 2024-12-03
> **Additional Clarification between Task-trees and Subgraphs [P2]**
>
> **Comparison to Subgraphs — A Case Study**: Our experiments (detailed below) indicate that subgraph encoding is the primary bottleneck. For instance, consider a graph with 100 nodes and 300 edges (average degree = 3). Sampling 2-hop ego-graphs for all nodes results in subgraphs with ~10 nodes and ~9 edges each. To encode these subgraphs simultaneously, they must be combined into a larger graph with 1,000 nodes and 900 edges. Then, they should conduct message passing on the larger graph to learn subgraph embeddings. By contrast, our task-tree approach simply adds 100 virtual nodes and 100 edges to the original graph, resulting in a graph with 200 nodes and 400 edges—making the message passing more efficient
>
> > W4: A motivating example
> >
>
> To motivate the concept of task-trees, we begin by considering the structures shared across different graph tasks. Existing subgraph-based methods typically extract subgraphs around task-relevant nodes to form induced graphs, unifying all graph tasks as graph classification. However, we approach this problem from a different perspective, focusing on the learning process of GNNs. Regardless of the graph task, predictions ultimately rely on the embeddings of task-relevant nodes. For instance, in graph classification, embeddings of all nodes in the graph are aggregated for prediction. Existing methods achieve this aggregation using a separate pooling module, but we ask: why not simply append virtual nodes to the original graphs and use message passing to directly mimic this aggregation process? By doing so, predictions for any graph-related task can be made directly through the embeddings of these virtual nodes.
>
> By adding virtual nodes to the original graph, we construct what we call **task-trees**, which stem from these virtual nodes. Conceptually, while existing methods unify graph tasks as graph classification, our approach unifies these tasks as **node classification**, specifically **task-tree classification**. This distinction simplifies the problem and allows us to leverage the natural compatibility of message passing with tree-like structures.
>
> Regarding the performance of task-trees, we hope our previous responses have provided sufficient case studies to highlight their advantages in terms of learnability and efficiency compared to subgraphs.
>
> ---
>
> We appreciate your thoughtful consideration and remain available to provide any further clarifications or additional insights as needed.

---

### Official Review · Reviewer_N4ou · 2024-11-05

**Soundness:** 3
**Presentation:** 3
**Contribution:** 3
**Rating:** 6
**Confidence:** 4

**Summary:**

This paper presents a solution towards graph foundation model which works in cross-task and cross-domain scenarios. With a task-tree-based appraoch to unify different graph tasks, along with a reconstruction-based pretraining objective, the authors provide theoretical analysis on the stability and transferability of their proposed method. By pretraining on a diverse set of graph datasets, the pretrained model is able to generalize to unseen graphs from various domains, and the specification process can further improve the performance.

**Strengths:**

1. The paper is well written and contains extensive information from theoretical and empirical perspectives.

2. Based on the experimental results, the proposed method achieves the goal of 'cross-task, cross-domain' transfer learning, which is quite impressive compared with existing GFM research.

**Weaknesses:**

I would urge the authors to provide a more detailed discussion of the difference between their proposed 'task-tree' method and existing subgraph-based methods.

**Questions:**

1. What are the differences between the proposed 'task-tree' method and existing subgraph-based methods? Could you explain this further, in terms of efficiency (i.e., why the task-tree method is more efficient), and effectiveness (i.e., how can the task-tree representation improve the performance of graph tasks).

2. The authors adopted a reconstruction-based objective for pretraining. Could you provide some intuitions to justify this choice? Is it possible to use a contrastive learning method instead?

3. In the fine-tuning experiments, did the model go through the specification process, or were they directly fine-tuned on specific downstream datasets after general pretraining?

---

> ### Author Response · Authors · 2024-11-17
> **Part 1**
>
> Dear Reviewer N4ou,
>
> We sincerely appreciate your recognition of our contributions and your insightful feedback, which will help us further refine our work. Below, we address your concerns in detail:
>
> > W1: Difference between Task-Trees and Subgraphs
> >
>
> **Subgraphs**: Some existing methods utilize subgraphs (typically k-hop subgraphs) as the basic learning instances [1,2,3]. For example, in node classification, these approaches extract ego-graphs around each node and assign the labels of the central nodes as labels for the induced subgraphs, transforming node-level classification into subgraph-level classification. This approach can also be adapted for link-level or graph-level tasks [1]. In general, these methods involve: (1) Extracting subgraphs around task-relevant nodes [1] and (2) Applying GNNs on each subgraph to obtain subgraph embeddings for predictions. However, these methods have two significant limitations: (1) **Efficiency Issue**: Extracting subgraphs incurs additional computational overhead, leading to increased time and memory usage due to the need to process and store these subgraphs. (2) **Learnability Issue**: The information preserved in subgraphs may not always be learnable by standard GNNs. Given the expressiveness limitations of message passing (bounded by the 1-WL test), these methods often struggle to capture essential substructures within subgraphs.
>
> **Task-Trees**: Compared to subgraphs, task-trees are both more efficient and learnable. The encoding of task-trees for a node/link/graph involves: (1) Adding virtual nodes to the original graphs and connecting them to task-relevant nodes. (2) Using GNNs over the augmented graph to encode the embeddings of these virtual nodes for predictions. For instance, in node classification, we first augment the original graph by adding virtual nodes connected to each node and perform GNNs on the new graph to learn virtual node embeddings for predictions. Thus, our method requires only the addition of nodes and edges to the original graph, making it **significantly more efficient than extracting and storing the subgraph**. Additionally, encoding task-trees is equivalent to directly encoding the virtual nodes via message passing, ensuring that the **information in task-trees is fully learnable by standard GNNs**. In conclusion, our task-tree approach offers both superior efficiency and learnability.
>
> **Efficiency Comparison (Section 5.4 of the Original Submission):** We conducted an empirical comparison between subgraphs and task-trees. For this, we implemented a subgraph version of our model, **GIT-SubG**, by replacing task-trees with subgraphs. Below is a comparison of time and memory usage (on a 48GB GPU) between GIT and GIT-SubG during pretraining (about 1,700,000 task-trees). To give a better understanding, using a batch size of 2048, **GraphMAE** requires 193 seconds per epoch with a memory allocation of 35%.
>
> | Batch Size | 512 | 1024 | 2048 | 4096 | 8192 | 16384 | 32768 |
> | --- | --- | --- | --- | --- | --- | --- | --- |
> | *GIT - Task-Tree* |  |  |  |  |  |  |  |
> | Time / Epoch (s) | 208 | 180 | 176 | 172 | 172 | 163 | 162 |
> | Memory Allocation (%) | 6 | 8 | 18 | 41 | 75 | 93 | 98 |
> | *GIT - SubG* |  |  |  |  |  |  |  |
> | Time / Epoch (s) | 280 | 243 | 234 | 223 | OOM | OOM | OOM |
> | Memory Allocation (%) | 21 | 39 | 74 | 97 | OOM | OOM | OOM |
>
> **Effectiveness Comparison (Section 5.4 of the Original Submission):** We also evaluated model performance across three domains: academic networks (node classification), knowledge graphs (edge classification), and molecule graphs (graph classification). Additionally, we compared our model against another popular subgraph-based method, **OFA** [3], which employs graph prompt learning for downstream tasks. The results (averaged over all graphs in each domain) are as follows.
>
> |  | OFA | GIT - SubG | **GIT** |
> | --- | --- | --- | --- |
> | Academia (node classification) | 72.18 | 73.48 | **75.82** |
> | KG (edge classification) | 72.38 | 73.59 | **75.73** |
> | Molecule (graph classification) | 74.03 | 72.67 | **75.73** |

---

> > ### Author Response · Authors · 2024-11-17
> > **Part 2**
> >
> > > W2: The Intuitions of Using Reconstruction Loss Function
> > >
> >
> > We chose the reconstruction-based method as a fundamental self-supervised training approach. The loss function we employed is inspired by [4], which uses a negative-free contrastive learning framework for improved training efficiency. The key intuition is that if the model can recognize similarities between two corrupted versions of a task-tree in a predictive manner, it will effectively **capture corruption-invariant information from the original task-tree**. This aligns with the underlying philosophy of contrastive learning, which also aims to capture invariant features.
> >
> > That said, our approach is flexible, and **it is certainly feasible to incorporate other contrastive loss** **functions**, such as those in [5, 6], or even more sophisticated reconstruction methods like [7].
> >
> > > W3: Did the model go through the specification process, or were they directly fine-tuned on specific downstream datasets after general pretraining?
> > >
> >
> > We explored both strategies: (1) **GIT-G** refers to the model pretrained on general datasets and then directly fine-tuned on specific downstream tasks. (2) **GIT-S** refers to the model that undergoes a specification process before fine-tuning on specific datasets. We provide comprehensive results for both approaches in Appendix F of the original submission. Additionally, in Figures 3 and 4, we report the best performance between GIT-G and GIT-S as the best performance of GIT for clearer visualization.
> >
> > ---
> >
> > Thank you once again for your valuable feedback. Please let us know if you have any further questions or require additional clarification.
> >
> > ---
> >
> > Reference:
> >
> > [1] All in One: Multi-task Prompting for Graph Neural Networks, KDD 23.
> >
> > [2] PRODIGY: Enabling In-context Learning Over Graphs, NeurIPS 23.
> >
> > [3] One for All: Towards Training One Graph Model for All Classification Tasks, ICLR 24.
> >
> > [4] Large-scaled Representation Learning on Graphs via Bootstrapping, ICLR 22.
> >
> > [5] Deep Graph Contrastive Representation Learning, Arxiv 20.
> >
> > [6] Graph Contrastive Learning with Adaptive Augmentation, WWW 21.
> >
> > [7] GraphMAE: Self-supervised Masked Graph Autoencoders, KDD 22.

---

### Author Response · Authors · 2024-11-17

We sincerely thank the reviewers for their valuable feedback and constructive suggestions.

We would like to take this opportunity to reiterate our contributions: In the context of graph foundation models, identifying general patterns across graphs (like shared tokens in languages) from diverse domains is essential. **To the best of our knowledge, this work is the first to theoretically demonstrate the existence of a graph learning instance, which we term the "task-tree", that preserves these cross-domain generalities.**

Some reviewers raised concerns that our proposed task-tree is similar to subgraphs (or equivalent concepts like prompt graphs, contextual graphs, NOI graphs, etc., found in existing literature), and therefore questioned the novelty of our approach. However, **our proposed task-tree concept is fundamentally different from subgraphs**, as we will clarify below.

> Difference between task-trees and subgraphs
>

**Subgraphs**: Some existing methods utilize subgraphs (typically k-hop subgraphs) as the basic learning instances [1,2,3]. For example, in node classification, these approaches extract ego-graphs around each node and assign the labels of the central nodes as labels for the induced subgraphs, transforming node-level classification into subgraph-level classification. This approach can also be adapted for link-level or graph-level tasks [1]. In general, these methods involve: (1) Extracting subgraphs around task-relevant nodes [1] and (2) Applying GNNs on each subgraph to obtain subgraph embeddings for predictions. However, these methods have two significant limitations: (1) **Efficiency Issue**: Extracting subgraphs incurs additional computational overhead, leading to increased time and memory usage due to the need to process and store these subgraphs. (2) **Learnability Issue**: The information preserved in subgraphs may not always be learnable by standard GNNs. Given the expressiveness limitations of message passing (bounded by the 1-WL test), these methods often struggle to capture essential substructures within subgraphs.

**Task-Trees**: Compared to subgraphs, task-trees are both more efficient and learnable. The encoding of task-trees for a node/link/graph involves: (1) Adding virtual nodes to the original graphs and connecting them to task-relevant nodes. (2) Using GNNs over the augmented graph to encode the embeddings of these virtual nodes for predictions. For instance, in node classification, we first augment the original graph by adding virtual nodes connected to each node and perform GNNs on the new graph to learn virtual node embeddings for predictions. Thus, our method requires only the addition of nodes and edges to the original graph, making it **significantly more efficient than extracting and storing the subgraph**. Additionally, encoding task-trees is equivalent to directly encoding the virtual nodes via message passing, ensuring that the **information in task-trees is fully learnable by standard GNNs**. In conclusion, our task-tree approach offers both superior efficiency and learnability.

**Efficiency Comparison (Section 5.4 of the Original Submission):** We conducted an empirical comparison between subgraphs and task-trees. For this, we implemented a subgraph version of our model, **GIT-SubG**, by replacing task-trees with subgraphs. Below is a comparison of time and memory usage (on a 48GB GPU) between GIT and GIT-SubG during pretraining (about 1,700,000 task-trees).

| Batch Size | 512 | 1024 | 2048 | 4096 | 8192 | 16384 | 32768 |
| --- | --- | --- | --- | --- | --- | --- | --- |
| *GIT - Task-Tree* |  |  |  |  |  |  |  |
| Time / Epoch (s) | 208 | 180 | 176 | 172 | 172 | 163 | 162 |
| Memory Allocation (%) | 6 | 8 | 18 | 41 | 75 | 93 | 98 |
| *GIT - SubG* |  |  |  |  |  |  |  |
| Time / Epoch (s) | 280 | 243 | 234 | 223 | OOM | OOM | OOM |
| Memory Allocation (%) | 21 | 39 | 74 | 97 | OOM | OOM | OOM |

**Effectiveness Comparison (Section 5.4 of the Original Submission):** We also evaluated model performance across three domains: academic networks (node classification), knowledge graphs (edge classification), and molecule graphs (graph classification). Additionally, we compared our model against another popular subgraph-based method, **OFA** [3], which employs graph prompt learning for downstream tasks. The results (averaged over all graphs in each domain) are as follows.

|  | OFA | GIT - SubG | **GIT** |
| --- | --- | --- | --- |
| Academia (node classification) | 72.18 | 73.48 | **75.82** |
| KG (edge classification) | 72.38 | 73.59 | **75.73** |
| Molecule (graph classification) | 74.03 | 72.67 | **75.73** |

---

Reference:

[1] All in One: Multi-task Prompting for Graph Neural Networks, KDD 23.

[2] PRODIGY: Enabling In-context Learning Over Graphs, NeurIPS 23.

[3] One for All: Towards Training One Graph Model for All Classification Tasks, ICLR 24.

---

### Meta-Review · Area_Chair_PdHR · 2024-12-22

**Metareview:**

The submission proposes a task-tree framework for graph foundation models, which aims to unify node, edge, and graph-level tasks. The reviewers have raised concerns regarding the novelty of the proposed task-tree concept, its differentiation from existing methods, and the effectiveness of the theoretical claims. While the reviewers acknowledged the paper's strengths, no reviewer strongly championed acceptance, and one reviewer explicitly recommended rejection; thus, I lean towards rejecting this submission.

**Additional Comments On Reviewer Discussion:**

The discussion phase highlighted several critical concerns, including questions about the novelty of task-trees compared to subgraph-based methods, the theoretical claims regarding learnability, and the practical impact of the proposed framework. The authors provided detailed responses, offering complexity analyses, additional empirical results, and theoretical clarifications to differentiate their approach. They also revised the manuscript to improve the presentation and integrate related work into the main text. Despite these efforts, the explanations did not fully address all the concerns of the reviewers, leading to the decision.

---

### Decision · Program_Chairs · 2025-01-22

Reject